



# Description and evaluation of a detailed gas-phase chemistry scheme in the TM5-MP global chemistry transport model (r112)

Stelios Myriokefalitakis[1], Nikos Daskalakis[2], Angelos Gkouvousis[3,1], Andreas Hilboll[†2], Twan van Noije[4], Jason E. Williams[4], Philippe Le Sager[4], Vincent Huijnen[4], Sander Houweling[5,6], Tommi Bergman[7], Johann Rasmus Nüß[2], Mihalis Vrekoussis[2,8,9], Maria Kanakidou[2,3] and Maarten C. Krol[10,11]

[1] Institute for Environmental Research and Sustainable Development (IERSD), National Observatory of Athens, Penteli, Greece
[2] Institute of Environmental Physics, University of Bremen, Bremen, Germany
[3] Environmental Chemical Processes Laboratory (ECPL), Department of Chemistry, University of Crete, Heraklion, Greece
[4] Royal Netherlands Meteorological Institute (KNMI), De Bilt, The Netherlands
[5] Department of Earth Sciences, Vrije Universiteit Amsterdam, The Netherlands
[6] SRON Netherlands Institute for Space Research, Utrecht, The Netherlands
[7] Finnish Meteorological Institute, Climate System Research, Helsinki, Finland
[8] Center for Marine Environmental Sciences, University of Bremen, Bremen, Germany
[9] Energy, Environment and Water Research Center (EEWRC), The Cyprus Institute, Cyprus
[10] Wageningen University, Wageningen, The Netherlands
[11] Institute for Marine and Atmospheric Research (IMAU), Utrecht University, Utrecht, The Netherlands

*Correspondence to*: Stelios Myriokefalitakis (steliosm@noa.gr) and Maarten C. Krol (maarten.krol@wur)

**Abstract.** This work documents and evaluates the tropospheric gas-phase chemical mechanism MOGUNTIA in the three-dimensional chemistry transport model TM5-MP. Compared to the modified CB05 chemical mechanism previously used in the model, the MOGUNTIA includes a detailed representation of the light hydrocarbons (C1-C4) and isoprene, along with a simplified chemistry representation of terpenes and aromatics. Another feature implemented in TM5-MP for this work is the use of the Rosenbrock solver in the chemistry code, which can replace the classical Euler Backward Integration method of the model. Global budgets of ozone ($O_3$), carbon monoxide (CO), hydroxyl radicals (OH), nitrogen oxides ($NO_X$) and volatile organic compounds (VOCs) are here analyzed and their mixing ratios are compared with a series of surface, aircraft and satellite observations for the year 2006. Both mechanisms appear to be able to represent satisfactorily observed mixing ratios of important trace gases, with the MOGUNTIA chemistry configuration yielding lower biases compared to measurements in most of the cases. However, the two chemical mechanisms fail to reproduce the observed mixing ratios of light VOCs, indicating insufficient primary emission source strengths, too weak vertical mixing in the boundary layer, and/or a low bias in the secondary contribution of C2-C3 organics via VOC atmospheric oxidation. Relative computational memory and time requirements of the different model configurations are also compared and discussed. Overall, compared to other chemistry schemes in use in global CTMs, the MOGUNTIA scheme simulates a large suite of oxygenated VOCs that are observed in the atmosphere at significant levels and are involved in aerosol formation, expanding, thus, the applications of TM5-MP.

---

[†] Deceased 25 March 2020





# 1 Introduction

Chemistry transport models (CTMs) are tools to effectively study the temporal and spatial evolution of atmospheric species at regional and global scales, as well as to understand how the main physical and chemical processes in the troposphere (e.g., emissions, chemistry, transport, and deposition) influence air quality. Model investigations and analyses of the changes of important tropospheric pollutants, such as ozone ($O_3$) and carbon monoxide (CO), can further provide essential information about the oxidative capacity of the atmosphere and thus the lifetime of important climate gases like methane ($CH_4$). The oxidative capacity also controls the rate of formation and growth of aerosols by conversion of sulfur oxides into particulate sulfate ($SO_4^{2-}$) and volatile organic compounds (VOCs) into condensable organic matter that forms organic particles. Under certain tropospheric conditions (e.g., intense sunlight and high temperatures) the oxidation of VOCs in the presence of nitrogen oxides ($NO_X \equiv NO + NO_2$) enhances the formation of secondary pollutants, such as $O_3$ (Crutzen, 1974; Derwent et al., 1996; Monks et al., 2009). VOCs and $NO_X$ arise from both natural and anthropogenic emission sources. $NO_X$ can be further converted into other chemical species such as $HNO_3$ and particulate nitrate ($NO_3^-$), that together with $SO_4^{2-}$ are key contributors to atmospheric acidity. The photochemical production of tropospheric $O_3$, a known toxic air pollutant that is transported over long distances, depends on the $NO_X$ and VOC availability in a nonlinear manner (e.g., Seinfeld and Pandis, 2006). Under very high $NO_X$ conditions, common in densely populated areas (i.e., VOC-limited regimes), the $O_3$ production is inhibited and reductions in $NO_X$ emissions can locally increase $O_3$. In contrast, in rural areas, the $O_3$ production is more efficient, and $NO_X$ emission reductions will decrease $O_3$ (i.e., $NO_X$-limited regimes). Thus, changes in emissions of $NO_X$ and VOC may lead to nonlinear responses in ozone and the oxidation capacity of the troposphere. Overall, understanding the photochemical processes in the troposphere via robust model simulations is key to the development of effective abatement strategies on pollutants that affect both air quality and climate, as well as to the prediction of the future atmospheric composition.

The gas-phase photochemistry in the troposphere consists of numerous and complex reactions between odd oxygen ($O_X \equiv O + O_3$) and $NO_X$, coupled to the oxidation of various VOCs (e.g., Atkinson, 2000; Atkinson et al., 2004, 2006). Several chemical mechanisms of varying complexity in the representation of VOC oxidation are currently included in state-of-the-art CTMs. One of the most explicit mechanisms ever built for the simulation of the tropospheric VOC oxidation cycles, the Master Chemical Mechanism (MCM v3), comprises more than 12690 reactions, involving more than 4350 organic species, and about 46 associated inorganic reactions (Jenkin et al., 1997a; 2003). Note that recent updates further include detailed aromatic hydrocarbon (Bloss et al., 2005) and isoprene oxidation (Jenkin et al., 2015) mechanisms. Since this level of chemical complexity is far beyond the computational resources potentially available for three-dimensional (3-D) global tropospheric CTMs, simplifications are required that retain the essential features of the chemistry. To this end, various chemical mechanisms of tropospheric chemistry have been developed with different levels of complexity, involving mainly reductions of the number of VOCs considered by lumping organic species into representative surrogates. For example, the Statewide Air Pollution Research Center (SAPRC) mechanism (SAPRC-99) is a well-documented gas-phase chemical mechanism used in many CTMs, including a rather detailed representation of tropospheric VOC oxidation based on an evaluation against over 1700





experiments performed in different smog chambers (e.g., Carter, 1995, 2010). SAPRC does not model the oxidation of each VOC individually as the MCM, but it uses a molecular lumping approach to assign VOCs to a smaller number of reactive species. Other well-documented mechanisms often used in CTMs are the Regional Acid Deposition Model chemical mechanism (RADM; e.g., Gross and Stockwell, 2003; Stockwell et al., 1997), the Regional Atmospheric Chemistry

Mechanism (RACM; e.g., Geiger et al., 2003; Goliff et al., 2013; Stockwell et al., 1997), and the Model of Ozone and Related Chemical Tracers mechanism (MOZART; Emmons et al., 2010; Horowitz et al., 2003). A molecular lumping mechanism has been also developed and initially used in the Model of the Global Universal Tracer transport In the Atmosphere (MOGUNTIA) 3-D climatological CTM (e.g., Kanakidou and Crutzen, 1999; Poisson et al., 2000; Baboukas et al., 2000), as well as in box model applications for field data interpretation (e.g., Poisson et al., 2001a; Vrekoussis et al., 2006); that latter chemical

mechanism has been the starting point for the model development presented here. Another mechanism that has been extensively used in numerous chemistry and climate modeling studies is the Carbon Bond Mechanism (CBM). CBM has several different versions with different levels of complexity (e.g., reaction rate constants updates, additions of inorganic reactions, as well as additions of organic species to better represent the respective species and radicals in the atmosphere), such as the CBM-IV (e.g., Gery et al., 1988; Houweling et al., 1998; Luecken et al., 2008), the CBM 2005 (CB05; e.g., Yarwood

et al., 2005; Williams et al., 2013, 2017; Flemming et al., 2015) and the CBM-Z (Zaveri and Peters, 1999). The lumped-structure approach of the CBM has been extensively evaluated against chamber studies (e.g., Yarwood et al., 2005).

Several studies focused on the impact of the chemical complexity of the gas-phase mechanism on tropospheric simulations, indicate an inevitable compromise between model accuracy and computational efficiency (e.g., Cai et al., 2011; Gross and Stockwell, 2003; Luecken et al., 2008; Sander et al., 2019). Indeed, for a given atmospheric condition, even different versions

of the same mechanism (e.g., the CBM family) can give significantly different results. For instance, the more explicit representation of VOCs in CB05 leads to a higher production of $O_3$ compared to the more lumped CBM-IV mainly due to a respective higher production of peroxy radicals, aldehydes and organic peroxides (Saylor and Stein, 2012). A comparison of CB05 with RACM (Kim et al., 2009) revealed that the most considerable differences appeared in areas with significant biogenic emissions, due to the more complex chemistry of aldehydes in the presence of anthropogenic alkenes and alkanes.

Box-model comparisons between the MCM and various state-of-the-art simplified tropospheric chemistry schemes also indicated that the differences between the chemistry schemes could be rather significant under high VOC loadings (Emmerson and Evans, 2009). The choice, thus, of a gas-phase mechanism used in a model may introduce uncertainties in predictions of regulated gas-phase pollutants (e.g., Knote et al., 2015). Computational restrictions however, such as memory and computing time savings, are always a critical point to consider for large-scale 3-D simulations, especially when higher spatial resolutions

are applied. On the other hand, the ability to validate the results of a particular chemical scheme in a global model can be significantly higher for the more extensive schemes that provide a rather explicit treatment of gases, such as in comparisons with satellite retrievals and *in situ* observations of a series of individual species.

In this work, a rather detailed and complete chemistry scheme is implemented in the global CTM TM5-MP, the massively parallel (MP) version of TM5 (Tracer Model version 5), with the aim to investigate whether the consistent biases in important





tropospheric tracers, such as $O_3$, CO, OH, $NO_X$ and light VOCs, found in previous work (e.g., Huijnen et al., 2010; van Noije et al., 2014; Williams et al., 2013, 2017) are sensitive to the chemistry scheme that is used. For this, we use the well-documented tropospheric gas-phase chemistry scheme MOGUNTIA (e.g., Myriokefalitakis et al., 2008 and refs. therein; along with recent updates), and benchmark its performance in TM5-MP. Section 2 provides a short description of the current model

version, focusing in particular on the new features implemented in the gas-phase chemistry and the chemistry integration method. Note that here we are mostly focusing on the performance of the new chemical scheme in comparison to the scheme previously included in the model (i.e., the modified CB05, mCB05; Williams et al., 2017), rather than the full description of the model itself which has been already described in detail in previous publications (e.g., Huijnen et al., 2010; Williams et al., 2013, 2017). In Sect. 3, the model's performance is analyzed for the different chemical configurations used for this study and

in Sect. 4 a detailed budget analysis of important gas-phase species is presented. Section 5 presents the evaluation of the different configurations of this work. The model's ability to reproduce the variability of important tropospheric species in both space and time is discussed, along with the associated uncertainties in atmospheric burdens and lifetimes. Finally, in Sect. 6 the main conclusions are presented, and some of the benefits and drawbacks of both chemical mechanisms are discussed, together with proposed directions for future model development.

## 2 Model description

### 2.1 General

The well-documented 3-D global CTM TM5 (Krol et al., 2005) is used for this study. Historically, the model has evolved from the original TM2 model (Heimann et al., 1988), via the TM3 model (Dentener et al., 2003; Houweling et al., 1998; Tsigaridis and Kanakidou, 2003), to TM4 (van Noije, 2004; Myriokefalitakis et al., 2008; Daskalakis et al., 2015) and TM5 (Krol et al.,

2005; Huijnen et al., 2010; van Noije et al., 2014; Williams et al., 2017). In TM5-MP the parallelization of the TM5 model has been redesigned, allowing for affordable global simulations at high resolution, i.e., 1°x1° globally (Williams et al., 2017). In this new MP version, the two-way zoom capability of TM5 is no longer available in the MP version. All applications of TM5 share the same methods for model discretization and operator splitting (Krol et al., 2005), the treatment of the meteorological fields, and the mass conserving tracer transport (Bregman et al., 2003). For the photolysis reactions, the model

utilizes a rather sophisticated online scheme, as described in Williams et al. (2012).

TM5-MP is primarily designed for simulation of the troposphere (i.e., no explicit stratospheric chemistry is considered in the model). To capture stratospheric ozone effects on actinic fluxes and to ensure realistic ozone stratosphere-troposphere exchange (STE), the overhead stratospheric profile is nudged to the ozone data set provided for the Coupled Model Intercomparison Project phase 6 (CMIP6; van Noije et al., manuscript in preparation). The boundary conditions for $CH_4$, both

in the lower troposphere and the stratosphere, are also based on the respective global mean value from CMIP6 data set (see also Sect 2.4) to scale the monthly 2-D climatological fields as derived from HALOE measurements (Grooß and Russell, 2005), with the same nudging heights and relaxation times as for the case of stratospheric $O_3$. Additionally, for $HNO_3$ and CO



in the stratosphere monthly mean latitudinal climatologies derived from ODIN space-based observations are applied by prescribing the ratio of $HNO_3/O_3$ (Jégou et al., 2008; Urban et al., 2009) and $CO/O_3$ (Dupuy et al., 2004), respectively. Thus, since TM5-MP does not consider a full stratospheric chemistry scheme, this study is mostly focused on the troposphere. For this, when presenting the tropospheric chemical budgets, a tropopause definition using the $O_3$ mixing ratio threshold of 150

5    ppb (e.g., see van Noije et al., 2014; Stevenson et al., 2006) is applied.

The gas-phase chemistry of the TM5-MP model is supplemented with the in-cloud oxidation of $SO_2$ through aqueous-phase reactions with $H_2O_2$ and $O_3$, that depend on the acidity of the solution (Dentener and Crutzen, 1993). The heterogeneous conversion of $N_2O_5$ into $HNO_3$ on the available surface area of cloud droplets, cirrus particles, and hydrated sulfate aerosols is also accounted for. For cloud droplets, the number of droplets per unit volume is calculated using the liquid water content

provided in the ECMWF meteorological data used by TM5-MP, assuming an effective droplet radius of 8 μm. For the heterogeneous conversion of $N_2O_5$ on hydrated sulfate particles, the approach of Dentener and Crutzen (1993) is employed, using a global mean reaction probability (γ value) of 0.02 and 0.01 on water and ice surfaces, respectively. Heterogeneous conversions also consider the total reactive surface area density (SAD) from aerosols, with contributions to accumulation mode aerosol from sulfate, nitrate, and ammonium being calculated by the EQuilibrium Simplified Aerosol Model (EQSAM)

approach (Metzger et al., 2002). The distribution of these aerosol species is calculated online and coupled to the gas-phase precursors $NH_3$, $H_2SO_4$, and $HNO_3$. Note that the aerosol microphysics module M7 (Vignati et al., 2004) is used in the model, as described in Aan de Brugh et al. (2011) and van Noije et al. (2014), along with recent updates on the inclusion of secondary organic aerosols (van Noije et al., manuscript in preparation). For $N_2O_5$, the uptake coefficient (γ) is considered as a function of temperature and relative humidity (Evans and Jacob, 2005), whilst for $HO_2$ and $NO_3$ radicals fixed γ values of 0.06 and

$10^{-3}$, respectively, are adopted across all aerosol types (Jacob, 2000).

The model considers the wet removal of atmospheric species by liquid and ice precipitation, by both in-cloud and below-cloud scavenging. The fraction of gases removed by precipitation depends on Henry's law (see Table S1 in the supplement), together with the dissociation constants, temperature, and liquid or ice water content. In-cloud scavenging in stratiform precipitation considers an altitude dependent precipitation formation rate (also describing the conversion of cloud water into rainwater). For

convective precipitation, highly soluble gases are assumed to be scavenged entirely in the vigorous convective updrafts producing rainfall rates of >1 mm/hour and exponentially scaled down for lower rainfall rates. For the dry deposition, the removal is calculated online in the model, based on a series of surface and atmospheric resistances on a 1°×1° spatial resolution (Wesely, 1989; Ganzeveld and Lelieveld, 1995; Ganzeveld et al., 1998). Overall, the calculated deposition velocities show both seasonal and diurnal cycles using 3-hourly meteorological and surface parameters, based on the uptake resistances for

vegetation (in-canopy aerodynamic, soil, and leaf resistance), soil, water, snow, and ice (see Table S2). A more detailed description of dry and wet deposition schemes for the removal of gases can be found in de Bruine et al. (2017).





### 2.1 Gas-phase chemistry

### 2.1.1. The original MOGUNTIA chemical scheme

The new chemical mechanism employed in TM5-MP for this study was originally developed for box (Poisson et al., 2001b) and global (Kanakidou and Crutzen, 1999; Poisson et al., 2000) modelling studies, and initially coupled to the global 3-D CTM

MOGUNTIA (Zimmermann, 1988). Since then, the scheme has been continuously updated for box modelling, coupled to the global TM4 model, and applied in numerous studies (e.g., Tsigaridis and Kanakidou, 2002; Gros et al., 2002; Myriokefalitakis et al., 2008; Daskalakis et al., 2015).

The MOGUNTIA chemical scheme employs a rather detailed oxidation scheme of light alkanes ($CH_4$, $C_2H_6$, and $C_3H_8$), light alkenes ($C_2H_4$ and $C_3H_6$), acetylene ($C_2H_2$), and isoprene ($C_5H_8$). Acetaldehyde ($CH_3CHO$), glyoxal (GLY; CHOCHO),

glycolaldehyde (GLYAL; HOCHCHO), methylglyoxal (MGLY; $CH_3COCHO$) and acetone ($CH_3COCH_3$) are also explicitly treated in the mechanism. The oxidation pathways of methacrolein (MACR; $CH_3(CH_2)CH=O$) and methylvinyl ketone (MVK; $CH_3C(O)CH=CH_2$) are considered together with the formation of formic (HCOOH) and acetic acid ($CH_3COOH$) in the mechanism. Higher VOCs (i.e., $C_{n>4}$), besides isoprene, are mainly represented in the mechanism by the surrogate species n-butane (n-$C_4H_{10}$), motivated by the similar $O_X$ and hydrogen oxides ($HO_X$) yields per oxidized carbon atom (e.g., see Poisson

et al., 2000; Stavrakou et al., 2009a). The second-generation oxidation products of higher hydrocarbons of biogenic origin (such as terpenes) and aromatics are also considered to follow the gas-phase oxidation pathways of the respective isoprene and surrogate n-$C_4H_{10}$ oxidation species.

The reactions of peroxy radicals ($RO_2$) with hydrogen peroxide ($HO_2$) and methyl peroxide ($CH_3O_2$) as well as with NO lead to organic hydroperoxides (ROOH), carbonyls and organic nitrates, respectively. ROOH is removed by photolysis and reaction

with OH. The subsequent addition of NO to the formed $RO_2$ radicals leads to alkyl nitrates ($RONO_2$), which are much longer lived than $NO_X$. $RONO_2$ can thus be transported over longer distances than $NO_X$ and serve as a sink for $NO_X$ in high-$NO_X$ areas and as a source for $NO_X$ in low-$NO_X$ regions. The $RONO_2$ compounds explicitly considered in this study are identified by R=$CH_3$, $C_2H_5$, $C_3H_7$, $C_4H_9$, $HOC_2H_4O$, and $C_5H_8(OH)$, i.e., the first-generation product of isoprene oxidation. Additionally, the reactions of the acyl peroxy radicals ($RC(O)O_2$) with $NO_2$ produce peroxyacyl nitrates ($RC(O)O_2NO_2$), in particular PAN

(R=$CH_3$), which is the most abundant organic nitrate observed in the troposphere and the only species of this group that is considered here. Thermal decomposition is dominant for peroxyacyl nitrates, while it is negligible for alkyl nitrates. $NO_3$ radical reactions with aldehydes, alcohols, n-$C_4H_{10}$, dimethylsulfide (DMS) and unsaturated hydrocarbons are also considered. A more detailed description of the chemical scheme used for this study can be found in Poisson et al. (2000) and Myriokefalitakis et al. (2008).

### 2.1.2 Updates of the MOGUNTIA chemical mechanism

Several updates have been applied here to the original MOGUNTIA chemical scheme compared to the previous implementations (e.g., Poisson et al., 2000; Myriokefalitakis et al., 2008, 2011), regarding the reactions and their rate constants,





as well as some oxidation paths of major hydrocarbons. Concerning the terpene chemistry, we here consider one lumped monoterpene species ($C_{10}H_{16}$) for all terpenes (assuming a 50:50 $\alpha$-:$\beta$-pinene distribution), in contrast to the consideration of the explicit oxidation of $\alpha$- and $\beta$-pinene as performed in the previous implementations of the MOGUNTIA scheme (e.g., Myriokefalitakis et al., 2008, 2010, 2016). Thus, monoterpenes represent here all terpenes and terpenoids species. Likewise,

toluene is used to represent all aromatics replacing benzene, xylene, and toluene used previously (Myriokefalitakis et al., 2008, 2016). Besides these compounds, toluene is also used to represent trimethyl-benzenes and higher aromatics. Moreover, for this work the coupling of the gas-phase chemistry with the aqueous-phase oxidation scheme of $SO_2$, as well as the gas-phase oxidation of dimethyl sulfide (DMS), methyl sulfonic acid (MSA) and ammonia ($NH_3$), follows the oxidation scheme outlined by Williams et al. (2013), which is slightly simpler compared to that used in previous studies using MOGUNTIA (see

Myriokefalitakis et al., 2010). Note that the lumping mentioned above, and the simplifications implemented here, aim at limiting the number of species without however degrading the general performance of the chemical scheme for global-scale tropospheric chemistry.

Isoprene (2-methyl-1,3-butadiene; ISOP) oxidation has been extended here with the production of isoprene epoxydiols (IEPOX) and hydroperoxyaldehydes (HPALD), as well as the $HO_X$-recycling mechanism under low-$NO_X$ conditions (Paulot

et al., 2009; Peeters and Müller, 2010a; Crounse et al., 2011; Browne et al., 2014). The latter species replaces the lumped second-generation oxidation product considered in previous implementations of the MOGUNTIA mechanism (Poisson et al., 2000; Myriokefalitakis et al., 2008). The oxidation of isoprene by the OH radical leads to the formation of several isomers of an unsaturated hydroxy hydroperoxide. In the presence of $NO_X$, this leads to the formation of carbonyl compounds. However, under low-$NO_X$ conditions, the major product from unsaturated hydroxy hydroperoxides oxidation is the IEPOX (i.e., cis- and

trans-isomers). The organic peroxy radicals formed from OH oxidation of isoprene, can react with either 1) $HO_2$ to form hydroperoxides, or 2) NO to form hydroxynitrates, formaldehyde (HCHO), MVK, MACR and $HO_2$ (e.g., Paulot et al., 2009), or hydroperoxyenals (HPALDs). The latter are produced by the isomerisation of the initial isoprene organic hydroperoxy radicals followed by reaction with $O_2$ and other oxidized products (Peeters et al., 2009; Peeters and Müller, 2010). Under $HO_2$-dominated conditions, the main products are unsaturated hydroperoxides (all possible isomers referred to as ISOPOOH; see

Table 2). The fate of isoprene peroxy radicals is highly dependent on the mixing ratios of $HO_2$, NO, the organic peroxy radicals, and the local meteorological conditions that affect the thermal and photochemical reaction rates as well as dry and wet removal of intermediates. Subsequent reactions of ISOPOOH with OH produce epoxydiols (cis- and trans- isomers referred to as IEPOX) and regenerate OH radicals (Paulot et al., 2009). Moreover, the isoprene peroxy radical 1,6-H-shift isomerizations (Peeters et al., 2014; Peeters and Müller, 2010) leads to the formation of photolabile C5-hydroperoxyaldehydes (i.e., all

possible isomers referred to as HPALDs; see Table 1). Overall, these additions to the chemistry scheme provide a better representation of OH regeneration during isoprene oxidation (Browne et al., 2014) compared to the previous implementation of the MOGUNTIA mechanism.

The MOGUNTIA chemistry scheme is in line with the VOCs oxidation pathways as proposed by the Master Chemistry Mechanism (MCM v3.3.1) (e.g., Bloss et al., 2005; Saunders et al., 2003). The thermal and pressure-dependent reaction rate



coefficients of the MOGUNTIA chemical mechanism are taken (where available) from the IUPAC kinetic data evaluation (Atkinson et al., 2004; Wallington et al., 2018) and supplemented with reaction rates based on recommendations given in by JPL (Burkholder et al., 2015). Photolysis rates needed to drive MOGUNTIA are taken from the IUPAC database (Atkinson, 1997; Atkinson et al., 2004) along with the updates from MCM v3.3.1 (Bloss et al., 2005; Jenkin et al., 1997, 2003, 2015;

Saunders et al., 2003). The comprehensive lists of all photochemical and thermal kinetic reactions included in the current MOGUNTIA chemical scheme are presented in Tables 1 and 2, respectively.

### 2.3 The chemical solver

The Kinetic PreProcessor (KPP) version 2.2.3 (Damian et al., 2002; Sandu and Sander, 2006) is employed to generate Fortran 90 code for the numerical integration of the gas-phase chemical mechanisms. Upon the translation of the chemistry mechanism

(e.g., species, reactions, rate coefficients) from the KPP language into a Fortran 90 code, a model driver was developed that arranges the coupling to TM5-MP. Minor changes, however, were needed in the KPP code to deal with TM5-MP I/O requirements. The photolysis and the thermal reactions are not calculated in KPP, but explicitly calculated by the respective modules of TM5-MP and then directly provided to the aforementioned chemistry driver. To this end, only the integration method has been updated in the model, replacing the default hand coded chemical solver set-up. Moreover, the NO emission

rates (as well as the dry deposition terms of all deposited species) are imported to KPP through the application of appropriate production (and loss) rates, as previously done for the EBI solver, owing mainly to the numerical stiffness of the $NO$-$NO_2$-$O_3$ photo-stationary state and their fast interactions (e.g., see Huijnen et al., 2010).

In this study, the Rosenbrock solver is used as the numerical integrator (Sander et al., 2019). Rosenbrock has been shown to be robust and capable of integrating very stiff sets of equations (Sander et al., 2011). For all previous versions of the model

the Euler Backward Iterative (EBI) solver (Hertel et al., 1993) has been used. This holds for the modified CBM4 (Houweling et al., 1998), the mCB05 (Williams et al., 2013) and the MOGUNTIA (Myriokefalitakis et al., 2008, 2011) mechanisms. EBI was originally designed for the CBM4 mechanism (Gery et al., 1989) and is a rather fast and robust solver suitable for the use in large-scale atmospheric models that incorporate operator splitting (Huang and Chang, 2001). However, the implementation of KPP generated code in the model is less prone to errors that may occur when relatively detailed chemistry schemes such as

the MOGUNTIA need to be manually coded. An automatic code generator, thus, offers higher flexibility for testing, updating, and further developing the chemistry code.

The favorable comparison of the Rosenbrock solver against other widely used methods, such as Facsimile (Curtis and Sweetenham, 1987), has already been described in the literature (e.g., Sander et al., 2005). Focusing specifically on the comparison of a series of Rosenbrock solvers to EBI, Sandu et al. (1997) concluded that, although EBI appears robust,

especially when it is used with a relatively large timestep, the Rosenbrock methods with variable timesteps are significantly more accurate and clearly superior for accuracies in the range of 1% compared to EBI for a range of species examined. In this study, we do not aim however to compare the two chemistry solvers (i.e., the Rosenbrock vs. the EBI); on the contrary, we present model simulations using the Rosenbrock solver as produced by KPP for the mCB05 scheme (see Sect. 2.5) to isolate





the impact of the solver on various species mixing ratios of this work, and as an intermediate step between the earlier model version (i.e., using the EBI solver and the mCB05 chemistry scheme) and the new version (i.e., using the Rosenbrock solver and MOGUNTIA chemistry scheme).

## 2.4 Emission set-up

For the present study, emissions from anthropogenic activities including aircraft emissions (Hoesly et al., 2018) and biomass burning, including agricultural waste burning, deforestation fires, boreal forest fires, peat fires, savanna fires and temperate forest fires (van Marle et al., 2017), are adopted from the sectoral and gridded historical inventories as developed for the CMIP6 (Eyring et al., 2016). In more details, anthropogenic and biomass burning emissions of CO, $NO_X$, black carbon aerosol (BC), particulate organic carbon (OC), sulfur dioxide and sulfates ($SO_x$), as well as speciated non-methane volatile organic

compounds (NMVOCs) are considered, such as emissions of ethane ($C_2H_6$), methanol ($CH_3OH$), ethanol ($C_2H_5OH$), propane ($C_3H_8$), acetylene ($C_2H_2$), ethane ($C_2H_4$), propene ($C_3H_6$), isoprene ($C_5H_8$), monoterpenes ($C_{10}H_{16}$), benzene ($C_6H_6$), toluene ($C_7H_8$), xylene ($C_8H_{10}$) and other aromatics, higher alkenes, higher alkanes, HCHO, acetaldehyde ($CH_3CHO$), acetone ($CH_3COCH_3$), dimethylsulfide (DMS; $C_2H_6S$), formic acid (HCOOH), acetic acid ($CH_3COOH$), methyl ethyl ketone (MEK; $CH_3CH_2COCH_3$), methylglyoxal (MGLY; $CH_3COCHO$), and hydroxyacetaldehyde ($HOCH_2CHO$). Note that all biomass

burning emissions (open forest and grassland fires) are vertically distributed in the model over latitude-dependent injection heights, i.e., for tropical (30° S–30° N), temperate (30–60° S/N) and high-latitude (60–90° S/N) forest fires (see Appendix in van Noije et al., 2014).

Biogenic emissions from vegetation include isoprene, terpenes and other volatile organic compounds, and CO. They are based on the Model of Emissions of Gases and Aerosols from Nature (MEGAN) version 2.1 (Sindelarova et al., 2014). Isoprene and

20 terpenes emissions are distributed over the first ~50 m and a diurnal cycle is imposed. The biogenic emissions from soils include $NO_X$ (Yienger and Levy, 1995), $NH_3$ and terrestrial DMS emissions from soils and vegetation (Spiro et al., 1992). Oceanic emissions of CO and NMVOCs come from the POET database (Granier et al., 2005), oceanic emissions of $NH_3$ from Bouwman et al. (1997), while the DMS oceanic emissions are calculated online (van Noije et al., manuscript in preparation) using the sea water concentration climatology from Lana et al., (2011). The $NO_X$ production by lightning is parameterized

based on convective precipitation fields (Meijer et al., 2001) and the $SO_x$ fluxes from continuously emitting volcanoes are taken from Andres and Kasgnoc (1998). Note that here our focus is mostly on the NMVOCs speciation used for the MOGUNTIA chemical scheme since other emissions of tropospheric species in the gas and the particulate phase in the model are described in detail in previous studies (e.g., van Noije et al., 2014).

Overall, the MOGUNTIA chemical scheme considers direct emissions of CO, $CH_4$, HCHO, HCOOH, $CH_3OH$, $C_2H_6$, $C_2H_4$,

$C_2H_2$, $CH_3CHO$, $CH_3COOH$, $C_2H_5OH$, $HOCH_2CHO$, CHOCHO, $C_3H_8$, $C_3H_6$, $n$-$C_4H_{10}$, MEK, $C_5H_8$, $C_{10}H_{16}$, $C_7H_8$ as well as $NO_X$, $NH_3$, DMS, and $SO_x$. Butanes, pentanes, hexanes, and higher alkanes emissions are summed up into the lumped n-$C_4H_{10}$ species, which represents the alkanes containing four or more carbon atoms. For reactivity purposes, higher alkenes emissions containing four or more carbon atoms (butenes and higher alkenes) are accounted for as equivalent $C_3H_6$ emissions. Higher





ketones (i.e., except for acetone) from open biomass burning emissions are represented as MEK. Emissions of benzene ($C_6H_6$), toluene ($C_7H_8$), xylene ($C_8H_{10}$), trimethyl-benzenes, and other higher aromatics and VOCs are represented by toluene as in the MOZART mechanism (Emmons et al., 2010a). Note that for the MOGUNTIA, when emissions are assigned to lumped species, corrections are made to preserve mass conservations and atmospheric reactivity (see also notes in Tables 1 and 2).

The explicit parameterization of VOC species in the MOGUNTIA chemical scheme requires, however, emissions that are not routinely included in available emission databases. Direct biofuel and biomass burning emissions of light carbonyls have been reported in several studies (e.g., Christian et al., 2003; Fu et al., 2008; Hays et al., 2002), representing overall a significant contribution to their global budgets (e.g., Fu et al., 2008; Myriokefalitakis et al., 2008; Stavrakou et al., 2009b, 2009a; Vrekoussis et al., 2009). To take this into account, emissions from biofuel use of 1.4 Tg yr$^{-1}$, 2.4 Tg yr$^{-1}$, and 1.6 Tg yr$^{-1}$ are

considered for GLYAL, GLY, and MGLY, respectively. For the biomass burning sector, global emissions of GLYAL and GLY amount in the model to about 4.3 Tg yr$^{-1}$ and 5.2 Tg yr$^{-1}$, respectively. For this work, the aforementioned emission rates are based on the HCHO distributions used in the model, owing to the highly correlated mass emission rates of low molecular weight carbonyls, such as HCHO and GLY (e.g., Hays et al., 2002). Global emissions of roughly 1.4 Tg yr$^{-1}$ (Emmons et al., 2010) are also considered here for MEK, pointing to a contribution from anthropogenic emissions (Rodigast et al., 2016), such

as domestic burning and solvent use (e.g., Ware, 1988), while for all other carbonyls, primary anthropogenic emissions are considered negligible (e.g., Fu et al., 2008). A list of the emissions strengths considered in the model for the MOGUNTIA chemical configuration of this work is presented in Table 3. For completeness, we note that primary aerosol emissions of OC, BC, sea salt, and dust are also considered in the model with sea-salt and dust emissions calculated online. A more detailed description of the gas (and aerosol) emissions used in the model will be presented in van Noije et al. (manuscript in

preparation).

## 2.5 Simulations

For this work, we present the analysis of simulations with the mCB05 and MOGUNTIA chemical mechanisms for the year 2006, which has been the chosen year of previous benchmarking studies (Huijnen et al., 2010; Williams et al., 2013, 2017). For all simulations, a 1-year spin-up (i.e., for the year 2005) is applied. Two simulations have been performed for the mCB05

configuration: one with the EBI solver, i.e., the mCB05(EBI) and one with the Rosenbrock solver as generated by the KPP, i.e., the mCB05(KPP). This approach isolates differences that are caused solely by the applied chemistry solver. By comparing MOGUNTIA with mCB05(KPP), as both generated by the KPP, the differences due to the chemistry set-up in the model are further isolated.





## 3 Model performance

All simulations that are presented have been performed at 1°x1° horizontal resolution (e.g., Williams et al., 2017) and 34 vertical layers, driven by meteorological fields from the ECMWF ERA-Interim reanalysis (Dee et al., 2011) with an update frequency of 3 hours. Concerning the TM5-MP performance, simulations performed on the ECMWF CRAY XC40 high-performance computer facility using 360 cores, indicate that the coupling of KPP software alone, increases the time spent in chemistry by ~59% and overall slows down the code by ~18% compared to the (hand coded) EBI version for the mCB05 mechanism. As expected, the coupling of the MOGUNTIA atmospheric chemistry scheme further increases the model runtime; with 100 transported and 28 non-transported tracers for a full simulation, which are significantly more than in the mCB05 configuration (i.e., 69 transported and 21 non-transported tracers), the MOGUNTIA mechanism increases the transport by ~43% and the chemistry by ~55% in the model. Altogether, the newly coupled MOGUNTIA chemistry scheme in TM5-MP is computationally ~27% more expensive than the mCB05 configuration. Overall, the simulated years per day for the mCB05(EBI), mCB05(KPP) and MOGUNTIA configurations of the model correspond to 0.73, 0.60 and 0.44, respectively (Table S3a). Note that an additional series of simulations with 450 cores presents only marginal changes of the above results (Table S3b). Finally, our results indicate that the runtime values for the different model configurations presented here are highly hardware dependent, owing mainly to the large I/O component due to the meteorological fields reading.

## 4 Comparison of budgets and tropospheric mixing ratios

### 4.1 Ozone ($O_3$)

Table 4 presents a detailed description of the chemical budget of tropospheric ozone as calculated by the TM5-MP model, for the three chemical configurations. Following Stevenson et al. (2006), chemical production of ozone is derived from all reactions that convert NO to $NO_2$, since $NO_2$ is rapidly photo-dissociated and forms $O_3$, i.e.,

$$NO + HO_2 \rightarrow NO_2 + OH \tag{1}$$

$$NO + RO_2 \rightarrow NO_2 + RO \tag{2}$$

where, $RO_2$ represents all the major organic peroxy radicals of the corresponding chemistry mechanism used in the model; for the MOGUNTIA scheme $RO_2$ includes the $CH_3O_2$, $C_2H_5O_2$, $HYEO_2$, n-$C_3H_7O_2$, i-$C_3H_7O_2$, $ACO_2$, $HYPO_2$, n-$C_4H_9O$, $MEKO_2$, $ISOPO_2$, $IEPOXO_2$, $MVKO_2$, $MACRO_2$, $TERO_2$, and $AROO_2$ radicals, and for the mCB05 scheme $RO_2$ includes the $CH_3O_2$ radical and $XO_2$ (i.e., the operator for the NO to $NO_2$ conversion which represents all lumped alkyl-peroxy radicals in mCB05; see Williams et al., 2017 and Yarwood et al., 2005).

The chemical $O_3$ loss is derived as the sum of the $O_3$ photolysis to $O(^1D)$, i.e.,

$$O_3 + hv \rightarrow O(^1D) + O_2 \tag{3}$$

followed by reaction with $H_2O$ to form OH, i.e.,

$$O(^1D) + H_2O \rightarrow 2OH \tag{4}$$

the $O_3$ destruction by $HO_2$ and OH catalytic cycles, i.e.,





$O_3 + HO_2 \rightarrow OH + 2\,O_2$ (5)

$O_3 + OH \rightarrow HO_2 + O_2$ (6)

as well as the reactions of $O_3$ with unsaturated VOCs. Note that for the chemical loss calculations of this work, the $HNO_3$, $NO_3$ and $N_2O_5$ contributions, together with the other fast cycles between ozone-related species are ignored, as proposed by

Stevenson et al. (2006).

For the MOGUNTIA scheme, the tropospheric chemical production in the model is calculated to be 5709 Tg yr$^{-1}$, which is ~10 Tg yr$^{-1}$ smaller compared to the mCB05(KPP) configuration. Chemical destruction in the troposphere is higher for the MOGUNTIA compared to the mCB05(KPP) chemistry configuration (Table 4), which can be attributed to the respective changes in the mixing ratios of the $O_3$ precursor gases. Moreover, the use of EBI compared to the Rosenbrock solver in the

model decreases the $O_3$ chemical production (5719 vs. 5589 Tg yr$^{-1}$) and destruction (5216 vs. 5192 Tg yr$^{-1}$) terms in the troposphere (Table 4). Besides some expected differences due to the behavior of the two solvers, the calculated differences is also attributed to the applied mass fixing for $NO_Y$ (i.e., the sum of NO, $NO_2$, $NO_3$, $HNO_3$, $HNO_4$, $2\times N_2O_5$, PAN and the organic nitrate compounds) in the mCB05(EBI) configuration to ensure no artificial loss of nitrogen. This $NO_Y$ fixing occurs in the model mainly over highly polluted regions with active $NO_X$ photochemistry.

Focusing on the impact of the stratosphere on the tropospheric $O_3$ budget, the net STE flux of $O_3$ for the MOGUNTIA configuration is somewhat lower (~1%) than for mCB05(KPP). Considering that all configurations use the same stratospheric ozone relaxation parameterization, this difference can only be attributed to the chemical schemes. Note that the global STE of $O_3$ is here defined by simply considering the chemical production and loss budget terms, as proposed by Stevenson et al. (2006) and routinely used in many global models. Thus, the differences among the TM5-MP chemistry configurations (Table 3) do

not imply that the tropospheric chemistry impacts on the $O_3$ transport from the stratosphere, but rather that the budget is closed by an inferred stratospheric input term. In short, the higher tropospheric net chemistry $O_3$ production implies a lower stratosphere influx contribution for roughly the same deposition losses. The calculated net influx from the stratosphere for the MOGUNTIA configuration (~424 Tg yr$^{-1}$) remains within one standard deviation of a multi-model mean (552 ± 168 Tg yr$^{-1}$), as reported by both Stevenson et al. (2006) and Young et al. (2013), in line with estimates (~400 Tg yr$^{-1}$) based on observations

(Hsu, 2005; Olsen, 2004), although higher compared to the 306 Tg yr$^{-1}$ as calculated by an earlier version of the TM5 model driven by the same meteorological fields (van Noije et al., 2014). Overall, compared to the mCB05(EBI) simulation, the lower net stratosphere-troposphere exchange flux simulated in the MOGUNTIA configuration brings the model results closer to the current best estimates of the net STE.

The MOGUNTIA configuration also results in a reduction of roughly 2% in the tropospheric $O_3$ burden compared to both

mCB05 configurations. No significant change in the $O_3$ lifetime in the troposphere (i.e., 22.3 - 22.8 days) is found and the calculated lifetimes remain close to other model estimates of ~22 days (Stevenson et al., 2006; Young et al., 2013). Compared to previous studies, the tropospheric $O_3$ burden calculated for the MOGUNTIA chemical configuration (~375 Tg) is ~12% higher compared to the multi-model mean estimate of Stevenson et al. (2006) (336 ± 27 Tg), the 335±10 Tg burden derived from $O_3$ climatology from pre-2000 data (Wild, 2007), and ~20% higher compared to the tropospheric burden of 309 Tg





reported by van Noije et al. (2014). The calculated burden for the MOGUNTIA chemistry configuration is also ~11% higher compared to the burden derived from the ACCMIP models (337 ± 23 Tg; Young et al. 2013), roughly 17% higher than the burden reported by Schultz et al. (2018) and 8-15% higher than the Lamarque et al. (2012) estimations who used a tropopause level at 100 ppb of $O_3$ mixing ratios. Note that tropospheric burden estimates remain susceptible to the tropopause definition,

leading potentially to significant differences between modelling studies and for this, the tropopause level(s) should always be reported when comparing modelling estimates. Overall, the use of the MOGUNTIA mechanism seems to bring the model closer to other published estimates, by lowering the $O_3$ burden compared to the mCB05 scheme in TM5-MP.

Ozone surface and zonal mean mixing ratios simulated by the MOGUNTIA configuration for the year 2006 are presented in Figs. 1a,b, respectively. Figures 1c,d show small differences in surface and zonal mean mixing ratios between MOGUNTIA

and mCB05(KPP). Differences in surface simulated $O_3$ mixing ratios between the two mechanisms are evident downwind of regions with biogenic and tropical fire emissions. The mCB05(KPP) simulation shows higher mixing ratios (~2-4 ppb) over the ITCZ compared to the MOGUNTIA configuration, as well as over India and East Asia (up to ~10 ppb). This is mainly due to the different speciation of VOC emissions used in the two model configurations, since the VOC speciation of the emissions has to follow the VOC speciation used in the chemical schemes, with MOGUNTIA being significantly more explicit than the

mCB05. This behavior can also be seen in the zonal mean $O_3$ distribution in Fig. 1d, where the impact of the different speciation of VOCs, originating mainly from the tropics where the majority of emissions occur, is reaching the mid- and upper troposphere lifted by convection following the upward branch of the tropical Hadley cell. The use of different solvers alone does not result in any critical difference in the $O_3$ mixing ratios for mCB05 (Fig. 1e,f), presenting although some small negative differences of ~1 ppb for the mCB05(EBI), downwind of regions with high anthropogenic emissions (e.g., India).

**4.2 Hydroxyl radical (OH)**

The hydroxyl radical (OH) is the primary oxidant in the atmosphere under sunlit conditions, initiating the oxidation of various VOCs, and thus the production of hydroperoxy ($HO_2$) and organic peroxy ($RO_2$) radicals. However, due to the high complexity of OH recycling pathways in atmospheric VOC degradation, the different representations of VOC oxidation pathways in chemical mechanisms may lead to significant discrepancies between models. $CH_4$ is routinely used as a diagnostic for the

25 calculated OH abundance in the troposphere since its background concentration is highly sensitive the OH abundance in the tropics (where water vapor is high), where biogenic emissions are also high. Uncertainties in $CH_4$ global sources (e.g., a rapid rise in the $CH_4$ growth rates since 2007; Nisbet et al., 2019) together with uncertainties in anthropogenic emissions of the $NO_X$, CO, and NMVOC (e.g., Hoesly et al., 2018), may cause considerable divergence in model simulated $CH_4$ mixing ratios, for different simulation years. For the present study, however, the surface mixing ratios of $CH_4$ are prescribed according to the

30 CMIP6 recommendations for each simulation year (van Noije et al., manuscript in preparation). This approach is justified by the relatively long lifetime of $CH_4$, which allows fast global mixing of the emitted amounts.

For this work, the global tropospheric OH production budgets for the various chemical configurations are presented in Table 5. TM5-MP simulations with the MOGUNTIA model configuration yield a primary OH formation due to $O_3$ photolysis





(Reaction 3) in the presence of water molecules in the gas-phase (Reaction 4) of about 1878 Tg yr$^{-1}$. The radical recycling terms (Reactions 1 and 5) contribute 1987 Tg yr$^{-1}$, the $H_2O_2$ photodissociation, i.e.,

$$H_2O_2 + h\nu \rightarrow 2\ OH \tag{7}$$

produces another 303 Tg yr$^{-1}$, and all other reactions of the mechanism add another 120 Tg yr$^{-1}$ to the OH production in the

model. Overall, the total tropospheric OH production amounts to 4288 Tg yr$^{-1}$. The total OH production in the troposphere is in close agreement with the respective budget estimations by Lelieveld et al. (2016), i.e., ~4270 Tg yr$^{-1}$. Some difference is however expected due to the definition of the troposphere in Lelieveld et al. (2016), where they define the tropopause in the tropics using temperature, and in the extratropics using potential vorticity gradients. We remind the reader that for the present study the chemical troposphere is defined using a threshold of 150 ppb $O_3$. For clarity, we also note that no differences are

calculated in the tropopause levels between the model configurations of this work. It is notable, however, that the OH chemical production calculated here for the MOGUNTIA model set-up is much higher (28 - 35%) than for previous TM5 model configurations (i.e., 3355±30 and 3184±20 Tg yr$^{-1}$) as presented by van Noije et al. (2014). This difference is mainly attributed to the various updates of the model, such as the emission database and the applied VOC speciation (i.e., CMIP5; Lamarque et al. (2010) vs. CMIP6 for this study), the chemistry scheme (i.e., mCBM-IV vs. the more detailed MOGUNTIA), as well as the

photolysis scheme (i.e., the previous implemented Landgraf et al. (1998) photolysis scheme vs. the Modified Band Approach scheme as coupled in the model by Williams et al. (2012)).

Focusing on the differences between the MOGUNTIA and the mCB05 mechanism in the TM5-MP set up using the Rosenbrock solver, the OH production calculated by the MOGUNTIA chemistry scheme is very close to CB05(KPP) on a global scale (Table 5). Note also that for mCB05, the comparison of the two solvers alone indicates that the EBI calculates a ~1% lower

chemical destruction of OH in the troposphere than Rosenbrock. On the other hand, the contribution of the CO and $CH_4$ oxidation terms to the global tropospheric OH losses are calculated 41% and 15%, respectively, for the MOGUNTIA scheme, which is slightly higher (by ~6% and ~3%, respectively) than using mCB05(KPP).

Focusing further on the MOGUNTIA scheme, the calculated tropospheric $CH_4$ chemical lifetime is ~8.0 yr, as obtained through dividing the $CH_4$ global atmospheric mean burden (~4871 Tg) by the loss due to oxidation by OH radicals in the troposphere

(~607 Tg yr$^{-1}$). Accounting, however, for additional $CH_4$ sinks due to oxidation in soils and the stratosphere with assumed lifetimes of 160 yr and 120 yr (Ehhalt et al., 2001), respectively, we arrive at an atmospheric lifetime of about 7.18 yr, which is somewhat shorter than the ensemble model mean atmospheric lifetime reported by Stevenson et al. (2006) of 8.45±0.38 yr. The multi-model chemistry-climate simulations performed during the Atmospheric Chemistry and Climate Model Intercomparison Project (ACCMIP) (Naik et al., 2013; Voulgarakis et al., 2013), revealed vast diversities among models with

a wide range of $CH_4$ chemical lifetime values (i.e., ~7-14 yr) and a mean value of 9.7±1.5 yr (i.e., 5–10% higher than observation-derived estimates). Lelieveld et al. (2016) demonstrated a $CH_4$ chemical lifetime to 8.5 yr for the year 2010, Schultz et al. (2018) estimated a tropospheric $CH_4$ chemical lifetime of about 9.9 yr using also an $O_3$ threshold of 150 ppb to define the tropopause, and Lamarque et al. (2012) reported a chemical lifetime of ~8.7 yr but taking into account a tropopause level at 100 ppb $O_3$.



### 4.3 Carbon monoxide (CO)

Table 6 presents the chemical CO budget calculated by TM5-MP for the three chemical configurations. The different model configurations show that approximately 62±1% of the CO global production in the troposphere is due to the oxidation of $CH_4$ and NMVOC, with the remaining owing to direct emissions. Overall, the global CO budget is significantly affected by the

interactions between OH and CO. Thus, changes in OH tropospheric chemical production (i.e., ~ -0.2% from mCB05(KPP) to MOGUNTIA) modulate the tropospheric secondary formation of CO from the oxidation of $CH_4$ and NMVOC (~ -10% change) and the CO chemical loss (~ -3% change) in the model. The total chemical production of CO for the MOGUNTIA and mCB05(KPP) chemical configurations, 2018 and 1844 Tg yr$^{-1}$ respectively, are however higher than the multi-model mean estimate (1505 ± 236 Tg yr$^{-1}$) reported by Shindell et al. (2006), which can be partially attributed to the different year of

NMVOC emissions used (i.e., 2000 vs. 2006 for this work).

The dominant chemical reactions responsible for the reduction in the tropospheric CO chemical production for the MOGUNTIA chemistry are the HCHO photolysis and OH oxidation (i.e., ~2% increase compared to mCB05(KPP)). Indeed, the lumped nature of the mCB05(KPP) mechanism leads to a higher tropospheric HCHO chemical production (~1896 Tg yr$^{-1}$) compared to MOGUNTIA configuration (~1843 Tg yr$^{-1}$). HCHO is mainly formed via the oxidation of $CH_4$, isoprene, and

other NMVOC in the model. However, for both mCB05 configurations, the HCHO production via $CH_3O_2H$ photolysis is calculated to be ~1.65 times higher compared to MOGUNTIA. The latter scheme seems to recycle the methyl-peroxy radical ($CH_3O_2$) more efficiently via the $CH_3O_2$ gas-phase reactions with organic peroxy radicals ($RO_2$) produced by higher-order NMVOC oxidation. In contrast, other higher aldehydes which represent the second most important producer of CO contribute more significantly in MOGUNTIA than in mCB05. This could be due to the more detailed representation of the higher

aldehydes in the MOGUNTIA mechanism (e.g., considering the production and destruction reaction of GLY, GLYAL, and $C_2H_5CHO$) compared to the single lumped species (i.e., the ALD2) that represents all higher aldehydes in mCB05.

The global annual mean burden of CO for the MOGUNTIA chemical scheme is 361 Tg, almost the same as in the mCB05(KPP) configuration, but ~2 % lower compared to mCB05(EBI). Higher CO losses by OH oxidation and deposition in MOGUNTIA lead to a CO atmospheric lifetime of ~44 days, i.e., about 6% shorter compared to the mCB05(KPP) chemical

mechanism. Note, here, that the reduction in the atmospheric lifetime of CO is in line with the respective reduction in the atmospheric lifetime of $CH_4$ (~3%), reflecting overall an increase in tropospheric OH mixing ratios for the MOGUNTIA configuration compared to mCB05(KPP). Higher OH levels in the atmosphere lead to a proportionally larger CO and $CH_4$ sinks.

Focusing further on the impact of the solver alone, we calculate roughly a 3% reduction in the CO atmospheric burden when

the EBI solver is applied on the mCB05 mechanism in the model. This is directly connected to the respective ~1% increase in OH mixing ratios that is calculated when the Rosenbrock solver is used in the model. Furthermore, the CO tropospheric production is increased by ~0.5% in mCB05(KPP) compared to mCB05(EBI). Overall, the presented differences between the





EBI and Rosenbrock solvers confirm that the choice of solver may impact the simulated mixing ratios, owing mainly to the use of a constant versus a variable timestep in the chemistry integration (e.g., see Sandu et al., 1997).

Zonal mean CO mixing ratios at the surface for the year 2006 using the MOGUNTIA scheme are presented in Fig. 2 (a,b). Compared to mCB05(KPP), the results from MOGUNTIA show slightly higher surface CO mixing ratios (up to ~2 ppb) over

5   highly populated regions, such as India. This regional increase is due to the differences in surface OH mixing ratios, owing mainly to the respective differences $NO_X$ chemistry between the two simulations (see also Sect. 5.2). On the contrary, at South America negative differences of ~5-15 ppb are calculated at the surface (Fig. 2c). The effective $HO_x$ regeneration together with the rather detailed VOC speciation and oxidation pathways considered in the MOGUNTIA scheme result in an increase of the surface OH mixing ratios where high biogenic VOC emissions occur, leading thus to a regional decrease in the

10  tropospheric CO mixing ratios compared to the mCB05(KPP) configuration. Similar results are found for the zonal mean CO distribution. Free tropospheric CO mixing ratios in the tropics are also affected due to effective tropical convection. Finally, the use of different solvers for the mCB05 mechanism does not lead to any notable differences in the annual mean CO mixing ratios (Fig. 2e,f).





## 5 Model evaluation

Model simulations are here evaluated with a series of surface, flask, aircraft, and sonde measurements, as well as with satellite retrievals and climatological data. The simulated $NO_2$ tropospheric columns are compared with satellite retrievals from the European project Quality Assurance for Essential Climate Variables (QA4ECV) project (Boersma et al., 2017), provided by

5 the Ozone Monitoring Instrument (OMI) and the SCanning Imaging Absorption SpectroMeter for Atmospheric CHartographY (SCIAMACHY) instruments. The simulated OH mixing ratios are evaluated against calculations of global mean tropospheric values from other modelling studies, as well as against climatological data compiled by Spivakovsky et al. (2000). Modeled $O_3$ mixing ratios are evaluated against surface observations and ozonesonde data for the year 2006, as compiled by the World Ozone and Ultraviolet Radiation Data Centre (WOUDC; http://www.woudc.org; last access 20/08/2019); surface observations

from the European Monitoring Evaluation Program network (EMEP; http://www.emep.int; last access 20/08/2019) have been also used. For the CO model evaluation, flask observations for the year 2006 are used, as compiled by National Oceanic and Atmospheric Administration Earth System Research Laboratory, Global Monitoring Division (NOAA, https://www.esrl.noaa.gov/gmd; last access 20/08/2019). $O_3$ and CO mixing ratios in the upper troposphere/lower stratosphere (UTLS) are also compared to *in-situ* measurements from the MOZAIC (Measurement of Ozone and Water Vapour by Airbus

In-Service Aircraft) data record (Thouret et al., 1998). The modelled CO total columns are compared with satellite retrievals from Measurement of Pollution in the Troposphere (MOPITT) instrument, version MOP02J_V008 (Deeter et al., 2013, 2019; Ziskin, 2019), i.e., the combined thermal/near-infrared data product. Finally, light VOCs (i.e., $C_2H_4$, $C_2H_6$, $C_3H_6$, $C_3H_8$) as simulated by the model for the year 2006 are evaluated against flask measurements from the NOAA database, as well as against climatological data from aircraft campaigns, as produced by Emmons et al. (2000). Overall, to quantify and discuss the model's

performance, commonly used statistical parameters are calculated, such as the correlation coefficient (R) which reflects the strength of the linear relationship between model results and observations (the ability of the model to simulate the observed variability), the absolute bias (BIAS) and the normalized mean bias (NMB), as well as the root mean square error (RMSE) as a measure of the mean deviation of the model from the measurement due to random and systematic errors. All equations used for the statistical analysis of model results are provided in the supplementary material (Eq. S1–S5).

### 5.1 Nitrogen dioxide ($NO_2$)

$NO_X$ is a rate-limiting precursor of $O_3$ formation and thus an essential species for other tropospheric oxidants, such as OH. $NO_X$ is emitted by both natural (lightning, soils, and fires) and anthropogenic combustion sources, with lightning mainly impacting $NO_X$ mixing ratios at the top of convective up-drafts and anthropogenic fuel emissions being the principal source of NO at the surface. Tropospheric $NO_2$ vertical column densities retrieved from OMI (Boersma et al., 2017) are compared

against the MOGUNTIA and mCB05(KPP) simulations (Fig. 3). Note that since the differences between mCB05(EBI) and mCB05(KPP) are small for tropospheric $NO_2$ columns, mCB05(EBI) is not shown. $NO_2$ column densities are retrieved using a consistent set of retrieval parameters and validated against ground-based MAX-DOAS measurements (Boersma et al., 2018).





To consider the vertical sensitivity of the satellite measurements to $NO_2$ molecules at different altitudes, the tropospheric column averaging kernels, provided in the QA4ECV data product, are applied separately to both sets of modelled $NO_2$ vertical profiles, extracted from the hourly 3-D model output by linear and nearest-neighbor interpolation in space and time. The resulting $NO_2$ tropospheric column density is what would have been retrieved by the satellite if the actual vertical profile of

$NO_2$ mixing ratios were identical to the modeled profile. The tropospheric $NO_2$ columns retrieved from the satellite are averaged per model grid cell and day, resulting in a comparison dataset consisting of one $NO_2$ vertical column density per model grid cell and day.

For the MOGUNTIA configuration, the model shows a mean overestimation of $1.78 \times 10^{14}$ (R=0.71) and $1.96 \times 10^{14}$ molecules $cm^{-2}$ (R=0.95) against OMI measurements for daily and annual values, respectively, slightly better than the correlation of

mCB05(KPP) configuration (R=0.71 and R=0.94 for daily and annual values). An overview of the statistical comparison of the three model simulations against OMI measurements is given in Fig. S1a. Some discrepancies are nevertheless expected in such a comparison. since not a significant annual cycle is usually applied to anthropogenic emissions which are the principal source of $NO_X$, especially in the Northern Hemisphere (NH). Over the biomass burning source regions in Africa, the model overestimates the satellite retrievals. Note also that when the model is compared against $NO_2$ tropospheric columns from the

SCIAMACHY instrument using the QA4ECV retrieval (not shown), the MOGUNTIA configuration shows again a similar improvement over mCB05(KPP), as with the OMI data.

Williams et al. (2017) showed a significant underestimation in the NO and $NO_2$ mixing ratios as calculated by the TM5-MP model both at the surface and in vertical profiles, with the model satisfactorily reproducing the $NO_2$ mixing ratios in the boundary layer but overestimating mixing ratios at higher altitudes and in pristine environments. The MOGUNTIA scheme

shows generally a better agreement with satellite retrievals compared to the mCB05(KPP) configuration, as expressed by a higher correlation coefficient and a generally lower bias (Fig. S1a). The differences between the two chemistry schemes can be mainly attributed to the speciation of organic $NO_X$ reservoir species in the two mechanisms. Overall, since deep convection efficiently transports organic $NO_X$ reservoir species to the upper troposphere, the more explicit representation of VOC chemistry in the MOGUNTIA chemistry scheme efficiently alters the distribution of $NO_X$ reservoir species compared to the

more lumped chemistry of the mCB05.

Several modelling studies have compared the simulated $NO_2$ columns with satellite (e.g., Travis et al., 2016; Williams et al., 2017), resulting in an overestimate of the observed $NO/NO_2$ ratios possibly due to a respective underestimate of peroxy radicals that contribute to the NO to $NO_2$ conversion. A deviation in the $NO/NO_2$ ratio has also been reported for the GEOS-Chem model (Silvern et al., 2018; Travis et al., 2016), with the model to significantly underestimate the observed upper tropospheric

$NO_2$ observations from the SEAC[4]RS aircraft campaign over the southeast United States. Silvern et al. (2018) calculated that the reaction with ozone accounts for roughly 75% of the NO to $NO_2$ conversion in the upper troposphere; thus, this deviation from the photochemical equilibrium could be due to an error in the kinetic data used in the model. Overall, the authors indicated that reducing the $NO_2$ photolysis by 20% and increasing the low-temperature $NO + O_3$ reaction rate constant by 40%, improves the model simulation of the $NO/NO_2$ ratio in the upper tropospheric data significantly compared to the aircraft data. Another





source of uncertainty, could also be the strength of the direct soil emissions that, according to Miyazaki et al. (2017), are lower in our model (i.e., ~5 Tg-N yr$^{-1}$; Yienger and Levy, 1995) compared to the emissions of 7.9 Tg-N yr$^{-1}$ derived using a multi-constituent satellite data assimilation.

## 5.2 Hydroxyl radical (OH)

Figures 4a and 4b illustrate the zonal mean tropospheric distributions of OH and for two seasons (i.e., boreal winter and boreal summer) for 2006, as simulated by the model for the MOGUNTIA chemistry scheme. The highest atmospheric OH mixing ratios in the model are calculated in the tropics from close to surface up to roughly the tropopause, as a result of intense solar radiation and high humidity in the region, with the main OH maximum being roughly below 400 hPa (and a secondary one at ~300 hPa). The differences in OH zonal mean mixing ratios compared to the mCB05(KPP) configuration for the two seasons

are also presented in Figs. 4c,d. During the boreal winter, the mCB05(KPP) configuration results on average in lower OH mixing ratios in the northern subtropical lower troposphere (~3-6%) than the MOGUNTIA simulation (Fig. 4c), with the higher decreases (~20-30%) to be calculated at around 20º-40º N. In the subtropical Southern Hemisphere (SH) during boreal summer, OH mixing ratios are on average lower (~2-3%) in the MOGUNTIA configuration than in mCB05(KPP) (Fig. 4d) almost everywhere, except a small increase (up to 10%) at around 30º S. These relatively small differences in OH mixing ratios are

mainly related to the HOx regeneration, as well as the NOx and the organic NOx reservoir species that influence the production and thus the distribution of OH in the troposphere. The more detailed representation of organic NOx reservoir species in the MOGUNTIA chemistry scheme results in more efficient transport of these species to higher altitudes by deep convection increasing the OH mixing ratios at higher altitudes. Globally, the NO + HO$_2$ reaction is roughly 9% higher in the MOGUNTIA configuration on an annual basis compared to mCB05(KPP) (see Table 5).

Focusing on global mean mixing ratios, a mean tropospheric OH of 10.1×10$^5$ molecules cm$^{-3}$ is obtained from the MOGUNTIA chemistry configuration for the year 2006, which is roughly 4% higher than in the mCB05(KPP) configuration, but closer to the low end of the multi-model mean of 11 ± 1.6×10$^5$ molecules cm$^{-3}$ as derived by Naik et al. (2013) for the year 2000, and the mean tropospheric mixing ratios of 11.3×10$^5$ molecules cm$^{-3}$ as calculated by Lelieveld et al. (2016) for the year 2013. In the tropical troposphere (30ºS - 30ºN), the mean OH level in the MOGUNTIA configuration of 16.74×10$^5$ molecules

25    cm$^{-3}$ is ~6% higher than in mCB05(KPP). In all model configurations, higher OH mixing ratios are calculated in the NH compared to the SH, directly related to the asymmetry in the hemispheric O$_3$ and NOx burdens. Figures 4e,f show the climatological mean OH mixing ratios from the surface up to ~200hPa from Spivakovsky et al. (2000) but reduced here by 8% based on the observed decay of methyl-chloroform mixing ratios (see Huijnen et al., 2010; van Noije et al., 2014). The mean tropospheric OH concentration for the MOGUNTIA configuration is calculated to be roughly 25% and 30% higher

compared to the optimized climatology from Spivakovsky et al. (2000) for boreal winter and summer, respectively. Moreover, a ~28% higher NH/SH ratio of annual mean hemispheric OH mixing ratios in the troposphere is derived for the MOGUNTIA configuration compared to Spivakovsky et al. (2000). Note that in this work, the NH/SH ratios are calculated ~1.37 and ~1.35 for the MOGUNTIA and the mCB05(KPP) configuration, respectively, being on the high end of other modeling estimates,





such as the multi-model estimate of a NH/SH ratio of 1.28 ± 0.10 by Naik et al. (2013) and the roughly 12-15% lower ratio reported by Lelieveld et al. (2016) for the year 2013.

## 5.3 Ozone (O₃)

The evaluation of modeled $O_3$ mixing ratios against surface observations for the three simulations performed for this work for the year 2006 is presented in Fig. 5. The seasonal cycle across surface stations is generally well captured by all model configurations for most of the cases. TM5-MP, however, generally overestimates $O_3$ mixing ratios at most NH sites and for all model configurations, as for example can be seen at the Barrow (Fig. 5a) and Mace Head (Fig. 5b) stations, especially during the summer (June-July-August, JJA) season, when $O_3$ is overestimated by about 8 and 3 ppb, respectively. However, at Viznar (Spain) and Mauna Loa (USA) (Figs. 5c and 5d, respectively), model results are closer to the observed $O_3$ mixing ratios, showing overall lower biases (i.e., ~1-3 ppb). In the SH (except for the polar circle), the model simulates the seasonal cycle of the $O_3$ surface mixing ratios well, however, with average positive biases of ~6-10 ppb in Cape Point (South Africa) and Baring Head (New Zealand) (Figs. 5e,f). At the South Pole (USA) and Sayowa (Japan) stations in Antarctica (Figs. 5g,h), the model also captures the observed seasonality well (R= ~0.9), except for a negative bias of ~3 ppb during the local winter season. Focusing further on the chemistry mechanisms applied in the model, a slightly better consistency is achieved for the MOGUNTIA chemistry scheme in most of the cases. For the mCB05 chemistry scheme, the choice of the solver does not result in any notable difference in simulated surface $O_3$ mixing ratios. Considering all surface $O_3$ observations available in this work for the year 2006 (Fig. S2), the MOGUNTIA chemistry configuration tends to overestimate the available observations by a mean bias of ~6.5 ppb. Note that although the differences between the different chemistry configurations for surface $O_3$ are small, the mCB05(KPP) configuration shows the lowest bias (~5.2 ppb) whereas the mCB05(EBI) bias is closer to that of the MOGUNTIA configuration (~6.1 ppb).

Ozonesonde observations are used to evaluate the models' ability to reproduce the $O_3$ vertical profiles. Indicatively, Fig. 6 presents the comparison of model results with ozonesonde observations in 2006 at the Hohenpeissenberg in Germany and at the Macquarie Island in the Southwestern Pacific Ocean, at five pressure levels (900 hPa, 800 hPa, 500 hPa, 400 hPa, and 200 hPa) covering the boundary layer and the low and high free troposphere. For this evaluation, all ozonesonde data have been binned to the 34 model pressure levels (see Sect. 1). The seasonal cycle at the two stations is well captured by each model configuration. For the highest model levels, above 200 hPa, all simulations are very close to the measurements, since $O_3$ mixing ratios are mainly determined by the upper boundary condition that is used (see Sect. 2). Comparisons for other stations around the globe for the year 2006 available by WOUDC are presented in the supplementary material (Fig. S3). Overall, all model simulations capture the $O_3$ distribution quite well at almost all sites in the lower troposphere. The MOGUNTIA scheme shows a slightly better agreement with observations than the mCB05 configurations with smaller biases in most of the cases, especially at lower levels (i.e. from ~900hPa and up to ~500hPa). Some differences in the model performance at the various stations are nevertheless expected due to their different characteristics such as precursor source regions and transport patterns. Concerning the impact of the chemistry solver, the vertical $O_3$ concentration simulated using the mCB05 mechanism shows



no notable differences between the use of KPP and EBI in most of the cases. Overall, considering all available ozonesonde data for the year 2006 (Fig. S4), the MOGUNTIA chemistry in TM5-MP results in an overestimation of the ozonesondes observations by roughly 16% (R = 0.96, BIAS = 4.7 ppb, NME=15.6%), which is slightly smaller compared to the mCB05 chemistry configurations.

Figure S4 further presents a comparison of $O_3$ mixing ratios in the upper troposphere/lower stratosphere (UTLS) simulated by TM5-MP for the two chemistry configurations (i.e., mCB05(KPP) and MOGUNTIA). For this $O_3$ evaluation, *in-situ* measurement data from MOZAIC are used as a function of latitude. The accuracy of the MOZAIC $O_3$ measurements is ±2 ppb (Marenco et al., 1998). For this comparison, the MOZAIC measurements are binned on the vertical grid of TM5-MP. The model evaluation at pressure levels < 300 hPa indicates a positive bias in April of the order of ∼20 ppb, but smaller biases are

clearly seen at latitudes north of 40ºN (Fig. S4a). The negative model bias in the tropical UTLS points at a weak convective uplift in tropical Africa in April. Generally, there is a good agreement with the observations for both simulations, except for a constant bias of ∼20 ppb in October (Fig. 4Sb), although the variability is somewhat larger in the observations compared to the simulations. This could be caused by the limited vertical resolution of this model version in the UTLS region; note that 34 vertical levels were employed for this study with a higher resolution in the upper troposphere–lower stratosphere region. Part

of the model overestimation can also be attributed to a systematic error, also presented in previous studies, caused possibly by cumulative effects, such as a lack of a diurnal or weekly variation in the $NO_X$ emissions from the road transport sector, missing surface deposition during summer, errors in the representation of nocturnal boundary layer dynamics, or a weak convective mixing out of the boundary layer (e.g., see Williams et al., 2012).

**5.4 Carbon monoxide (CO)**

Figure 7 presents the model performance concerning surface CO mixing ratios, by comparing a series of flask observations for the year 2006. CO is underestimated at most sites in the NH for all TM5-MP configurations, e.g., at the Barrow Observatory and Mace Head station (Figs. 7a,b), especially during boreal spring (March-April-May, MAM), by about 30 ppb on average. In the tropics, negative biases (∼16-20 ppb) are observed at Mauna Loa and Mahe Island (Figs. 7c,d). At other stations in the SH, the model simulates the CO surface mixing ratios well with both positive and negative biases depending on the season

(Figs. 7e,f). In Antarctica, at the South Pole and Sayowa stations (Figs. 7g,h), the model also shows a small positive bias up to ∼3 ppb during the local winter season. The seasonal cycle across stations is generally well captured by all three model's chemistry configurations (i.e., R = 0.7- 0.9). The full set of CO comparisons with flask data is further presented in the supplement (Fig. S5). Overall, the MOGUNTIA and the mCB05(KPP) configurations underestimate the flask observations for the year 2006 with a negative bias of around 30 ppb, and with a correlation coefficient for both configurations of R=0.45.

Notably, the mCB05(EBI) model configuration tends to produce lower biases (approximately -3 vs. -4 and -5 ppb for mCB05(KPP) and MOGUNTIA, respectively) compared to the other two configurations that use the Rosenbrock solver in the SH, where the emission strengths are low. In contrast, the MOGUNTIA chemistry configuration results in lower biases in the



NH (approximately -30 vs. -31 and -33 ppb for mCB05(EBI) and mCB05(KPP), respectively), where the majority of emissions occur.

Total CO columns from the MOGUNTIA and the mCB05(KPP) model configurations are compared to the total column densities retrieved from the MOPITT satellite instrument (Deeter et al., 2013, 2019; Ziskin, 2019) for the year 2006 (Fig. 8).

Co-sampling with averaging kernel has been applied to the modelled CO concentration profiles (i.e., in the same manner as for NO$_2$; see Sect. 5.1). Note that when the absolute difference in surface pressure between the MOPITT retrieval and the TM5-MP simulation is larger than 5 hPa, the measurements were excluded from the comparison. For the MOGUNTIA configuration, the model shows a mean underestimation of -8.54×10$^{16}$ (R=0.82) and -1.18×10$^{17}$ molecules cm$^{-2}$ (R=0.91) compared to daily and annual averages of MOPITT data, respectively. However, the correlation is slightly improved compared

to the mCB05(KPP) configuration (R=0.78 and R=0.88 for daily and annual values, respectively). As in the comparison with surface data, the biases in total column CO in the MOGUNTIA and mCB05(KPP) configurations deteriorated compared to the mCB05(EBI) configuration, albeit biases are still small (~-5% and ~-7% for daily and annual values, respectively). As this pattern can be seen in both KPP configurations, this difference seems to be caused by the implementation of the more accurate Rosenbrock solver. An overview of the statistical comparison of the three model configurations against MOPITT CO

measurements is given in Fig. S1b.

Figure S4 further presents the comparison of CO mixing ratios in the upper troposphere/lower stratosphere (UTLS) simulated by TM5-MP with the *in-situ* measurements from the MOZAIC data (see Sect. 3.1). Model evaluation at pressure levels < 300 hPa shows a positive bias for mCB05(KPP) configuration in April of the order of ~20 ppb and a small negative bias (~10 ppb) for the MOGUNTIA for latitudes below 10ºN, but a negative bias for both simulations for latitudes above 20ºN (Fig. S4c).

For October, both configurations show a good correlation for the SH, with a negative bias (~20 ppb) for latitudes above 20ºN (Fig. S4d). However, a strong positive bias of roughly 30 ppb is simulated in the tropics, for latitudes up to 20ºN. The positive model bias in the UTLS in the tropics could point to an excessively strong convective uplift in tropical Africa in April, and/or during the summer monsoon over Southern Asia in October. The model shows both positive and negative biases compared to the MOZAIC observations, with the observations exhibiting larger latitudinal CO variability. This might also point to a

possible misrepresentation of biomass burning emissions that are generally uncertain. Overall, and also considering the respective O$_3$ evaluation, this suggests that the MOGUNTIA chemistry scheme tends to slightly better simulate the chemical composition of the troposphere at remote locations away from direct emission sources and improves upon the mCB05 scheme.

## 5.5 Volatile organic compounds (VOCs)

### 5.5.1 Ethane and propane

Ethane (C$_2$H$_6$) is the lightest alkane with emissions primarily of anthropogenic origin, associated mainly with fossil fuel extraction and use. In the model, the global ethane emission is 11 Tg yr$^{-1}$ (Table 3) with an atmospheric lifetime of about 56 days for all chemistry configurations, in close agreement with other studies (e.g., Hodnebrog et al., 2018). Flask measurements





indicate that $C_2H_6$ surface mixing ratios are strongly underestimated by all configuration at Mace Head (Fig. 9a) by ~80%, mainly during the winter, indicating also an opposite annual cycle. Significant underestimations are also observed in the tropics at Mauna Loa, Hawaii (Fig. 9c), of roughly 98% (R ≈ -0.5). In contrast, at Cape Grim, Australia (Fig. 9e), the model is better capable of reproducing the measured $C_2H_6$ mixing ratios for all configurations, with a higher correlation coefficient (R = 0.5)

and an NME of around 63%. This likely indicates that the model lacks primary emissions of $C_2H_6$ and thus, it can better reproduce atmospheric observations in the SH where the anthropogenic emissions are not as strong as in the NH. The full set of $C_2H_6$ comparisons with flask data is presented in the supplement (Fig. S6).

Propane ($C_3H_8$) is also emitted mainly from anthropogenic sources, and in the current simulations the total emission is 8.5 Tg $yr^{-1}$ (Table 3), appearing here however lower compared to other reported emission estimates of ~15 Tg $yr^{-1}$ (Jacob et al., 2002).

Model comparison with flask observations (Fig. 9) shows that the model tends to underestimate the measured mixing ratios for all simulations, however, with higher correlation coefficients compared to $C_2H_6$ in most of the cases. $C_3H_8$ is underestimated in the NH at Mace Head (Fig. 9b) during the winter and autumn seasons by 72-74%. In the tropics, strong negative biases of ~100 ppt are also observed at Mauna Loa (Fig. 9d). However, the model simulates the $C_3H_8$ surface mixing ratios better in the SH at Cape Grim compared to stations in the NH (Figs. 9b,d,f) due to the weaker impact of anthropogenic emissions, as for

$C_2H_6$. In contrast to the $C_2H_6$ evaluation however, the model satisfactory simulates the observed $C_3H_8$ mixing ratios at the South Pole (Fig. 9h), with a small overestimation however of the measured mixing ratios during the local summer season. The full set of $C_2H_6$ comparisons with flask data is also presented in Fig. S7.

Comparison with $C_2H_6$ and $C_3H_8$ aircraft climatological data (Fig. 10) further indicates that all chemistry configurations tend to underestimate the observed mixing ratios (~20-60%) in most of the cases, especially in the upper troposphere. In more

detail, at Boulder and East Brazil, the model significantly underestimates the observed mixing ratios for both compounds, while at Hawaii the $C_2H_6$ is underestimated but the $C_3H_8$ is well simulated by all three configurations. On the contrary, at Easter Island, all schemes overestimate the observed mixing ratios for both compounds, although the MOGUNTIA overestimates higher the $C_2H_6$ but lower the $C_3H_8$ compared to the two mCB05 configurations. The full sets of $C_2H_6$ and $C_3H_8$ comparisons with aircraft climatological data are further presented in the supplement (Fig. S8 and Fig. S9, respectively). Overall,

considering that the model reasonably simulates the oxidative capacity of the atmosphere, direct emissions should be the reason for these differences, since both alkanes are oxidized in the troposphere by OH radicals and not any secondary production terms of these alkanes are known. Note, however, that alkanes emission fluxes in the model should be considered on the low side, as also suggested by other studies in the literature (e.g., Aydin et al., 2011; Huijnen et al., 2019; Monks et al., 2018).

### 5.5.2 Ethene and propene

Ethene is mainly emitted from biogenic sources as well as by the incomplete combustion from biomass burning, power plants, and combustion engines. For this work, $C_2H_4$ emissions account for roughly 30 Tg $yr^{-1}$ (Table 3), close to the estimate of Huijnen et al. (2019), however on the high side compared to the 21 Tg $yr^{-1}$ reported by Toon et al. (2018). The three chemistry configurations produce similar mixing ratios of $C_2H_4$ in TM5-MP. Nevertheless, the comparison with aircraft observations





(Fig. 11) indicates significantly underestimated mixing ratios for all model configurations, especially in the upper troposphere (~10 km) at almost all locations. In more detail, the model reproduces well the vertical distribution of $C_2H_4$ at Boulder (USA) (R=0.97), however, it overestimates the observed mixing ratios close to the surface (up to ~ 2 km) and underestimates at the higher levels (up to ~ 6 km) for all configurations, with an average positive bias of around 5 ppt. In the tropics, the model underestimates the observed mixing ratios in the lower and upper troposphere (e.g., at Hawaii) by roughly 25 ppt. In remote regions, where the impact of direct emissions is negligible (e.g., at the Easter Island), the model overestimates $C_2H_4$ close to the surface (~1 km), but negative biases appear aloft. Overall, these underestimations can be also attributed to the underestimated background concentrations, because of severe uncertainties in emission fluxes or even due to not well-understood chemistry (e.g., Huijnen et al., 2019; Pozzer et al., 2007), such as the $C_2H_4$ production during the VOC decomposition in the atmosphere.

For propene ($C_3H_6$), the two mCB05 configurations in TM5-MP produce similar mixing ratios but the MOGUNTIA tends to simulate higher mixing ratios, especially in the tropics at Hawaii (Fig. 11d) and at East Brazil (Fig. 11f). However, close to the surface, where the impact of the emissions is stronger, the model severely overestimates observations (Figs. d,f), except for Japan (Fig 11b). For the MOGUNTIA configuration, this overestimation is more substantial in the tropics compared to the mCB05 chemistry scheme. This can be attributed to the more involved chemistry of MOGUNTIA. An overestimation of the observed mixing ratio close to the surface is also shown in other regions, especially in the SH, such as in Eastern Brazil (Fig. 11f) or in remote regions, where the direct impact of emissions is negligible, such as in the Easter Island (Fig. 11h). Overall, even though the evaluation of vertical profiles should be considered only as a climatological comparison, the reason for the calculated model underestimation is a combination of the emission strengths, the simulated vertical distribution, and the potential but still unaccounted secondary production from VOC oxidation. All comparisons for $C_2H_4$ and $C_3H_6$ with aircraft climatological data are presented in Fig. S10 and Fig. S11, respectively.



## 6. Summary and conclusions

This study documents and evaluates the implementation of the detailed tropospheric chemistry scheme MOGUNTIA in the global chemistry and transport model TM5-MP. The MOGUNTIA scheme is a comprehensive gas-phase chemistry mechanism that explicitly accounts for the oxidation of light hydrocarbons, coupled with an updated representation of isoprene

oxidation, along with a simplified representation of terpenes and aromatics chemistry. The newly coupled chemistry scheme in TM5-MP is compared to the existing chemistry scheme of the model, the mCB05 scheme. Another feature implemented in the TM5-MP chemistry code is the Rosenbrock solver, replacing the classical EBI method. For this, a simple preprocessor directive has been implemented in the model to choose between the two solvers during model compilation. In the case of the Rosenbrock solver, the KPP software has been used to generate the chemistry code coupled with the TM5-MP. To further

examine the impact of the solver on the TM5-MP atmospheric simulations and performance, the mCB05 scheme is also tested using the Rosenbrock solver, as well as an intermediate step between the existing and new chemistry version of the model. Global budgets of $O_3$, CO, and OH, for all simulations performed for this work, are calculated and compared with estimates published in the literature. In more detail, the $O_3$ budget calculated with the MOGUNTIA chemistry scheme falls within one standard-deviation of mean estimates from other modelling studies. However, the new MOGUNTIA scheme reduces the

tropospheric $O_3$ burden by ~3% compared with the mCB05 configurations. For tropospheric CO, a respective reduction in the atmospheric lifetime (~6%) provides evidence that the implementation of the MOGUNTIA chemistry leads to an increase in the oxidative capacity of the troposphere in TM5-MP. This also holds for the atmospheric $CH_4$ chemical lifetime that is calculated here to be about 8.0 yr for the MOGUNTIA chemistry scheme, which is roughly 3-5% shorter compared to mCB05(KPP) and mCB05(EBI) configurations.

The large-scale variability in space and time of modeled tropospheric $NO_2$, OH, $O_3$, CO, and light VOCs (i.e., $C_2H_6$, $C_2H_4$, $C_3H_8$, $C_3H_6$) has been evaluated for the year 2006 and compared to several sets of in-situ observations, satellite retrievals, and climatological data. Overall, both the lumped-structure (i.e., the mCB05) and the lumped-molecule (i.e., the MOGUNTIA) mechanisms appear to be able to satisfactorily represent the tropospheric chemistry. In most of the cases however, when the model uses the MOGUNTIA chemistry configuration generally produces lower biases compared to measurements. The model

simulates well the major observed features of the spatial and temporal variability in surface observations for $O_3$ and CO. The observed background surface $O_3$ mixing ratios are captured with a bias of ~6.5 ppb for the MOGUNTIA configuration, very close to the mCB05 configurations. Ozone in the vertical matches on average within ~5 ppb for all configurations, and the model is able to capture the variability observed by ozone sondes well. On the other hand, for all configurations, the model underestimates the available CO flask observations by roughly 30%, most likely linked to uncertainties in the seasonal cycle

of anthropogenic emissions and the representation of biomass burning CO emissions. For the model comparison with observed light VOC mixing ratios, all chemistry configurations clearly show that significant uncertainties still exist regarding their emission strength in the model or not well-understood chemistry, such as the secondary chemical production during the decomposition of higher VOC in the atmosphere.



The presented model configurations result in a benchmark of the TM5-MP tropospheric chemistry version upon which future model improvements may take place. Inherent uncertainties need to be reduced and further work is required, focusing mainly on the most poorly understood chemistry-related processes. For example, further attention concerns the uncertainties in NO-$NO_2$-$O_3$ cycling along with the resulted organic $NO_X$ reservoirs formation and their impacts on the oxidative capacity of the

5 troposphere, as well as the treatment of aerosols and clouds, in particular ice clouds and their impact on photolysis rates. Other issues needed to be resolved are related to the significant uncertainties in light hydrocarbons mixing ratios, as clearly noticed by the model comparison to surface and aircraft observations and, thus, their potential impact on the oxidative capacity of the troposphere. Considering, however, that both chemistry schemes underestimate light VOCs mixing ratios in most of the cases, the use of a more detailed scheme such as the MOGUNTIA will allow us to better understand the causes of this deviation

compared to a lumped representation of VOC chemistry as in the CB05 mechanism, e.g., especially over tropical regions with high biogenic VOC emissions under low-$NO_X$ conditions. For this, a more dedicated comparison of the model with *in-situ* observations and satellite retrievals is planned to be performed in the future. MOGUNTIA contains also an ample number of oxygenated VOCs that are observed in the atmosphere at significant levels and further involved in aerosol formation, making thus the use of this scheme appropriate for such studies. On top of this, the implementation of the KPP software in the model

makes the code a lot more flexible for chemistry updates compared to the previous EBI-based chemistry versions. The use of the KPP in TM5-MP reduces the uncertainties in solving stiff chemistry equations and opens up new possibilities on model development, such as to construct an adjoint of the chemistry mechanism that can be used in the 4D-VAR data assimilation system of the model (e.g., Henze et al., 2007), or even to more accurately explore atmospheric chemistry-climate interactions, since TM5-MP is also coupled to the Earth System Model EC-Earth (e.g., Van Noije et al., 2014; Van Noije et al., manuscript

in preparation). On the other hand, and despite the clear benefits regarding code development and management, the use of a more sophisticated solver such as the Rosenbrock solver, and the implementation of a detailed chemistry scheme such as the MOGUNTIA, make unavoidably the code computationally more expensive, as indicated here by increases of the model's performance (i.e., ~18% and ~27%, respectively). Overall, this work shows that the newly coupled chemistry version of TM5-MP works as good, or better in some of the cases, as the previous chemistry versions of the model, giving confidence to further

chemistry developments and more detailed tropospheric investigations from the TM and EC-Earth communities.



**Code availability.** The TM5-MP code used for this study can be downloaded from Zenodo (doi:10.5281/zenodo.3759201); a request to generate a new user account for access the SVN server hosted at KNMI, the Netherlands, can be made by e-mailing to P. Le Sager (sager@knmi.nl). Any new user groups need to agree to the protocol set out for use, where it is expected that any developments are accessible to all users after the publication of results. Attendance at 9-monthly TM5 international
5  meetings is encouraged to avoid duplicity and conflict of interests.

**Supplement.** The Supplement related to this article is available online.

**Competing interests.** The authors declare that they have no conflict of interest.

**Author contributions.** This paper resulted from the deliberations of 27th International TM5 Meeting, 28-29 June 2018, Utrecht, the Netherlands (SM, MCK, TvN, PLS, SH, ND, MK). SM and MCK developed the chemistry code coupled to the model. SM and MK provided the original chemistry scheme equations. JEW developed both the photolysis code and mCB05 chemical mechanism, including the implementation of updated photolysis frequencies for the additional organics included in
the MOGUNTIA chemistry scheme. AG contributed to reaction data updates and coupling. AH developed and provided model evaluation tools with satellite retrievals. VH provided model evaluation tools and a collection of observation data. SM, ND, AH, and PLS performed the model evaluation. SM wrote the manuscript and all authors contributed to the preparation of this paper.

**Acknowledgements.** Stelios Myriokefalitakis acknowledges financial support for this research from the European Union's Horizon 2020 research and innovation programme under the Marie Skłodowska-Curie grant agreement no. 705652 – ODEON. Maarten C. Kroll is supported by the European Research Council (ERC) under the European Union's Horizon 2020 research and innovation programme under grant agreement no. 742798 – COS-OCS. Stelios Myriokefalitakis and Angelos Gouvousis acknowledge financial support from the National Observatory of Athens research grant (no. 5065 - Atmospheric Deposition
Impacts on the Ocean System). Model development was carried out on the Greek Research & Technology Network (GRNET) in the National HPC facility ARIS, the Dutch national e-infrastructure with the support of the SURF Cooperative and the ECMWF CRAY XC40 high-performance computer facility. Model simulations were performed at the GRNET HPC ARIS and the AETHER HPC cluster at the University of Bremen, funded by DFG within the scope of the Excellence Initiative. MOPITT CO data were obtained from the NASA Langley Research Center Atmospheric Science Data Center. NO$_2$ and HCHO
satellite data from OMI and SCIAMACHY were produced in the scope of the European FP7 project QA4ECV (grant no. 6007405). This paper is dedicated to the memory of Dr. Andreas Hilboll.





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



## Tables

**Table 1. Photolysis reactions (J) in the MOGUNTIA chemistry scheme.**

| # | Reactants | | Products[#] | References |
|---|---|---|---|---|
| J1 | $O_3 + hv$ | → | $O(^1D)$ | 1 |
| J2 | $H_2O_2 + hv$ | → | 2 OH | 1 |
| J3 | $NO_2 + hv$ | → | NO + O | 1 |
| J4 | $NO_3 + hv$ | → | $NO_2 + O$ | 1 |
| J5 | $NO_3 + hv$ | → | NO | 1 |
| J6 | $N_2O_5 + hv$ | → | $NO_2 + NO_3$ | 1 |
| J7 | $N_2O_5 + hv$ | → | $NO + NO_3$ | 1 |
| J8 | $HONO + hv$ | → | OH + NO | 1 |
| J9 | $HNO_3 + hv$ | → | $NO_2 + OH$ | 1 |
| J10 | $HNO_4 + hv$ | → | $NO_2 + HO_2$ | 1 |
| J11 | $HCHO + hv$ | → | CO | 1 |
| J12 | $HCHO + hv$ | → | $CO + 2 HO_2$ | 1 |
| J13 | $CH_3OOH + hv$ | → | $HCHO + HO_2 + OH$ | 1 |
| J14 | $CH_3ONO_2 + hv$ | → | $HCHO + HO_2 + NO_2$ | 1 |
| J15 | $CH_3OONO_2 + hv$ | → | $CH_3OO + NO_2$ | 1 |
| J16 | $CH_3OONO_2 + hv$ | → | $HCHO + HO_2 + NO_3$ | 1 |
| J17 | $CH_3C(O)OONO_2 + hv$ | → | $CH_3C(O)OO + NO_2$ | J10 |
| J18 | $CH_3C(O)OONO_2 + hv$ | → | $CH_3OO + NO_3 + CO_2$ | J10 |
| J19 | $CH_3C(O)OOH + hv$ | → | $CH_3C(O)OO + OH$ | J13 |
| J20 | $C_2H_5OOH + hv$ | → | $CH_3CHO + HO_2 + OH$ | J13 |
| J21 | $C_2H_5ONO2 + hv$ | → | $HCHO + CO + HO_2 + NO_2$ | 1 |
| J22 | $HOCH_2CH_2OOH + hv$ | → | $2 HCHO + HO_2 + OH$ | f 0.5 * J13 |
| J23 | $HOCH_2CH_2OOH + hv$ | → | $HOCH_2CHO + HO_2 + OH$ | (1 − f) 0.5 *J13 |
| J24 | $HOCH_2CH_2ONO_2 + hv$ | → | $2 HCHO + HO_2 + NO_2$ | f 0.5 * JORGN |
| J25 | $HOCH_2CH_2ONO_2 + hv$ | → | $HOCH_2CHO + HO_2 + NO_2$ | (1 − f) 0.5 * JORGN |
| J26 | $CH_3CHO + hv$ | → | $CH_3OO + CO + HO_2$ | 1 |
| J27 | $HOCH_2CHO + hv$ | → | $CH_3OH + CO$ | 1 |
| J28 | $CHOCHO + hv$ | → | $2 CO + 2 HO_2$ | 1 |
| J29 | $CHOCHO + hv$ | → | HCHO + CO | 1 |
| J30 | $CHOCHO + hv$ | → | 2 CO | 1 |
| J31 | $CH_3C(O)CH_3 + hv$ | → | $2 CH_3OO + CO$ | 1 |
| J32 | $CH_3C(O)CH_3 + hv$ | → | $CH_3C(O)OO + CH_3OO$ | 1 |
| J33 | $HOCH_2C(O)CH_3 + hv$ | → | $CH_3C(O)OO + HCHO + HO_2$ | 1 |
| J34 | $CH_3C(O)CH_2OOH + hv$ | → | $0.3 CH_3C(O)CHO \ 0.7(CH_3C(O)OO + HCHO) + OH$ | J13 |
| J35 | $n\text{-}C_3H_7OOH + hv$ | → | $C_2H_5CHO + HO_2 + OH$ | 0.5 * J13 |
| J36 | $n\text{-}C_3H_7ONO_2 + hv$ | → | $C_2H_5CHO + HO_2 + NO_2$ | 1 |
| J37 | $i\text{-}C_3H_7OOH + hv$ | → | $CH_3C(O)CH_3 + HO_2 + OH$ | 0.5 * J13 |
| J38 | $i\text{-}C_3H_7ONO_2 + hv$ | → | $CH_3C(O)CH_3 + HO_2 + NO_2$ | 1 |
| J39 | $C_2H_5CHO + hv$ | → | $C_2H_5OO + CO + HO_2$ | 1 |





| J40 | $HOC_3H_6OOH + hv$ | → | $CH_3CHO + HCHO + HO_2$ | J13 |
|---|---|---|---|---|
| J41 | $CH_3COCHO + hv$ | → | $CH_3C(O)OO + CO + HO_2$ | 1 |
| J42 | $C_4H_9OOH + hv$ | → | $0.67(CH_3CH_2COCH_3 + HO_2) + 0.33(C_2H_5OO + CH_3CHO) + OH$ | J13 |
| J43 | $C_4H_9ONO_2 + hv$ | → | $0.67(CH_3CH_2COCH_3 + HO_2) + 0.33(C_2H_5OO + CH_3CHO) + NO_2$ | $J_{ORGN}$ |
| J44 | $CH_3CH_2C(O)CH_3 + hv$ | → | $CH_3C(O)OO + C_2H_5OO$ | 1 |
| J45 | $CH_3CH(OOH)COCH_3 + hv$ | → | $CH_3CHO + CH_3C(O)OO + OH$ | J13 |
| J46 | $CH_3CH(ONO_2)COCH_3 + hv$ | → | $CH_3CHO + CH_3C(O)OO + NO_2$ | $J_{ORGN}$ |
| J47 | $ISOPOOH + hv$ | → | $HCHO + 0.64 MVK + 0.36 MACR + HO_2 + OH$ | 13 |
| J48 | $ISOPONO_2 + hv$ | → | $HCHO + 0.64 MVK + 0.36 MACR + HO_2 + NO_2$ | $J_{ORGN}$ |
| J49 | $MACR + hv$ | → | $0.5 MACROO + 0.5 HCHO + 0.175 CH_3C(O)OO + 0.325 CH_3OO + 0.825 CO + HO_2$ | 1 |
| J50 | $MACROOH + hv$ | → | $CH_3COCH_2OH + CO + HO_2 + OH$ | J13 |
| J51 | $MACRONO_2 + hv$ | → | $CH_3COCH_2OH + CO + HO_2 + NO2$ | $J_{ORGN}$ |
| J52 | $MVK + hv$ | → | $0.6 (C_3H_6 + CO) + 0.4 (CH_3C(O)OO + CH_3OO + HCHO)$ | 1 |
| J53 | $MVKOOH + hv$ | → | $0.7(CH_3C(O)OO + HOCH_2CHO) + 0.3(CH_3COCHO + HCHO + HO_2) + OH$ | J13 |
| J54 | $MVKONO_2 + hv$ | → | $0.7(CH_3C(O)OO + HOCH_2CHO) + 0.3(CH_3COCHO + HCHO + HO_2) + NO_2$ | $J_{ORGN}$ |
| J55 | $CH_3C(O)C(O)CH_3 + hv$ | → | $2 CH_3C(O)OO$ | 1 |
| J56 | $CH_3C(O)COOH + hv$ | → | $CH_3C(O)OO + HO_2 + CO_2$ | 1 |
| J57 | $HPALD + hv$ | → | $0.5 HOCH_2C(O)CH_3 + 0.5 CH_3COCHO + 0.25 HOCH_2CHO + 0.25 CHOCHO + HCHO + HO_2 + OH$ | 4, 5 |
| J58 | $O_2 + hv$ | → | $O_3$ | 1 |

[#] The reaction products $O_2$, $H_2$, and $H_2O$ are not shown.

[1] http://iupac.pole-ether.fr

[2] Atkinson, (1997):

$R_1 = 2.7 \times 10^{14} exp(-6350/T)$
$R_2 = 6.3 \times 10^{-14} exp(-550/T)$

$f = R_1/(R_1 + R_2 \times [O_2])$

[3] $J_{ORGN}$ is calculated based on average of σ-values for 1-$C_4H_9ONO_2$ and 2-$C_4H_9ONO_2$ as described in Williams et al. (2012)

[4] Browne et al. (2014)

[5] Peeters and Müller (2010)





**Table 2. Thermal reactions (K) in MOGUNTIA chemistry scheme.**

| # | Reactants | | Products# | Rate expression$ | References |
|---|-----------|---|-----------|------------------|------------|
| K0a | $O(^1D)$ (+ M) | | O | $3.3\times10^{-11}\exp(55/T)[O_2]$ + $2.5\times10^{-11}\exp(110/T)$ $[N_2]$ | 1 |
| K0b | $O(^1D)$ +$H_2O$ | | OH + OH | $1.63\times10^{-10}\exp(60/T)$ | 1 |
| K1 | $O_3$ + OH | → | $HO_2$ | $1.7 \times 10^{-12}\exp(-940/T)$ | 1 |
| K2 | $HO_2$ + $O_3$ | → | OH | $2.03 \times 10^{-16} (T/300)^{4.57} \exp(693/T)$ | 1 |
| K3 | $HO_2$ + OH | → | $H_2O$ | $4.8 \times 10^{-11}\exp(250/T)$ | 1 |
| K4 | $HO_2$ + $HO_2$ | → | $H_2O_2$ | $2.2\times10^{-13}\exp(600/T)$ $1.9\times10^{-33}$ $[N_2]$ $\exp(980/T)$ $1.4\times10^{-21}$ $[H_2O]$ $\exp(2200/T)$ | 1 |
| K5 | $H_2O_2$ + OH | → | $HO_2$ | $2.9 \times 10^{-12}\exp(-160/T)$ | 1 |
| K6 | $HO_2$ + NO | → | $NO_2$ + HO | $3.45 \times 10^{-12}\exp(270/T)$ | 1 |
| K7 | NO + $O_3$ | → | $NO_2$ | $2.07 \times 10^{-12}\exp(-1400/T)$ | 1 |
| K8 | NO + $NO_3$ | → | $2NO_2$ | $1.8 \times 10^{-11}\exp(110/T)$ | 1 |
| K9 | $NO_2$ + $O_3$ | → | $NO_3$ | $1.4 \times 10^{-13}\exp(-2470/T)$ | 1 |
| K10 | OH + NO {+ M} | → | HONO | $7.4\times10^{-31} \times(T/300)^{-2.4}$ $[N_2]$ $3.3\times10^{-11}(T/300)^{-0.3}$ $Fc = 0.81$ | 1 |
| K11 | OH + $NO_2$ {+ M} | → | $HONO_2$ | $3.2 \times 10^{-30}(T/300)^{-4.5}[N_2]$ $3.0 \times 10^{-11}$ $Fc = 0.41$ | 1 |
| K12 | $NO_2$ + $NO_3$ {+ M} | → | $N_2O_5$ | $3.6 \times 10^{-30}(T/300)^{-4.1}[N_2]$ $1.9 \times 10^{-12}(T/300)^{0.2}$ $Fc = 0.35$ | 1 |
| K13 | $NO_2$ + $HO_2$ | → | $HO_2NO_2$ | $1.4 \times 10^{-31}(T/300)^{-3.1}[N_2]$ $4.0 \times 10^{-12}$ $Fc = 0.40$ | 1 |
| K14 | $HO_2$ + $NO_3$ | → | OH + $NO_2$ | $4.0 \times 10^{-12}$ | 1 |
| K15 | HONO + OH | → | $NO_2$ | $2.5\times10^{-12}\exp(260/T)$ | 1 |
| K16 | $HNO_3$ + OH | → | $NO_3$ | $2.4\times10^{-14}\exp(460/T)$ $6.5\times10^{-34}\exp(1335/T)$ $2.7\times10^{-17}\exp(2199/T)$ | 1 |
| K17 | $HO_2NO_2$ + OH | → | $NO_2$ | $1.9 \times 10^{-12}\exp(270/T)$ | 1 |
| K18 | $HO_2NO_2$ | → | $HO_2$ + $NO_2$ | $4.1 \times 10^{-5}\exp(-10650/T)[N_2]$ $6.0 \times 10^{15}\exp(-11170/T)$ $Fc = 0.40$ | 1 |
| K19 | $N_2O_5$ | → | $NO_2$ + $NO_3$ | $1.3 \times 10^{-3}(T/300)^{-3.5}\exp(-11000/T)[N_2]$ $9.7 \times 10^{14}(T/300)^{0.1}\exp(-11080/T)$ $Fc = 0.35$ | 1 |
| K20 | OH + $H_2$ | → | $HO_2$ | $7.7\times10^{-12}\exp(-2100/T)$ | 1 |
| K21 | $CH_4$ + OH | → | $CH_3OO$ | $2.45 \times 10^{-12}\exp(-1775/T)$ | 2 |
| K22 | $CH_3OO$ + $HO_2$ | → | $CH_3OOH$ | $3.8 \times 10^{-13}\exp(780/T)*$ $(1-1/(1+498.0\exp(-1160/T)))$ | 1, 3 |



| | | | | | |
|---|---|---|---|---|---|
| K23 | $CH_3OO + HO_2$ | → | HCHO | $3.8 \times 10^{-13}exp(780/T)*$ $(1/(1+498.0exp(-1160/T)))$ | 1, 3 |
| K24 | $CH_3OO + NO$ | → | 0.999 (HCHO + $HO_2$ + $NO_2$) + 0.001 $CH_3ONO_2$ | $2.3 \times 10^{-12}exp(360/T)$ | 1, 3 |
| K25 | $CH_3OO + NO_2$ | → | $CH_3O_2NO_2$ | $2.5 \times 10^{-30}(T/300)^{-5.5}[N_2]$ $1.8 \times 10^{-11}$ $F_c = 0.36$ | 1 |
| K26 | $CH_3OO + NO_3$ | → | HCHO + $NO_2$ | $1.2 \times 10^{-12}$ | 1 |
| K27 | $CH_3OO + CH_3OO$ | → | 2HCHO + $2HO_2$ | 7.4 exp(-520/T) x $1.03 \times 10^{-13}exp(365/T)$ | 1, 3 |
| K28 | $CH_3OO + CH_3OO$ | → | $CH_3OH$ + HCHO | (1 -7.4 exp(-520/T) x $1.03 \times 10^{-13}exp(365/T)$ | 1, 3 |
| K29 | $CH_3OOH + OH$ | → | HCHO + OH | $0.4 \times 5.3 \times 10^{-12}exp(190/T)$ | 1 |
| K30 | $CH_3OOH + OH$ | → | $CH_3OO$ | $0.6 \times 5.3 \times 10^{-12}exp(190/T)$ | 1 |
| K31 | $CH_3ONO_2 + OH$ | → | HCHO + $NO_2$ | $4.0 \times 10^{-13}exp(-845/T)$ | 1 |
| K32 | $CH_3OONO_2$ | → | $CH_3O_2 + NO_2$ | $9.0 \times 10^{-5}exp(-9690/T )$ $[N_2]$ $1.1 \times 10^{16}exp(-10560/T )$ $F_c = 0.40$ | 1 |
| K33 | HCHO + OH | → | CO + $HO_2$ | $5.4 \times 10^{-12}exp(135/T)$ | 1 |
| K34 | HCHO + $NO_3$ | → | CO + $HO_2$ + $HNO_3$ | $2.0 \times 10^{-12}exp(-2440/T)$ | 1 |
| K35 | $CH_3OH + OH$ | → | HCHO + $HO_2$ | $2.85 \times 10^{-12}exp(-345/T)$ | 1 |
| K36 | $CH_3OH + NO_3$ | → | HCHO + $HO_2$ + $HNO_3$ | $9.4 \times 10^{-13}exp(-2650/T)$ | 1 |
| K37 | HCOOH + OH | → | $CO_2$ + $HO_2$ | $4.5 \times 10^{-13}$ | 1 |
| K38 | CO + OH | → | $CO_2$ + $HO_2$ | $5.9 \times 10^{-33}(300/T)^{1.4}$ $1.1 \times 10^{-12}(300/T)^{-1.3}$ $1.5 \times 10^{-13}(300/T)^{-0.6}$ $2.9 \times 10^{9}(300/T)^{-6.1}$ | 2 |
| K39 | $C_2H_6 + OH$ | → | $C_2H_5OO$ | $6.9 \times 10^{-12}exp(-1000/T)$ | 1 |
| K40 | $C_2H_5OO + HO_2$ | → | $C_2H_5OOH$ | $6.4 \times 10^{-13}exp(710/T)$ | 1 |
| K41 | $C_2H_5OO + NO$ | → | $CH_3CHO + HO_2 + NO_2$ | (1 -RTC2P) x $2.55 \times 10^{-12}$ exp(380/T) | 1, 4 |
| K42 | $C_2H_5OO + NO$ | → | $C_2H_5ONO_2$ | RTC2P x $2.55 \times 10^{-12}$ exp(380/T) | 1, 4 |
| K43 | $C_2H_5OO + CH_3OO$ | → | $CH_3CHO$ + HCHO + $2HO_2$ | 0.8 x $(6.4 \times 10^{-14} \times 1.03 \times 10^{-13}exp(365/T))^{0.5}$ | 3 |
| K44 | $C_2H_5OO + CH_3OO$ | → | 0.5 $CH_3CHO$ + 0.5 $CH_3CH_2OH$ + $CH_3OH$ | 0.2 x $(6.4 \times 10^{-14} \times 1.03 \times 10^{-13}exp(365/T))^{0.5}$ | 3 |
| K45 | $C_2H_5OOH + OH$ | → | $C_2H_5OO$ | $1.90 \times 10^{-12}exp(190/T)$ | 1 |
| K46 | $C_2H_5OOH + OH$ | → | $CH_3CHO$ + OH | $6.0 \times 10^{-12}$ | 3 |
| K47 | $C_2H_5ONO_2 + OH$ | → | $CH_3CHO + NO_2$ | $6.7 \times 10^{-13}exp(-395/T)$ | 1 |
| K48 | $CH_3CHO + OH$ | → | $CH_3C(O)OO$ | $4.7 \times 10^{-12}exp(345/T)$ | 1 |
| K49 | $CH_3CHO + NO_3$ | → | $CH_3C(O)OO + HNO_3$ | $1.4 \times 10^{-12}exp(-1860/T)$ | 1 |
| K50 | $CH_3C(O)OO + HO_2$ | → | $CH_3C(O)OOH$ | $0.41 * 5.2 \times 10^{-13}exp(980/T)$ | 3 |
| K51 | $CH_3C(O)OO + HO_2$ | → | $CH_3COOH + O_3$ | $0.15 * 5.2 \times 10^{-13}exp(980/T)$ | 3 |
| K52 | $CH_3C(O)OO + HO_2$ | → | $CH_3O_2 + CO_2 + OH$ | $0.44 * 5.2 \times 10^{-13}exp(980/T)$ | 3 |
| K53 | $CH_3C(O)OO + NO$ | → | $CH_3OO + CO_2 + NO_2$ | $7.5 \times 10^{-12}exp(290/T)$ | 1 |
| K54 | $CH_3C(O)OO + NO_2$ | → | $CH_3C(O)OONO_2$ | $3.28 \times 10^{-28}(T/300)^{-6.87}[N_2]$ | 1 |





| | | | | |
|---|---|---|---|---|
| | | | $1.125 \times 10^{-11}(T/300)^{-1.105}$ | |
| | | | $Fc = 0.3$ | |
| K55 | $CH_3C(O)OO + NO_3$ | $\rightarrow$ $CH_3OO + NO_2$ | $4.0 \times 10^{-12}$ | 2 |
| K56 | $CH_3C(O)OO + CH_3OO$ | $\rightarrow$ $CH_3C(O)OOH + HCHO$ | $0.9 * 2.0 \times 10^{-12}\exp(500/T)$ | 2 |
| K57 | $CH_3C(O)OO + CH_3OO$ | $\rightarrow$ $CH_3COOH + HCHO$ | $0.1 * 2.0 \times 10^{-12}\exp(500/T)$ | 2 |
| K58 | $CH_3C(O)OO + CH_3C(O)OO$ | $\rightarrow$ $2 (CH_3OO + CO_2)$ | $2.9 \times 10^{-12}\exp(500/T)$ | 2 |
| K59 | $CH_3C(O)OO+ CH_3COCH_2O_2$ | $\rightarrow$ $CH_3COOH + CH_3COCHO$ | $2.5 \times 10^{-12}$ | 2 |
| K60 | $CH_3C(O)OO+ CH_3COCH_2O_2$ | $\rightarrow$ $CH_3OO + CH_3COCH_2OH + CO_2$ | $2.5 \times 10^{-12}$ | 2 |
| K61 | $CH_3C(O)OO + C_2H_5OO$ | $\rightarrow$ $CH_3CHO + 2 CH_3OO$ | $0.7 * 4.4 \times 10^{-13}\exp(1070/T)$ | 1, 3 |
| K62 | $CH_3C(O)OO + C_2H_5OO$ | $\rightarrow$ $CH_3CHO + CH_3COOH$ | $0.3 * 4.4 \times 10^{-13}\exp(1070/T)$ | 1, 3 |
| K63 | $CH_3C(O)OONO_2 + OH$ | $\rightarrow$ $HCHO + CO + NO_2$ | $3.0 \times 10^{-14}$ | 1 |
| K64 | $CH_3C(O)OONO_2$ | $\rightarrow$ $CH_3C(O)OO + NO_2$ | $1.1 \times 10^{-5}\exp(-10100/T)[N_2]$ $1.9 \times 10^{17}\exp(-14100/T)$ $Fc = 0.3$ | 1 |
| K65 | $CH_3C(O)OONO_2$ | $\rightarrow$ $CH_3ONO_2 + CO_2$ | $2.1 \times 10^{12} \exp(-12525/T)$ | 5 |
| K66 | $CH_3C(O)OOH + OH$ | $\rightarrow$ $CH_3C(O)OO$ | $1.1 \times 10^{-11}$ | 3 |
| K67 | $C_2H_4 + OH$ | $\rightarrow$ $HOCH_2CH_2OO$ | $8.6 \times 10^{-29}(T/300)^{-3.1}[N_2]$ $9.0 \times 10^{-12}(T/300)^{-0.85}$ $Fc = 0.48$ | 1 |
| K68 | $C_2H_4 + NO_3$ | $\rightarrow$ $HOCH_2CH_2ONO_2$ | $3.3 \times 10^{-12}\exp(-2880/T)$ | 1 |
| K69 | $C_2H_4 + O_3$ | $\rightarrow$ $1.37 HCHO + 0.63 CO + 0.13 HO_2 + 0.13 OH$ | $6.82 \times 10^{-15}\exp(-2500/T)$ | 1 |
| K70 | $HOCH_2CH_2OO + HO_2$ | $\rightarrow$ $HOCH_2CH_2OOH$ | $1.3 \times 10^{-11}$ | 1 |
| K71 | $HOCH_2CH_2OO + NO$ | $\rightarrow$ $NO_2 + 2HCHO + HO_2$ | $(1-RTC2P) \times f \times 2.7 \times 10^{-12} \exp(360/T)$ | 3 |
| K72 | $HOCH_2CH_2OO+ NO$ | $\rightarrow$ $NO_2 + HOCH_2CHO + HO_2$ | $(1-RTC2P) \times (1-f) \times 2.7 \times 10^{-12} \exp(360/T)$ | 3 |
| K73 | $HOCH_2CH_2OO+ NO$ | $\rightarrow$ $HOCH_2CH_2ONO_2$ | $RTC2P \times 2.7 \times 10^{-12} \exp(360/T)$ | 1 |
| K74 | $HOCH_2CH_2OO + CH_3OO$ | $\rightarrow$ $HOCH_2CHO + HCHO + 2HO_2$ | $0.8 * (7.8 \times 10^{14}\exp(1000/T) * 1.03 \times 10^{-13}\exp(365/T))^{0.5}$ | 3 |
| K75 | $HOCH_2CH_2OO + CH_3OO$ | $\rightarrow$ $HOCH_2CHO + CH_3OH$ | $0.2 * (7.8 \times 10^{14}\exp(1000/T) * 1.03 \times 10^{-13}\exp(365/T))^{0.5}$ | 3 |
| K76 | $HOCH_2CH_2OOH + OH$ | $\rightarrow$ $HOCH_2CH_2OO$ | K45 | |
| K77 | $HOCH_2CH_2OOH + OH$ | $\rightarrow$ $HOCH_2CHO + OH$ | $1.38 \times 10^{-11}$ | 3 |
| K78 | $HOCH_2CH_2ONO_2 + OH$ | $\rightarrow$ $HOCH_2CHO + NO_2$ | K47 | |
| K79 | $C_2H_2 + OH$ | $\rightarrow$ $0.636(CHOCHO + OH) + 0.364(HCOOH + CO + HO_2)$ | $5.0 \times 10^{-30}(T/300)^{-1.5}[N_2]$ $1.0 \times 10^{-12}$ $Fc = 0.37$ | 1 |
| K80 | $C_2H_2 + NO_3$ | $\rightarrow$ $0.635 CHOCHO + 0.365(HCOOH + CO) + HNO_3$ | $1.0 \times 10^{-16}$ | 1 |
| K81 | $C_2H_2 + O_3$ | $\rightarrow$ $0.635 CHOCHO + 0.365(HCOOH + CO)$ | $1.0 \times 10^{-20}$ | 1 |





| | | | | | |
|---|---|---|---|---|---|
| K82 | $HOCH_2CHO + OH$ | → | $HCHO + CO_2$ | $6.4 \times 10^{-12}$ | 1 |
| K83 | $HOCH_2CHO + OH$ | → | $CHOCHO + HO_2$ | $1.6 \times 10^{-12}$ | 1 |
| K84 | $CHOCHO + OH$ | → | $2CO + HO_2$ | $3.1 \times 10^{-12} \exp(340/T)$ | 1 |
| K85 | $CHOCHO + NO_3$ | → | $2CO + HO_2 + HNO_3$ | $4.0 \times 10^{-16}$ | 1 |
| K86 | $CH_3COOH + OH$ | → | $CH_3OO + CO_2$ | $4.0 \times 10^{-14} \exp(850/T)$ | 1 |
| K87 | $CH_3CH_2OH + OH$ | → | $0.95\ (CH_3CHO + HO_2) + 0.05\ HOCH_2CH_2OO$ | $3.0 \times 10^{-12} \exp(20/T)$ | 1 |
| K88 | $C_3H_8 + OH$ | → | $0.264\ n\text{-}C_3H_7O_2 + 0.736\ i\text{-}C_3H_7O_2$ | $7.6 \times 10^{-12} \exp(-585/T)$ | 1, 3 |
| K89 | $n\text{-}C_3H_7O_2 + HO_2$ | → | $n\text{-}C_3H_7OOH$ | $0.52 \times 2.91 \times 10^{-13} \exp(1300/T)$ | 3 |
| K90 | $n\text{-}C_3H_7O_2 + NO$ | → | $C_2H_5CHO + HO_2 + NO_2$ | $(1 - RTC3P) \times 2.9 \times 10^{-12} \exp(350/T)$ | 1, 4 |
| K91 | $n\text{-}C_3H_7O_2 + NO$ | → | $n\text{-}C_3H_7ONO_2$ | $RTC3P \times 2.9 \times 10^{-12} \exp(350/T)$ | 1, 4 |
| K92 | $n\text{-}C_3H_7O_2 + CH_3OO$ | → | $C_2H_5CHO + CH_3OH$ | $0.8 \times (3.5 \times 10^{-13} \times 3.0 \times 10^{13})^{0.5}$ | 3 |
| K93 | $n\text{-}C_3H_7O_2 + CH_3OO$ | → | $C_2H_5CHO + HCHO + 2HO_2$ | $0.2 \times (3.5 \times 10^{-13} \times 3.0 \times 10^{13})^{0.5}$ | 3 |
| K94 | $n\text{-}C_3H_7OOH + OH$ | → | $n\text{-}C_3H_7O_2$ | K76 | |
| K95 | $n\text{-}C_3H_7OOH + OH$ | → | $C_2H_5CHO + OH$ | $1.66 \times 10^{-11}$ | 3 |
| K96 | $n\text{-}C_3H_7ONO_2 + OH$ | → | $C_2H_5CHO + NO_2$ | $5.8 \times 10^{-13}$ | 1 |
| K97 | $i\text{-}C_3H_7O_2 + HO_2$ | → | $i\text{-}C_3H_7OOH$ | K89 | |
| K98 | $i\text{-}C_3H_7O_2 + NO$ | → | $CH_3COCH_3 + HO_2 + NO_2$ | $(1 - RTC3S) * 2.7 \times 10^{-12} \exp(360/T)$ | 1, 4 |
| K99 | $i\text{-}C_3H_7O_2 + NO$ | → | $i\text{-}C_3H_7ONO_2$ | $RTC3S * 2.7 \times 10^{-12} \exp(360/T)$ | 1, 4 |
| K100 | $i\text{-}C_3H_7O_2 + CH_3OO$ | → | $CH_3COCH_3 + HCHO + 2HO_2$ | $0.8 * (1.03 \times 10^{-13} \exp(365/T) *$ $1.6 \times 10^{-12} \exp(-2200/T))^{0.5}$ | 3 |
| K101 | $i\text{-}C_3H_7O_2 + CH_3OO$ | → | $CH_3COCH_3 + CH_3OH$ | $0.2 * (1.03 \times 10^{-13} \exp(365/T) \times$ $1.6 \times 10^{-12} \exp(-2200/T))^{0.5}$ | 3 |
| K102 | $i\text{-}C_3H_7OOH + OH$ | → | $i\text{-}C_3H_7O_2$ | $1.9 \times 10^{-12} \exp(190/T)$ | 3 |
| K103 | $i\text{-}C_3H_7OOH + OH$ | → | $CH_3COCH_3 + OH$ | $1.66 \times 10^{-11}$ | 3 |
| K104 | $i\text{-}C_3H_7ONO_2 + OH$ | → | $CH_3COCH_3 + NO_2$ | $6.2 \times 10^{-13} \exp(-230/T)$ | 1 |
| K105 | $C_2H_5CHO + OH$ | → | $CH_3C(O)OO + CO$ | $4.9 \times 10^{-12} \exp(405/T)$ | 1 |
| K106 | $C_2H_5CHO + NO_3$ | → | $CH_3C(O)OO + CO + HNO_3$ | $6.3 \times 10^{-15}$ | 1 |
| K107 | $CH_3COCH_3 + OH$ | → | $CH_3COCH_2OO$ | $8.8 \times 10^{-12} \exp(-1320/T) +$ $1.7 \times 10^{-14} \exp(423/T)$ | 1 |
| K108 | $CH_3COCH_2OO + NO$ | → | $CH_3COCHO + NO_2 + HO_2$ | $2.7 \times 10^{-13} \exp(360/T)$ | 3 |
| K109 | $CH_3COCH_2OO + HO_2$ | → | $CH_3COCH_2OOH$ | $1.36 \times 10^{-13} \exp(1250/T)$ | 3 |
| K110 | $CH_3COCH_2OOH + OH$ | → | $0.7\ CH_3COCHO + 0.3\ CH_3COCH_2OO + OH$ | $1.90 \times 10^{-12} \exp(190/T)$ | 3 |
| K111 | $C_3H_6 + OH$ | → | $HOC_3H_6OO$ | $8 \times 10^{-27}(T/300)^{-3.5}[N_2]$ $3.0 \times 10^{-11}(T/300)^{-1.0}$ $Fc = 0.5$ | 1 |
| K112 | $C_3H_6 + NO_3$ | → | $0.35\ n\text{-}C_3H_7ONO_2 + 0.65\ i\text{-}C_3H_7ONO_2$ | $4.6 \times 10^{-13} \exp(-1155/T)$ | 1, 3 |
| K113 | $C_3H_6 + O_3$ | → | $0.62\ HCHO + 0.62\ CH_3CHO + 0.38\ CH_3OO +$ $0.56\ CO + 0.36\ HO_2 + 0.36\ OH + 0.2\ CO_2$ | $5.77 \times 10^{-15} \exp(-1880/T)$ | 1, 3 |
| K114 | $HOC_3H_6OOH + OH$ | → | $0.928\ CH_3COCH_2OH + 0.072\ HOC_3H_6OO +$ $0.928\ OH$ | $2.44 \times 10^{-11} + 1.9 \times 10^{-12} \exp(190/T)$ | 3 |
| K115 | $HOC_3H_6OO + HO_2$ | → | $HOC_3H_6OOH$ | K89 | 3 |
| K116 | $HOC_3H_6OO + NO$ | → | $CH_3CHO + HCHO + HO_2 + NO_2$ | $(1 - 0.35RTC3P - 0.65RTC3S) *$ $2.55 \times 10^{-12} \exp(380/T)$ | 1, 3 |
| K117 | $HOC_3H_6OO + NO$ | → | $0.35\ n\text{-}C_3H_7ONO_2 + 0.65\ i\text{-}C_3H_7ONO_2$ | $(0.35RTC3P + 0.65RTC3S) *$ | 1, 3 |



| | | | | | |
|---|---|---|---|---|---|
| | | | | $2.55 \times 10^{-12}\exp(380/T)$ | |
| K118 | $HOC_3H_6OO + CH_3OO$ | $\rightarrow$ | $CH_3CHO + 2HCHO + 2HO_2$ | $0.8 * 6.0 \times 10^{-13}$ | 3 |
| K119 | $HOC_3H_6OO + CH_3OO$ | $\rightarrow$ | $CH_3COCH_2OH + CH_3OH$ | $0.2 * 6.0 \times 10^{-13}$ | 3 |
| K120 | $CH_3COCH_2OH + OH$ | $\rightarrow$ | $CH_3COCHO + HO_2$ | $1.6 \times 10^{-12}\exp(305/T)$ | 1 |
| K121 | $CH_3COCHO + OH$ | $\rightarrow$ | $CH_3C(O)OO + CO$ | $1.9 \times 10^{-12}\exp(575/T)$ | 1 |
| K122 | $CH_3COCHO + NO_3$ | $\rightarrow$ | $CH_3C(O)OO + CO + HNO_3$ | $5.0 \times 10^{-16}$ | 1 |
| K123 | $CH_3C(O)COOH + OH$ | $\rightarrow$ | $CH_3C(O)OO + CO_2$ | $8.0 \times 10^{-13}$ | 3 |
| K124 | $C_4H_{10} + OH$ | $\rightarrow$ | $C_4H_9OO$ | $9.8 \times 10^{-12}\exp(-425/T)$ | 3 |
| K125 | $C_4H_{10} + NO_3$ | $\rightarrow$ | $C_4H_9OO + HNO_3$ | $2.8 \times 10^{-12}\exp(-3280/T)$ | 1 |
| K126 | $C_4H_9OO + HO_2$ | $\rightarrow$ | $C_4H_9OOH$ | $0.625 * 2.91 \times 10^{-13}\exp(1300/T)$ | 3 |
| K127 | $C_4H_9OO + NO$ | $\rightarrow$ | $NO_2 + 0.67(CH_3CH_2COCH_3 + HO_2) + 0.33(C_2H_5OO + CH_3CHO)$ | $(1 - RTC4P) \times 8.3 \times 10^{-12}$ | 1, 4 |
| K128 | $C_4H_9OO + NO$ | $\rightarrow$ | $C_4H_9ONO_2$ | $RTC4P \times 8.3 \times 10^{-12}$ | 1, 4 |
| K129 | $C_4H_9OO + CH_3OO$ | $\rightarrow$ | $HCHO + HO_2 + 0.67(CH_3CH_2C(O)CH_3 + HO_2) + 0.33(CH_3CHO + CH_3CH_2OO)$ | $0.8 * 1.3 \times 10^{-12}$ | 3 |
| K130 | $C_4H_9OO + CH_3OO$ | $\rightarrow$ | $CH_3CH_2COCH_3 + CH_3OH$ | $0.2 * 1.3 \times 10^{-12}$ | 3 |
| K131 | $C_4H_9OOH + OH$ | $\rightarrow$ | $C_4H_9OO$ | $1.90 \times 10^{-12}\exp(190/T)$ | 3 |
| K132 | $C_4H_9OOH + OH$ | $\rightarrow$ | $CH_3CH_2COCH_3 + OH$ | $2.15 \times 10^{-11}$ | 3 |
| K133 | $C_4H_9ONO_2 + OH$ | $\rightarrow$ | $CH_3CH_2COCH_3 + NO_2$ | $8.6 \times 10^{-13}$ | 1 |
| K134 | $CH_3CH_2COCH_3 + OH$ | $\rightarrow$ | $CH_3CH(OO)COCH_3$ | $1.5 \times 10^{-12}\exp(-90/T)$ | 1 |
| K135 | $CH_3CH(OO)COCH_3 + HO_2$ | $\rightarrow$ | $CH_3CH(OOH)COCH_3$ | K126 | |
| K136 | $CH_3CH(OO)COCH_3 + NO$ | $\rightarrow$ | $CH_3CHO + CH_3C(O)OO + NO_2$ | $(1 - RTC4S) \times 2.55 \times 10^{-12} \exp(380/T)$ | 1, 4 |
| K137 | $CH_3CH(OO)COCH_3 + NO$ | $\rightarrow$ | $CH_3CH(ONO_2)COCH_3$ | $RTC4S \times 2.55 \times 10^{-12} \exp(380/T)$ | 1, 4 |
| K138 | $CH_3CH(OOH)COCH_3 + OH$ | $\rightarrow$ | $CH_3CH(OO)COCH_3$ | K131 | |
| K139 | $CH_3CH(OOH)COCH_3 + OH$ | $\rightarrow$ | $CH_3C(O)C(O)CH_3 + OH$ | $1.88 \times 10^{-11}$ | 3 |
| K140 | $CH_3CH(ONO_2)COCH_3 + OH$ | $\rightarrow$ | $CH_3C(O)C(O)CH_3 + NO_2$ | $1.2 \times 10^{-12}$ | 1 |
| K141 | $ISOP + OH$ | $\rightarrow$ | $0.98\ ISOPOO + 0.0003\ ELVOC + 0.007\ SVOC$ | $2.7 \times 10^{-11}\exp(390/T)$ | 1, 3 |
| K142 | $ISOP + NO_3$ | $\rightarrow$ | $ISOPONO_2$ | $2.95 \times 10^{-12}\exp(-450/T)$ | 1, 3 |
| K143 | $ISOP + O_3$ | $\rightarrow$ | $0.98 * (0.3\ MACR + 0.3\ MACROO + 0.2\ MVK + 0.2\ MVKOO + 0.78\ HCHO + 0.22CO + 0.125\ HO_2 + 0.125OH) + 0.0001\ ELVOC + 0.009\ SVOC$ | $1.05 \times 10^{-14}\exp(-2000/T)$ | 1, 3 |
| K144 | $ISOPOO + HO_2$ | $\rightarrow$ | $ISOPOOH$ | $2.06 \times 10^{-13}\exp(1300/T)$ | 3, 7 |
| K145 | $ISOPOO + NO$ | $\rightarrow$ | $HCHO + 0.64\ MVK + 0.36\ MACR + HO_2 + NO_2$ | $(1-RTC5S) * 2.7 \times 10^{-12}\exp(360/T)$ | 3 |
| K146 | $ISOPOO + NO$ | $\rightarrow$ | $ISOPONO_2$ | $RTC5S * 2.7 \times 10^{-12}\exp(360/T)$ | 3 |
| K147 | $ISOPOO + NO_3$ | $\rightarrow$ | $HCHO + 0.64\ MVK + 0.36\ MACR + HO_2 + NO_2$ | $2.3 \times 10^{-12}$ | 3 |
| K148 | $ISOPOO + CH_3OO$ | $\rightarrow$ | $0.64\ MVK + 0.36\ MACR + 2HCHO + 2HO_2$ | $0.8 * 2.65 \times 10^{-12}$ | 3 |





| K149 | ISOPOO + CH$_3$OO | → | 0.64 MVK + 0.36 MACR + HCHO + CH$_3$OH | 0.2 * 2.65 x 10$^{-12}$ | 3 |
|------|------|---|------|------|---|
| K150 | ISOPOO | → | HPALD + HO$_2$ | 4.12×10$^8$exp(-7700/T) | 6, 7 |
| K151 | ISOPOOH + OH | → | IEPOX + OH | 1.9×10$^{-11}$exp(-390/T) | 8 |
| K152 | ISOPOOH + OH | → | ISOPOO | 0.7 * 3.8×10$^{-12}$exp(-200/T) | 8 |
| K153 | ISOPOOH + OH | → | 0.64 CH$_3$COCHO + 0.64 HOCH$_2$CHO + 0.36 HOCH$_2$C(O)CH$_3$ + 0.36 CHOCHO + OH | 0.3 * 3.8×10$^{-12}$exp(-200/T) | 8, 9 |
| K154 | ISOPONO$_2$ + OH | → | 0.64 CH$_3$COCHO + 0.64 HOCH$_2$CHO + 0.36 HOCH$_2$C(O)CH$_3$ + 0.36 CHOCHO + NO$_2$ | 1.77×10$^{-11}$exp(-500/T) | 8 |
| K155 | HPALD + OH | → | 0.5 HOCH$_2$C(O)CH$_3$ + 0.5 CH$_3$C(O)CHO + 0.25 HOCH$_2$CHO + 0.25 CHOCHO + HCHO + HO$_2$ + OH | 4.6×10$^{-11}$ | 6 |
| K156 | IEPOX + OH | → | IEPOXOO | 5.78×10$^{-11}$exp(-400/T) | 8 |
| K157 | IEPOXOO + HO$_2$ | → | 0.725 HOCH$_2$C(O)CH$_3$+ 0.275 HOCH$_2$CHO + 0.275 HOCH$_2$CHO + 0.275 CH$_3$C(O)CHO + 1.125 OH + 0.825 HO$_2$ + 0.2 CO$_2$ + 0.375 HCHO + 0.074 HCOOH + 0.251 CO | 7.4×10$^{-13}$exp(700/T) | 8 |
| K158 | IEPOXOO + NO | → | 0.725 HOCH$_2$C(O)CH$_3$+ 0.275 HOCH$_2$CHO + 0.275 HOCH$_2$CHO + 0.275 CH$_3$C(O)CHO + 1.125 OH + 0.825 HO$_2$ + 0.2 CO$_2$ + 0.375 HCHO + 0.074 HCOOH + 0.251 CO + NO$_2$ | 2.7×10$^{-12}$exp(360/T) | 3 |
| K159 | IEPOXOO + NO$_3$ | → | 0.725 HOCH$_2$C(O)CH$_3$+ 0.275 HOCH$_2$CHO + 0.275 HOCH$_2$CHO + 0.275 CH$_3$C(O)CHO + 1.125 OH + 0.825 HO$_2$ + 0.2 CO$_2$ + 0.375 HCHO + 0.074 HCOOH + 0.251 CO + NO$_2$ | 1.74 * 2.3×10$^{-12}$ | 3 |
| K160 | MVK + OH | → | MVKOO | 2.6 x 10$^{-12}$exp(610/T) | 1 |
| K161 | MVK + NO$_3$ | → | 0.65 HCOOH + 0.65 CH$_3$COCHO + 0.35 HCHO + 0.35 CH$_3$C(O)OOH + HNO3 | 6.0 x 10$^{-16}$ | 1 |
| K162 | MVK + O$_3$ | → | 0.38 CH$_3$COCHO + 0.2088 CH$_3$C(O)OO + 0.26 CH$_3$COCOOH + 0.26 CO + 0.0432 CH$_3$COOH + 0.108 CH$_3$CHO + 0.62 HCHO + 048 CO$_2$ + 0.54 HO$_2$ + 0.1008 OH | 8.5 x 10$^{-16}$exp(-1520/T) | 1, 3 |
| K163 | MVKOO + HO$_2$ | → | MVKOOH | K144 | |
| K164 | MVKOO + NO | → | 0.295 CH$_3$C(O)CHO + 0.295 HCHO + 0.670 CH$_3$CHO + 0.670 HOCHCHO + 0.295 HO$_2$ + 0.965 NO$_2$ + 0.0352 MVKONO$_2$ | 2.6 x 10$^{-12}$exp(380/T) | 3 |
| K165 | MVKOOH + OH | → | CH$_3$C(O)CHO + CO + 2HO$_2$ + OH | 2.55 x 10$^{-11}$ | 3 |
| K166 | MVKOOH + OH | → | MVKOO | 1.9 x 10$^{-12}$exp(190/T) | 3 |
| K167 | MVKONO$_2$ + OH | → | CH$_3$C(O)CHO + CO + HO$_2$ + NO$_2$ | 1.33 x 10$^{-12}$ | 3 |
| K168 | MACR + OH | → | MACROO | 8.0 x 10$^{-12}$exp(380/T) | 1 |
| K169 | MACR + NO$_3$ | → | MACROO + HNO$_3$ | 3.4 x 10$^{-15}$ | 1 |
| K170 | MACR + O$_3$ | → | 0.90 CH$_3$COCHO + 0.5 HCHO + 0.5 CO + 0.14 HO$_2$ + 0.24 OH | 1.4 x 10$^{-15}$exp(-2100/T) | 1, 3 |
| K171 | MACROO + HO$_2$ | → | MACROOH | 0.625 * 2.91 x 10$^{-13}$exp(1300/T) | 3 |
| K172 | MACROO + NO | → | 0.987 (CH$_3$COCH$_2$OH + CO + NO$_2$ + HO$_2$) + 0.013 MACRONO$_2$ | K164 | 1, 3 |
| K173 | MACROOH + OH | → | CH$_3$COCH$_2$OH + CO + OH | 3.77 x 10$^{-11}$ | |



| | | | | | |
|---|---|---|---|---|---|
| K174 | MACROOH + OH | → | MACROO | K166 | |
| K175 | MACRONO$_2$ + OH | → | CH$_3$COCHO + CO + HO$_2$ + NO$_2$ | 4.34 x 10$^{-12}$ | 3 |
| K176 | TERP + OH | → | 0.81 TERPOO + 0.05 ELVOC + 0.14 SVOC | 0.5 * 1.34 x 10$^{-11}$ exp(410/T) + 0.5 * 1.62 x 10$^{-11}$ exp(460/T) | 1, 10 |
| K177 | TERP + NO$_3$ | → | TERPOO + HNO$_3$ | 0.5 * 1.2 x 10$^{-12}$ exp(490/T) + 0.5 * 2.5 x 10$^{-12}$ | 1, 10 |
| K178 | TERP + O$_3$ | → | 0.915 MACR + 0.36 MVK + 0.24 PRV + 1.68 HCHO + 0.16 CO + 0.6 HCOOH + 0.08 C3H6 + 0.68 OH + 0.05 ELVOC + 0.14 SVOC | 0.5 * 8.22 x 10$^{-16}$ exp(-640/T) + 0.5 * 1.39 x 10$^{-15}$ exp(-1280/T) | 1, 10 |
| K179 | TERPOO + HO$_2$ | → | 2 ISOPOOH | K144 | |
| K180 | TERPOO + NO | → | 2 (HCHO + 0.64MVK + 0.36 MACR + HO$_2$) + NO$_2$ | K145 | |
| K181 | TERPOO + NO | → | 2 ISOPONO$_2$ | K146 | |
| K182 | TERPOO + NO$_3$ | → | 2 (HCHO + 0.64MVK + 0.36 MACR + HO$_2$) + NO$_2$ | K147 | |
| K183 | TERPOO + CH$_3$OO | → | 2 (0.64MVK + 0.36MACR + 2HCHO + 2HO$_2$) | K148 | |
| K184 | TERPOO + CH$_3$OO | → | 2 (0.64MVK + 0.36MACR + HCHO + CH$_3$OH) | K149 | |
| K185 | AROM + OH | → | AROMOO + HO$_2$ | A1 * 1.8 x 10$^{-12}$exp(340/T) + A2 * 1.72 x 10$^{-11}$ + A3 * 2.3 x 10$^{-12}$exp(-190/T) | 1, 11 |
| K186 | AROM + NO$_3$ | → | AROMOO + HNO$_3$ | A1 * 7.8 x 10$^{-17}$ + A2 * 3.54 x 10$^{-16}$ | 1, 11 |
| K187 | AROM + O$_3$ | → | AROMOO | A1 * 1.0x 10$^{-21}$ + A2 * (2.4 x 10$^{-13}$exp(-5586/T) + 5.37 x 10$^{-13}$exp(-6039/T) + 1.91 x 10$^{-13}$exp(-5586/T))/3 | 1, 11, 12 |
| K188 | AROMOO + HO$_2$ | → | C$_4$H$_9$OOH + CHOCHO + HCHO | K126 | |
| K189 | AROMOO + NO | → | NO$_2$ + 0.67CH$_3$CH$_2$COCH$_3$ + 0.67 HO$_2$ + 0.33C$_2$H$_5$OO + 0.33CH$_3$CHO + CHOCHO + HCHO | K127 | |
| K190 | AROMOO + NO | → | C$_4$H$_9$ONO$_2$ + CHOCHO + HCHO | K128 | |
| K191 | AROMOO + CH$_3$OO | → | HCHO + HO$_2$ + 0.67(CH$_3$CH$_2$C(O)CH$_3$ + HO$_2$) + 0.33(CH$_3$CHO + CH$_3$CH$_2$OO) + CHOCHO + HCHO | K129 | |
| K192 | AROMOO + CH$_3$OO | → | CH$_3$CH$_2$COCH$_3$ + CH$_3$OH + CHOCHO + HCHO | K130 | |
| K193 | SO$_2$ + OH | → | HO$_2$ + H$_2$SO$_4$ | 3.3 x 10$^{-31}$($T$/300)$^{-4.3}$[N$_2$] 1.6 x 10$^{-12}$ (T/300)$^{-0.7}$ $F$c = 0.6 | 2 |
| K194 | DMS + OH | → | CH$_3$OO + HCHO + SO$_2$ | 1.1 x 10$^{-11}$exp(-240/$T$) | 2 |
| K195 | DMS + OH | → | 0.75 CH$_3$OO + 0.75 HCHO + 0.75 SO$_2$ + 0.25 MSA | 1.0 x 10$^{-39}$[O$_2$] exp(5820/$T$) / (1 + 5.0 x 10$^{-30}$[O$_2$] exp(6280/$T$)) | 2 |
| K196 | DMS + NO$_3$ | → | CH$_3$OO + HCHO + SO$_2$ + HNO$_3$ | 1.9 x 10$^{-13}$exp(520/$T$) | 2 |
| K197 | NH$_3$ + OH | → | NH$_2$ + HO$_2$ | 1.7 x 10$^{-12}$exp(-710/$T$) | 2 |
| K198 | NH$_2$ + O$_2$ | → | NH$_2$O$_2$ | 6.0 x 10$^{-21}$ | 2 |





| K199 | $NH_2 + O_3$ | → | $NH_2O_2$ | $4.3 \times 10^{-12}exp(-930/T)$ | 2 |
|------|------|---|------|------|---|
| K200 | $NH_2 + OH$ | → | $NH_2O_2$ | $3.4 \times 10^{-11}$ | 2 |
| K201 | $NH_2 + HO_2$ | → | $NH_3$ | $3.4 \times 10^{-11}$ | 2 |
| K202 | $NH_2 + NO$ | → | $NH_2O_2 + NO_2$ | $4.0 \times 10^{-12}exp(450/T)$ | 2 |
| K203 | $NH_2 + NO_2$ | → | $NH_2O_2 + NO$ | $2.1 \times 10^{-12}exp(650/T)$ | 2 |
| K204 | $NH_2O_2 + O_3$ | → | $NH_2$ | K199 | |
| K205 | $NH_2O_2 + HO_2$ | → | $NH_2$ | K201 | |
| K206 | $NH_2O_2 + NO$ | → | $NH_2 + NO_2$ | K202 | |

[#] The reaction products $O_2$, $H_2$, and $H_2O$ are not shown.

[1] The chemical kinetic data and mechanistic information was taken from the website of the IUPAC Task Group on Atmospheric Chemical Kinetic Data Evaluation: www.iupac-kinetic.ch.cam.ac.uk

[2] The chemical kinetic data and mechanistic information was taken from the website of the NASA Panel for Data Evaluation (Evaluation No. 18, JPL Publication 15-10) http://jpldataeval.jpl.nasa.gov

[3] The chemistry mechanistic information was taken from the Master Chemical Mechanism (MCM v3.2):
- for non-aromatic schemes: Jenkin et al. (1997); Saunders et al. (2003)
- for the isoprene scheme: Jenkin et al. (2015)
- for aromatic schemes: Jenkin et al. (2003); Bloss et al. (2005)
- and via the website: http://mcm.leeds.ac.uk/MCM

[4] Atkinson (1997):

$R_1 = 2.7 \times 10^{14}exp(-6350/T)$
$R_2 = 6.3 \times 10^{-14}exp(-550/T)$

$f = R_1/(R_1 + R_2 \times [O_2])$

$R_1 = 1.94 \times 10^{-22} [AIR] exp(0.972 \times N_c)$
$R_2 = 0.826 \times (T/300)^{-8.1}$

$A = 1/(1+log_{10}(R_1/R_2)^2)$

$RTC(N_c)P = 0.4 \times R_1/(1+R_1/R_2) 0.411^A$

$RTC(N_c)S = R1/(1+R_1/R_2) 0.411^A$

where, $N_c$ is the number of carbons (i.e., 1-5)

[5] Orlando et al. (1992); Poisson et al. (2000)

[6] Peeters and Müller (2010)

[7] Crounse et al. (2011)

[8] Paulot et al. (2009)

[9] Browne et al. (2014)

[10] Average of α- and β-pinene

[11] A1, A2, A3 represents the relative contributions of *ortho-, meta-,* and *para*-xylene, toluene and benzene (roughly 0.4, 0.6 and 0.4, respectively, for the year 2006)

[12] Average of ortho-, meta- and para-isomers of xylene





**Table 3. Global annual emissions of trace gases used for the MOGUNTIA chemistry scheme in TM5-MP for the year 2006, in Tg yr$^{-1}$ unless specified otherwise.**

| Species | Long name | Emissions | | | | | | |
|---|---|---|---|---|---|---|---|---|
| | | Anthropogenic[&] | Biomass Burning | Biogenic | Soil | Oceanic | Other | Total |
| CO | carbon monoxide | 600.5 | 386.4 | 90.2 | | 19.9 | | 1097 |
| HCHO | formaldehyde | 2.4 | 5.2 | 4.7 | | | | 12.3 |
| HCOOH | formic acid | 4.6 | 1.8 | 3.5 | | | | 9.8 |
| CH$_3$OH | methanol | 4.7 | 9.8 | 131.9 | | | | 146.4 |
| C$_2$H$_6$ | ethane | 6.2 | 3.4 | 0.3 | | 1.0 | | 10.9 |
| C$_2$H$_4$ | ethene | 5.3 | 4.8 | 18.3 | | 1.4 | | 29.8 |
| C$_2$H$_2$ | acetylene | 3.3 | | | | | | 3.3 |
| CH$_3$CHO | acetaldehyde | 1.2 | 4.4 | 21.9 | | | | 27.5 |
| CH$_3$COOH | acetic acid | 4.6 | 18.0 | 3.5 | | | | 26.1 |
| CH$_3$CH$_2$OH | ethanol | 0.5 | 0.1 | 18.6 | | | | 19.3 |
| HOCH$_2$CHO | glycol-aldehyde | 1.4 | 4.3 | | | | | 5.7 |
| CHOCHO | glyoxal | 2.4 | 5.2 | | | | | 7.6 |
| C$_3$H$_8$ | propane | 6.5 | 0.7 | 0.03 | | 1.3 | | 8.5 |
| C$_3$H$_6$ | propene and higher alkenes | 8.3 | 4.8 | 17.5 | | 1.5 | | 32.1 |
| CH$_3$COCH$_3$ | acetone | 2.7 | 1.7 | 37.7 | | | | 42.1 |
| CH$_3$COCHO | methylglyoxal | 1.6 | 3.4 | | | | | 5.0 |
| C$_4$H$_{10}$ | butane and higher alkanes (including butane, pentane, hexane, higher alkanes, and other vocs) | 52.8 | 0.5 | 0.1 | | | | 53.4 |
| CH$_3$CH$_2$COCH$_3$ | methyl-ethyl-ketone (including higher ketones except for acetone) | 1.4 | 1.4 | 0.9 | | | | 3.7 |
| C$_5$H$_8$ | isoprene | | | 579.4 | | | | 579.4 |
| C$_{10}$H$_{16}$ | monoterpenes | | | 97.9 | | | | 97.9 |
| C$_7$H$_8$ | toluene and aromatics (including toluene, xylene benzene, trimethylbenzene and higher aromatics) | 25.3 | 4.0 | 1.5 | | | | 30.8 |





| | | | | | | | |
|---|---|---|---|---|---|---|---|
| $NO_X$ [#] | nitrogen oxides | 42.3 | 6.6 | | 5.0 | | 6.0 [*] | 59.9 |
| $NH_3$ | ammonia | 56.1 | 4.4 | | 2.3 | 8.1 | | 70.9 |
| $SO_2$ | sulfur dioxide | 120.5 | 2.3 | | | | 9.3 [$] | 132.1 |
| $CH_3SCH_3$ | dimethylsulphide | | | 1.7 | | 95.8 | | 97.5 |

[&] including aircraft emissions

[#] in Tg-N yr$^{-1}$

[*] $NO_X$ production from lightning

[$] $SO_2$ from volcanoes





**Table 4. Tropospheric budgets of $O_3$ for the year 2006 in $Tg(O_3)\ yr^{-1}$ and burden in $Tg(O_3)$.**

| Production terms | mCB05 (EBI) | mCB05 (KPP) | MOGUNTIA | Loss terms | mCB05 (EBI) | mCB05 (KPP) | MOGUNTIA |
|---|---|---|---|---|---|---|---|
| Stratospheric inflow* | 632 | 429 | 424 | Deposition | 955 | 932 | 913 |
| Trop. chem. production | 5589 | 5719 | 5709 | Trop. chem. loss | 5192 | 5216 | 5219 |
| Trop. burden | 385 | 384 | 375 | Trop. lifetime (days) | 22.8 | 22.8 | 22.3 |

*sum of the deposition and the tropospheric chemical loss minus the production*

**Table 5. Tropospheric chemical budget of OH for the year 2006 in $Tg(OH)\ yr^{-1}$.**

| Production terms | mCB05 (EBI) | mCB05 (KPP) | MOGUNTIA | Loss terms | mCB05 (EBI) | mCB05 (KPP) | MOGUNTIA |
|---|---|---|---|---|---|---|---|
| $O(^1D) + H_2O$ | 1960 | 1953 | 1878 | $OH + CO$ | 1665 | 1671 | 1775 |
| $NO + HO_2$ | 1268 | 1312 | 1426 | $OH + CH_4$ | 613 | 626 | 644 |
| $O_3 + HO_2$ | 560 | 566 | 561 | $OH + O_3$ | 254 | 260 | 262 |
| $H_2O_2 + h\nu$ | 262 | 265 | 303 | $OH + ISOP$ | 114 | 115 | 120 |
| Other | 203 | 201 | 120 | Other | 1606 | 1626 | 1487 |

**Table 6. Global budgets of CO for the year 2006 in $Tg(CO)\ yr^{-1}$ and burden in $Tg(CO)$.**

| Production terms | mCB05 (EBI) | mCB05 (KPP) | MOGUNTIA | Loss terms | mCB05 (EBI) | mCB05 (KPP) | MOGUNTIA |
|---|---|---|---|---|---|---|---|
| Emissions | 1097 | 1097 | 1097 | Deposition | 98 | 97 | 99 |
| Trop.chem. production | 1809 | 1818 | 1992 | Trop. chem. loss | 2840 | 2849 | 2924 |
| Strat. chem. production | 26 | 26 | 26 | Strat. chem. loss | 87 | 89 | 90 |
| Atmos. burden | 370 | 360 | 361 | Lifetime (days) | 47.5 | 46.2 | 43.6 |





**Figures**

**Figure 1: Simulated annual mean surface (left columns) and zonal mean (right columns) O₃ mixing ratios (ppb) for the MOGUNTIA chemistry scheme for the year 2006 (a,b), and the respective differences compared to mCB05(KPP) (c,d); the surface and zonal mean absolute differences between mCB05(KPP) and mCB05(EBI) are also presented (e,f).**

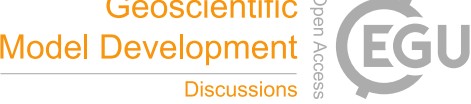

**Figure 2: Simulated annual mean surface (left columns) and zonal mean (right columns) CO mixing ratios (ppb) for the MOGUNTIA chemistry scheme for the year 2006 (a,b), and the respective differences compared to mCB05(KPP) (c,d); the surface and zonal mean absolute differences between mCB05(KPP) and mCB05(EBI) are also presented (e,f).**



## Trop. NO2 columns / OMI / 2006

MOGUNTIA          mCB05(KPP)

a) **Model (w/ AK)**      b) **Model (w/ AK)**

trop. NO2 column [1e+14 molec cm-2]

c) **Satellite (orig.)**      d) **Satellite (orig.)**

trop. NO2 column [1e+14 molec cm-2]

e) **abs. diff. model − satellite**      f) **abs. diff. model − satellite**

abs. diff. model − satellite [1e+14 molec cm-2]

g) **rel. diff. model − satellite**      h) **rel. diff. model − satellite**

rel. diff. model − satellite [%]

**Figure 3: Annual mean comparison of tropospheric NO$_2$ vertical columns (molecules cm$^{-2}$) for the two chemistry schemes MOGUNTIA and mCB05(KPP) (a,b), against the Ozone Monitoring Instrument (OMI) satellite data (c,d), using the respective averaging kernel information for 2006. The absolute (e,f) and relative (g,h) differences are also presented.**





Figure 4: Zonal mean OH mixing ratios for December-January-Fabruary (DJF; left) and June-July-August (JJA; right) 2006, as simulated by the TM5-MP model with the MOGUNTIA chemistry scheme (top), the differences (%) between the mCB05(KPP) and the MOGUNTIA chemical configuration (middle), and the optimized climatological average from Spivakovsky et al. (2000), up to
5    200 hPa (bottom).





**Figure 5: Monthly mean comparison of TM5-MP surface O$_3$ (ppb) against surface observations (black line) from EMEP and WOUDC databases for the two chemistry schemes, mCB05(KPP) (green line) and MOGUNTIA (blue line), using co-located model output for 2006 sampled at the measurement times; error bars indicate the standard deviation in the monthly means. For**
5 **comparison, model results of the mCB05 with the EBI solver (red line) are also presented.**







**Figure 6: Monthly mean comparison of TM5-MP O$_3$ (ppb) against sonde observations (black dots, mean and standard deviation) at a) Hohenpeissenberg and b) Macquarie Island, for different pressure levels (900; 800; 500; 400; 200 hPa) for the two chemistry schemes, mCB05(KPP) (green line) and MOGUNTIA (blue line), using co-located model output for 2006 sampled at the measurement times; error bars indicate the standard deviation in the monthly means. For comparison, the results of mCB05 with the EBI solver (red line) are also presented.**







Figure 7: **Monthly mean comparison of TM5-MP surface CO (ppb) against flask measurements (black line) for the two chemistry schemes, mCB05(KPP) (green line) and MOGUNTIA (blue line), using co-located model output for 2006 sampled at the measurement times; error bars indicate the standard deviation in the monthly means. For comparison, model results of the mCB05 with the EBI solver (red line) are also presented.**





**Figure 8: Annual mean comparison of total CO vertical columns (molecules cm⁻²) for the two chemistry schemes of TM5-MP, MOGUNTIA and mCB05(KPP) (a,b), against MOPITT satellite data (c,d), using the respective averaging kernel information for 2006. The absolute (e,f) and relative (g, h) differences are also presented.**





**Figure 9: Monthly mean comparison of TM5-MP surface $C_2H_6$ (left column) and $C_3H_8$ (right colimn) against flask measurements (black dots) in ppt for the two chemistry schemes, mCB05(KPP) (green line) and MOGUNTIA (blue line), using co-located model output for 2006 sampled at the measurement times; error bars indicate the standard deviation in the monthly means. For comparison, model results of the mCB05 with the EBI solver (red line) are also presented.**





**Figure 10:** Comparison of TM5-MP vertical profiles (in km) of C$_2$H$_6$ (left column) and C$_3$H$_8$ (right column) against aircraft observations (black line) in ppt, for the two chemistry schemes, mCB05(KPP) (green line) and MOGUNTIA (blue line), using co-located model output for 2006 sampled at the measurement times; error bars indicate the standard deviation. For comparison, model results of the mCB05 with the EBI solver (red line) are also presented. The numbers on the right vertical axis indicate the number of available measurements.



**Figure 11: Comparison of TM5-MP vertical profiles (in km) of C$_2$H$_4$ (left column) and C$_3$H$_6$ (right column) against aircraft observations (black line) in ppt, for the two chemistry schemes, mCB05(KPP) (green line) and MOGUNTIA (blue line), using co-located model output for 2006 sampled at the measurement times; error bars indicate the standard deviation. For comparison, model results of the mCB05 with the EBI solver (red line) are also presented. The numbers on the right vertical axis indicate the number of available measurements.**