# Peer review of "Description and evaluation of a detailed gas-phase chemistry scheme in the TM5-MP global chemistry transport model (r112)"

_Geoscientific Model Development, 2020_

## Referee Comment (RC1) · Anonymous Referee #1 · 27 May 2020

This work reports on the implementation of an extended VOC chemical mechanism and the Rosenbrock solver into the global TM5 model. In addition, the model is evaluated with respect to the performance of the original, the new integrator, and the new VOC mechanism. Observational comparisons are used to quantify the performance of the new model advances.

Even though both concepts, the use of the Rosenbrock solver in global models and the implemented VOC chemistry, are not novel, the advancement for the TM5 community is still considerable. However, I have substantial concerns regarding the model

description, the model analysis, and the presentation of the complete manuscript in general.

As such, I believe that major revisions are necessary before this manuscript is ready for publication in GMD. In the following, I present first the major concerns, followed by specific comments and technical corrections.

General comments:

I have multiple concerns on how the model is described within this manuscript. Very little general information on the TM5 model is provided, except for a long list of citations. For a non TM5 community member it is impossible to understand the key features of this model without opening another publication. A general description of the model needs to be provided, especially since many discrepancies in the model comparison are attributed to transport processes. A summary on how transport processes are simulated needs to be added. An additional evaluation of these transport processes would be useful to justify the later claims. Additionally, some information that should be included in the model description can be found in later sections (e.g. how the tropopause altitude is calculated between the different simulations). The manuscript should be harmonised such that all these information are included in the model description.

Within this study, two different chemical mechanisms are used but the manuscript only includes information on the newly developed one. A short description on the "standard" TM5 mechanism should be included and a list of all reactions of this mechanism needs to be added to the supplemental material. A box model comparison of all mechanisms (i.e. MOGUNTIA, CB05 and MCM) would be useful to understand the mechanistic

differences. Within the text it becomes evident that different emission data sets are used for the different mechanisms. However, this information is not at all included in Section 2.4. The emissions for the standard mechanism need to be provided (e.g. table in supplemental material).

Scientifically, many claims on what causes the differences between the model and the observations are not supported by the provided data and not enough evidence is given. In one particular case, too low upward transport is given as a reason and one page afterwards it is claimed that the model simulates a too high transport in the same region. The manuscript therefore needs to be checked if the claims are supported by the results. If so, more justification must be provided (e.g. presenting differences in $O_3$ precursors). Otherwise these statements should be removed.

All in all, the model tends to underestimate VOCs, which is mainly attributed to too low emission sources. Higher emission strengths of VOCs will lead to higher VOC concentrations in low-NOx regime, influencing the $O_3$ production. I therefore strongly suggest to perform a sensitivity simulation with up-scaled emission sources to investigate the impact on $O_3$ and $HO_x$.

Another major concern I have is the overuse of citations when referring to earlier work. A good example is page 4 line 17-20: This sentence has 12 citations but only 18 words with providing no important information about the model at all. It feels as if every paper that used the model is cited here (without evidence why this is necessary), which should not be the goal of the model description. It should be sufficient to cite e.g. Huijnen et al., 2010 since they focus on the chemical modelling in TM5. The same holds when referring to earlier studies using parts of the mechanism (e.g. page 6 line 6-7, page 6 line 32, page 7 line 3-4), especially if they are not further used in the manuscript. It would be scientifically more profound to only cite publications, in

which the approach was novel or were it was used first and not every publication using this part of the mechanism or model development. I therefore strongly advise you to recheck every citation in the manuscript and limit citations to a minimum.

Last but not least, when reading the manuscript it does not feel like a coherent story and each section feels like an isolated section. Additionally, the manuscript suffers from grammatical mistakes. I therefore suggest sweeping through the document focusing on simpler sentence structures.

Specific comments:

Page 1:
Line 31-33: Not much information is given about other global models in your manuscript. Therefore, you should only focus this statement on TM5.

Page 4:
Line 28-29: What influence does this approach have on the stratospheric-tropospheric exchange in your budget analysis?

Page 5:
Line 4-5: When using 150 ppb as definition, the tropopause altitude will differ when using different chemical mechanisms or integrators. Do you use the same tropopause altitude for each simulation? And if so, on which simulation is this definition based? Is the tropopause altitude calculated for each time step or is it based on mean data? What impact do you expect from this?
Line 7: The only $O_3$ chemical aqueous-phase sink considered here is $SO_2$. However, the major aqueous-phase sink of $O_3$ is the reaction with $O_2^-$ (Liang and Jacob, 1997).

By not taking this sink into account, what impact do you expect this has on the $O_3$ budget and the $O_3$ burden in your analysis?

Page 7:
Line 13-15: Due to the citation style used, it is not at all obvious in which publication each of the advances have been published.
Line 26: How are meteorological conditions simulated in TM5? This needs to be discussed in the general description of the model (Section 2.1).

Page 8:
Line 23-26: This information is useful to understand why KPP was implemented into TM5. I would suggest you mention this first (i.e. page 8 line 8 and in the introduction).

Page 10:
Line 12: What complexity has the chemical mechanism used for mCB05? Provide more information about this mechanism.

Page 11:
Line 1-15: How is this model performance analysis performed (e.g. which software)? What are the expected limitations?
Line 2-4: This information should be included in Section 2.5.
Line 8-9: The transport of tracers seems to be important for the model performance. How is it decided which tracer is transported and which not? This should be discussed in the model/mechanism description.

Page 12:
Line 7-9: This is not clear. Why is the chemical destruction higher due to changes in

the $O_3$ precursors?

Line 8-9: How do the changes in the $O_3$ precursors look like? This is a nice example were a statement is given without providing any results or argument why this must be the case (see general comments).

Line 12: Why is it necessary to used $NO_y$ mass fixing when using EBI? This needs to be discussed in the model description since this is a major difference between EBI and KPP!

Line 19: This is unclear. By referring to table 3 it implies that different emission datasets are used for the different simulations. If so, why is that the case? This needs to be elaborated in Section 2.4.

Page 13:

Line 4: With the 150 ppb definition your simulation are already up to 15% higher. How does your model compare to Lamarque et al. (2012) when using 100 ppb as tropopause definition? It would be best to provide both budgets (i.e. in Table 4) for the 100 and 150 ppb definition to allow a fair comparison.

Line 13: It is not at all clear in Section 2.4 that different emissions are used. What is the impact of using different emissions?

Line 30-31: This is a good argument for the model description to justify why this approach is used.

Page 14:

Line 4: The contribution of the "other reactions" changes from about 200 to 120 Tg/yr. What causes these changes and what is included in this category?

Line 9-10: This should be mentioned in the model description.

Line 12: Which tropopause definition did van Noije et al. (2014) use?

Line 27: The difference is about 15%, so using "somewhat shorter" is a slight under-estimation.

Line 34: What lifetime do you get when using 100 ppb as tropopause definition?

Page 15:
Line 9: To what else can these differences be attributed to?

Page 17:
Line 18-19: This is a bit confusing. The dataset used to compare 2006 is published in 2000? What are the limitations of this comparison when using different years?

Page 18:
Line 21-22: Due to the lack of specific details on mCB05 in the manuscript, it is impossible to identify why this must be the case. More details are necessary here.
Line 24: Provide more details on how NOx reservoir species differ in their concentration and spatial distribution between both mechanisms.

Page 19:
Line 2-3: How well does your model compare when using 7.9Tg-N/yr?
Line 14-17: Provide evidence why this is the case.

Page 20:
Line 3: What about comparing your model simulations to satellite observations of $O_3$ (e.g. OMI)?
Line 18-20: The surface ozone bias is lowest for mCB05(KPP) but at the same time the ozone burden is higher than for MOGUNTIA. What causes this difference? Are there significant differences in free tropospheric ozone?

Page 21:
Line 10-11: This conclusion is not obvious based on the results you provided. Further analysis is needed here. How well are transport processes modelled in TM5?
Line 15-18: Are these speculations or do you have evidence that this must be the case? If so provide further details.
Line 32: This statement is unclear. The current sentence structure implies that the emissions in the SH are lower when using KPP.

Page 22:
Line 21-23: Earlier (i.e. page 21, line 10-11) you state that the convective uplift is too low but now you state that it is too strong. Which is correct? The presented data do not support either. More evidence is needed. I strongly suggest you to perform an elaborated analysis of the performance of TM5 with respect to transport processes, to justify these claims.
Line 24-25: Have you analysed biomass burning hotspots to support this claim?

Page 23:
Line 2: What causes the opposite annual cycle?
Line 9: Your model underestimates propane but you use a lower emission than other studies. How does your model compare when you use higher emissions?

Page 24:
Line 1-2: Could this underestimation be related to underestimated transport processes (see Page 21 & 22)?
Line 20: What needs to be done to account for the "secondary production from VOC oxidations"?

[Figure]

Page 25:
Line 30-33: Can you provide some suggestions on how to improve these uncertainties?

Page 53:
Table 4: What about $O_3$ scavenging?

Technical corrections:

Page 2, line 4-20: A graphical illustration of the $NO_x$-VOC-$O_3$ relation would be helpful here.
Page 4, line 22 & Page 14, line 1-2 & Page 18, line 11-13: Check gramma and wording.
Page 5, line 3: The statement that this study focuses on the troposphere is stated multiple times. Do not use double statements, to improve the reading flow.
Page 6, line 1: This should be Section 2.2.
Page 9, line 10-14: Listing all species greatly disturbs the reading flow. I would remove this listing and just refer to Table 3 instead.
Page 10, line 12-28: A table summarizing all simulations performed could be useful.
Page 12, line 22-26: This is a rather complicated sentence. Consider using simpler language (i.e. multiple short sentences).
Page 14, line 26: The word "arrive" should not be used here.
Page 17, line 2-19: Presenting the different observations and possible comparisons in a table would be more efficient.
Page 17, line 25: In order to improve the reading flow, it would be best to first compare each tracer discussed in Section 4 (in the same order).
Page 20, line 25: Is the reference to the introduction correct?
Page 20, line 27: Please be more specific and refer to Section 2.1.
Page 41-51: Most of the information presented in Tables 1, 2 and even 3 are well

documented elsewhere. Thus, I strongly recommend you to move these tables to the supplemental material.

Supplement, page 3: Table S3 (including caption) cannot be read completely.

References:

Liang, J. and Jacob, D. J.: Effect of aqueous phase cloud chemistry on tropospheric ozone, Journal of Geophysical Research: Atmospheres, 102, 5993–6001, https://doi.org/10.1029/96JD02957, 1997.

————————————————————

---

## Referee Comment (RC2) · Anonymous Referee #2 · 9 Jun 2020

The manuscript describes the implementation of the MOGUNTIA chemical mechanism, using KPP and Rosenbrock solver in the TM5 model and evaluates the improvements and differences with respect to the earlier versions of the model. The definition of the new mechanism is extensively described but the earlier versions of the model is mainly referring to older publications. I think these versions used in the manuscript could be described a little more to address the differences later. I find the manuscript easy to follow and an adequate level of English, which can be improved during the review. I have some general and specific comments which could be addressed in order to have the manuscript suitable to be published at GMD.

General Comments

Additional analyses can be performed with regards to the transport of tracers as it is frequently used in the manuscript to explain differences. How good is the model with respect to transport, especially vertical transport?

Specific Comments

Page 4, lines 4-5: Use of 150 ppb, or any concentration level has caveats, e.g. model bias. Why not use the meteorological tropopause instead? The implications should be addressed.

Page 13, line 13. Use of different emissions are not clearly mentioned in section 2.4. Authors should justify the use of different emissions and how this impacts the changes they see in the different scenarios.

Page 20, line 3: It would be great if the results are compared with satellites.

---

## Author Comment (AC2) · 17 Jul 2020

We thank the reviewer for the careful reading of the manuscript and the insightful comments. Please find bellow our point-by-point replies:

**General comments:**

**GC1.**     Additional analyses can be performed with regards to the transport of tracers as it is frequently used in the manuscript to explain differences. How good is the model with respect to transport, especially vertical transport?

- The transport of TM5 has been successfully evaluated many times in the past, e.g., see (Koffi et al., 2016; Krol et al., 2005; Peters et al., 2004; Williams et al., 2017). For this, we consider such analysis outside the scope of the current paper that is focused on presenting the new chemistry developments. Note that the current version of the TM5 model was recently included in a model intercomparison (Krol et al., 2018), in which vertical resolution was specifically addressed. For this, we provide references for each major release of the model that can guide the reader for further reading.

Following, however, the reviewer's comment, brief description and references of the transport processes parameterizations in TM5 are added in Model Description (Sect. 2.1): "*The advection scheme used in TM5 is based on the slopes scheme (Russell and Lerner, 1981) and the deep and shallow cumulus convection scheme is parameterized according to Tiedtke (1989). The performance of the transport in the model has been evaluated by (Peters et al., 2004) using sulphur hexafluoride simulations and by analyzing the vertical and horizontal distribution of radon ($_{222}Rn$) to simulate the boundary layer dynamics (Koffi et al., 2016; Williams et al., 2017). More recently, global transport features, such as the transport times associated with inter-hemispheric transport, vertical mixing in the troposphere, transport to and in the stratosphere, and transport of air masses between land and ocean, were evaluated via an inter-comparison of six global transport models (Krol et al., 2018).*"

**Specific Comments:**

**SC1.** Page 4, lines 4-5: Use of 150 ppb, or any concentration level has caveats, e.g. model bias. Why not use the meteorological tropopause instead? The implications should be addressed. this?

- For this work, as we stated in the manuscript, we use the chemical tropopause level defined by a 150 ppb $O_3$ mixing ratio following the well-documented model intercomparison study by Stevenson et al. (2006). The use of the 150 ppb $O_3$ level has been used so far in numerous studies, as also with previous versions of the TM5 model, providing thus an opportunity of a direct comparison of model results with other estimates. On the other hand, the tropopause levels in a model may have various definitions, such as the temperature and the potential vorticity gradients, the altitude or the standard World Meteorological Organization definition that the lowest level above 500 hPa where the vertical temperature gradient decreases to less than or equal 2 $_o$C km$_{-1}$.

  We agree with the reviewer that the definition of the tropopause may lead to great differences, and for this, we stated in the manuscript that the tropopause definition should always be reported when comparing modelling estimates.

  For this work, however, we prefer to keep the tropopause based on the 150 ppb $O_3$ mixing ratio since we here mostly focused on the differences between the different configurations of the model. However, to show the impact of the use of different tropopause levels on the calculated tropospheric budgets, we now provide the relative differences of using the 100 ppb $O_3$ level, i.e.:

**Table 1. Tropospheric budgets of $O_3$ for the year 2006 in Tg($O_3$) yr$_{-1}$ and burden in Tg($O_3$), using the 150 ppb $O_3$ mixing ratio to define tropopause level. In parenthesis the relative differences using the 100 ppb $O_3$ mixing ratios are also presented, calculated by reference to the 150 ppb $O_3$ definition of tropopause level.**

| Production terms | mCB05 (EBI) | | mCB05 (KPP) | | MOGUNTIA | | Loss terms | mCB05 (EBI) | | mCB05 (KPP) | | MOGUNTIA | |
|---|---|---|---|---|---|---|---|---|---|---|---|---|---|
| Stratospheric inflow* | 632 | (10%) | 429 | (32%) | 424 | (30%) | Deposition | 955 | (0%) | 932 | (0%) | 913 | (0%) |
| Trop. chem. production | 5589 | (-3%) | 5719 | (-3%) | 5709 | (-3%) | Trop. chem. loss | 5192 | (-1%) | 5216 | (-1%) | 5219 | (-1%) |
| Trop. burden | 385 | (-8%) | 384 | (-8%) | 375 | (-8%) | Trop. lifetime (days) | 22.8 | (-8%) | 22.8 | (-8%) | 22.3 | (-6%) |

*sum of the deposition and the tropospheric chemical loss minus the production*

**Table 2. Tropospheric chemical budget of OH for the year 2006 in Tg(OH) yr$^{-1}$, using the 150 ppb $O_3$ mixing ratio to define tropopause level. In parenthesis the relative differences using the 100 ppb $O_3$ mixing ratios are also presented, calculated by reference to the 150 ppb $O_3$ definition of tropopause level.**

| Production terms | mCB05 (EBI) | | mCB05 (KPP) | | MOGUNTIA | | Loss terms | mCB05 (EBI) | | mCB05 (KPP) | | MOGUNTIA | |
|---|---|---|---|---|---|---|---|---|---|---|---|---|---|
| $O(_1D) + H_2O$ | 1960 | (0%) | 1953 | (0%) | 1878 | (0%) | $OH + CO$ | 1665 | (-2%) | 1671 | (-2%) | 1775 | (-2%) |
| $NO + HO_2$ | 1268 | (-4%) | 1312 | (-4%) | 1426 | (-4%) | $OH + CH_4$ | 613 | (0%) | 626 | (0%) | 644 | (-1%) |
| $O_3 + HO_2$ | 560 | (-1%) | 566 | (-1%) | 561 | (-1%) | $OH + O_3$ | 254 | (-2%) | 260 | (-2%) | 262 | (-3%) |
| $H_2O_2 + h\nu$ | 262 | (-1%) | 265 | (-1%) | 303 | (-1%) | $OH + ISOP$ | 114 | (-1%) | 115 | (-1%) | 120 | (0%) |
| Other | 203 | (-2%) | 201 | (-2%) | 120 | (-1%) | Other | 1606 | (-1%) | 1626 | (-1%) | 1487 | (-1%) |

**Table 3. Global budgets of CO for the year 2006 in Tg(CO) yr$^{-1}$ and burden in Tg(CO), using the 150 ppb $O_3$ mixing ratio to define tropopause level. In parenthesis the relative differences using the 100 ppb $O_3$ mixing ratios are also presented, calculated by reference to the 150 ppb $O_3$ definition of tropopause level.**

| Production terms | mCB05 (EBI) | | mCB05 (KPP) | | MOGUNTIA | | Loss terms | mCB05 (EBI) | | mCB05 (KPP) | | MOGUNTIA | |
|---|---|---|---|---|---|---|---|---|---|---|---|---|---|
| Emissions | 1097 | (0%) | 1097 | (0%) | 1097 | (0%) | Deposition | 98 | (0%) | 97 | (0%) | 99 | (0%) |
| Trop. chem. production | 1809 | (-1%) | 1818 | (-1%) | 1992 | (-1%) | Trop. chem. loss | 2840 | (-6%) | 2849 | (-6%) | 2924 | (-2%) |
| Strat. chem. production | 26 | (69%) | 26 | (73%) | 26 | (65%) | Strat. chem. loss | 87 | (68%) | 89 | (69%) | 90 | (68%) |
| Atmos. burden | 370 | (0%) | 360 | (0%) | 361 | (0%) | Lifetime (days) | 47.5 | (2%) | 46.2 | (2%) | 43.6 | (3%) |

**SC2.**    Page 13, line 13. Use of different emissions are not clearly mentioned in section 2.4. Authors should justify the use of different emissions and how this impacts the changes they see in the different scenarios.

- As explained in our replies to the other reviewer (RC1), we use the same emissions (and boundary conditions) for the different chemistry configurations of the model. This choice is made in order to specifically focus only on their differences between the two mechanisms in the model as explicitly presented in Sect. 3. In the manuscript we refer to the different "speciation" of the emitted volatile organic compounds (VOC) i.e. how the VOC emissions are distributed among the VOC species considered in the different chemical mechanisms: the more lumped mCB05 does not resolve all of the NMVOCs provided by the emission datasets, whereas MOGUNTIA explicitly simulates the NMVOCs (C1-4) and isoprene. To make this point clearer, however, we changed the word "speciation" with "*representation*" when we refer here to the differences between the two chemical schemes (see also our reply to SC17) and we clearly state in the manuscript that both mechanisms use the same emission datasets.
-

**SC3.**   Page 20, Line 3: It would be great if the results are compared with satellites

- For this work we used two extended surface ozone observation databases and one ozonesonde database to evaluate the model and discuss the differences of the different configurations. More extended model evaluation, although always interesting, is not however expected to change the conclusions of this work, especially for the simulated tropospheric ozone mixing ratios. On the other hand, as also we refer in the summary (Sect. 6) a more dedicated comparison of the model with the MOGUNTIA configuration with *in-situ* observations and satellite retrievals is planned to be performed in the future. As an example of our work in progress, the reviewer can find bellow an evaluation of tropospheric $O_3$ columns (for the three configurations of this study) with the respective OMI monthly tropospheric retrievals:

[Figure]

[Figure]

[Figure]

[Figure]

Overall, it is obvious from this evaluation, that the MOGUNTIA scheme simulates better the OMI retrievals, thus leading the model in the right direction. Note, again, that we choose not to present this evaluation in this paper, since a separate paper is in progress.

**References:**

Koffi, E. N., Bergamaschi, P., Karstens, U., Krol, M., Segers, A., Schmidt, M., Levin, I., Vermeulen, A. T., Fisher, R. E., Kazan, V., Klein Baltink, H., Lowry, D., Manca, G., Meijer, H. A. J., Moncrieff, J., Pal, S., Ramonet, M., Scheeren, H. A. and Williams, A. G.: Evaluation of the boundary layer dynamics of the TM5 model over Europe, Geosci. Model Dev., 9(9), 3137–3160, doi:10.5194/gmd-9-3137-2016, 2016.

Krol, M., Houweling, S., Bregman, B., van den Broek, M., Segers, A., van Velthoven, P., Peters, W., Dentener, F. and Bergamaschi, P.: The two-way nested global chemistry-transport zoom model TM5: algorithm and applications, Atmos. Chem. Phys., 5(2), 417–432, doi:10.5194/acp-5-417-2005, 2005.

Krol, M., de Bruine, M., Killaars, L., Ouwersloot, H., Pozzer, A., Yin, Y., Chevallier, F., Bousquet, P., Patra, P., Belikov, D., Maksyutov, S., Dhomse, S., Feng, W. and Chipperfield, M. P.: Age of air as a diagnostic for transport timescales in global models, Geosci. Model Dev., 11(8), 3109–3130, doi:10.5194/gmd-11-3109-2018, 2018.

Peters, W., Krol, M. C., Dlugokencky, E. J., Dentener, F. J., Bergamaschi, P., Dutton, G., Velthoven, P. v., Miller, J. B., Bruhwiler, L. and Tans, P. P.: Toward regional-scale modeling using the two-way nested global model TM5: Characterization of transport using SF 6, J. Geophys. Res., 109(D19), D19314, doi:10.1029/2004JD005020, 2004.

Russell, G. L. and Lerner, J. A.: A New Finite-Differencing Scheme for the Tracer Transport Equation, J. Appl. Meteorol., 20(12), 1483–1498, doi:10.1175/1520-0450(1981)020<1483:ANFDSF>2.0.CO;2, 1981.

Stevenson, D. S., Dentener, F. J., Schultz, M. G., Ellingsen, K., van Noije, T. P. C., Wild, O., Zeng, G., Amann, M., Atherton, C. S., Bell, N., Bergmann, D. J., Bey, I., Butler, T., Cofala, J., Collins, W. J., Derwent, R. G., Doherty, R. M., Drevet, J., Eskes, H. J., Fiore, A. M., Gauss, M., Hauglustaine, D. A., Horowitz, L. W., Isaksen, I. S. A., Krol, M. C., Lamarque, J.-F., Lawrence, M. G., Montanaro, V., Müller, J.-F., Pitari, G., Prather, M. J., Pyle, J. A., Rast, S., Rodriguez, J. M., Sanderson, M. G., Savage, N. H., Shindell, D. T., Strahan, S. E., Sudo, K. and Szopa, S.: Multimodel ensemble simulations of present-day and near-future tropospheric ozone, J. Geophys. Res., 111(D8), D08301, doi:10.1029/2005JD006338, 2006.

Tiedtke, M.: A Comprehensive Mass Flux Scheme for Cumulus Parameterization in Large-Scale Models, Mon. Weather Rev., 117(8), 1779–1800, doi:10.1175/1520-0493(1989)117<1779:ACMFSF>2.0.CO;2, 1989.

Williams, J. E., van Velthoven, P. F. J. and Brenninkmeijer, C. A. M.: Quantifying the uncertainty in simulating global tropospheric composition due to the variability in global emission estimates of Biogenic Volatile Organic Compounds, Atmos. Chem. Phys., 13(5), 2857–2891, doi:10.5194/acp-13-2857-2013, 2013.

Williams, J. E., Boersma, K. F., Le Sager, P. and Verstraeten, W. W.: The high-resolution version of TM5-MP for optimized satellite retrievals: description and validation, Geosci. Model Dev., 10(2), 721–750, doi:10.5194/gmd-10-721-2017, 2017.

---

## Author Response (AR1)

We thank the reviewer for the careful reading of the manuscript and the insightful comments. Please find bellow our point-by-point replies:

**General comments:**

GC1.       I have multiple concerns on how the model is described within this manuscript. Very little general information on the TM5 model is provided, except for a long list of citations. For a non TM5 community member it is impossible to understand the key features of this model without opening another publication. A general description of the model needs to be provided, especially since many discrepancies in the model comparison are attributed to transport processes. A summary on how transport processes are simulated needs to be added. An additional evaluation of these transport processes would be useful to justify the later claims.

- Indeed, our point is not to present the whole model, nor to reevaluate each part of it. This has been already presented in detail in numerous publications. Instead, our focus here is to present the new chemistry developments as stated in Sect.1. The model and specifically the transport of TM5 has been successfully evaluated in the past, e.g., see (Koffi et al., 2016; Krol et al., 2005; Peters et al., 2004; Williams et al., 2017). For this, we provide references for each major release of the model that can guide the reader for further reading. Following the reviewer's comment, however, a statement on the reference of the transport processes in TM5 is added in Model Description (Sect. 2.1): *"The advection scheme used in TM5 is based on the slopes scheme (Russell and Lerner, 1981) and the deep and shallow cumulus convection scheme is parameterized according to Tiedtke (1989). The performance of the transport in the model has been evaluated by (Peters et al., 2004) using sulphur hexafluoride simulations and by analyzing the vertical and horizontal distribution of radon (222Rn) to simulate the boundary layer dynamics (Koffi et al., 2016; Williams et al., 2017). More recently, global transport features, such as the transport times associated with inter-hemispheric transport, vertical mixing in the troposphere, transport to and in the stratosphere, and transport of air masses between land and ocean, were evaluated via an inter-comparison of six global transport models (Krol et al., 2018)."*

GC2.       Additionally, some information that should be included in the model description can be found in later sections (e.g. how the tropopause altitude is calculated between the different simulations). The manuscript should be harmonised such that all this information is included in the model description.

- All information related to model description has been moved to the model description as suggested by the reviewer in the specific and technical comments (please see also our replies to respective comments).

GC3.       Within this study, two different chemical mechanisms are used but the manuscript only includes information on the newly developed one. A short description on the "standard" TM5 mechanism should be included and a list of all reactions of this mechanism needs to be added to the supplemental material. A box model comparison of all mechanisms (i.e. MOGUNTIA, CB05 and MCM) would be useful to understand the mechanistic differences.

- CB05 is a well-established mechanism that already presented in numerous publications. Specifically, the modified version of the CB05 mechanism used in the standard configuration of the model (i.e., mCB05) is already described in several publications of the TM5 community, such as the publications by (Williams et al., 2013, 2017); the full table of reactions is freely available for the reader, i.e., see Table A1 and A2 there, https://www.atmos-chem-phys.net/13/2857/2013/;. For this, we believe that it is needless to repeat here the same tables. However, to make it more clear we now state that for the mechanism we refer to *"Williams et al. (2013), along with updates presented in Williams et al. (2017)."*

- We present bellow an example of the box model comparison for $O_3$, CO, NOy ($=NO+NO_2+NO_3+2*N_2O_5+HNO_4$) and OH, between the mCB05 and MOGUNTIA mechanisms, using the KPP files from the TM5-MP model of this study. Note that, to our knowledge, MCM does not exist in a KPP format (e.g., see Sommariva et al., 2020), and our comparison is therefore limited to the comparison of mCB05 and MOGUNTIA.
  - Initial conditions:
    - $O_3$: 40 ppb
    - $HO_2$: 1ppb
    - $H_2O_2$: 1 ppb
    - OH: 0.003 ppb
    - NO: 0.6 ppb
    - $NO_2$: 1.5 ppb
    - $NO_3$: $9 \times 10_{-7}$
    - $N_2O_5$: $4 \times 10_{-9}$ ppb
    - HCHO: 0.5 ppb
    - $CH_3O_2$: 0.025 ppb
    - $CH_3O_2H$: 5 ppb
    - $CH_4$: 1700 ppb
    - CO: 150 ppb
    - HCOOH: 0.1 ppb
    - ISOPRENE: 0.1 ppb
    - Temperature: 298.15 K
    - Pressure: 1023 hPa
    - Relative humidity: 45%
    - Emissions: None
    - Deposition: None
  - Photolysis rates; represent equator, noontime, in $s_{-1}$ based on (Lim et al., 2005) box modelling study. Note that a prescribed diurnal cycle of radiation is applied.
    - $JO_3$ = 1.36E-5
    - $JNO_2$ = 4.65E-3
    - $JH_2O_2$ = 7.65E-6
    - $JNO_3a$ = 1.10E-1
    - $JNO_3b$ = 1.30E-1
    - JHONO = 3.05E-3
    - $JHNO_3$ = 2.69E-7
    - $JHNO_4$ = $JHNO_3$
    - $JN_2O_5$ = 2.54E-5
    - $JCH_2Oa$ = 2.54E-5
    - $JCH_2Ob$ = 1.31E-5
    - $JCH_3O_2H$ = 3.63E-6
    - JPAN = 1.47E-6
    - JORGNTR = 1.47E-6
    - JALD = 6.71E-6
    - JGLYa = 6.82E-5
    - JGLYb = 7.08E-5
    - JGLYAL = 1.30E-5
    - JMGLY = 2.02E-4
    - JACETONE = 1.40E-6

    For all organic hydroperoxides the photolysis rate of $CH_3O_2H$ is used.

    For all organic nitrates, the photolysis rates of the lumped species (ORGNTR) is used

[Figure]

Although we studied both mechanisms in detail in box models to understand the differences, we feel that a box-model addition to the paper would be of limited value. Reasons are the heterogeneous conditions that are encountered in the atmosphere in terms of emissions, radiation, and temperature.

**GC4.** Within the text, it becomes evident that different emission data sets are used for the different mechanisms. However, this information is not at all included in Section 2.4. The emissions for the standard mechanism need to be provided (e.g. table in the supplementary material).

- We could not find evidence for this in the text. Both mechanisms use the same emission data sets and boundary conditions (see Sect. 2). This choice is made in order to specifically focus only on the differences between the two mechanisms in the model as explicitly presented in Sect. 3. The only difference is on how the two mechanisms distribute the VOC emissions to the species considered in the mechanisms: the more lumped mCB05 does not resolve all of the NMVOCs provided by the emission datasets, whereas MOGUNTIA explicitly simulates the NMVOCs (C1-3) and isoprene.
  To make this point clearer, however, we changed the word "speciation" with "*representation*" when we refer here to the differences between the two chemical schemes (see also our reply to SC17) and we now clearly state in the manuscript that both mechanisms use the same emission datasets.

**GC5.** Scientifically, many claims on what causes the differences between the model and the observations are not supported by the provided data and not enough evidence is given. In one particular case, too low upward transport is given as a reason and one page afterwards it is claimed that the model simulates a too high transport in the same region. The manuscript therefore needs to be checked if the claims are supported by the results. If so, more justification must be provided (e.g. presenting differences in $O_3$ precursors). Otherwise these statements should be removed.

- We thank the reviewer for attracting our attention to this issue. Particularly, we removed the sentence: "The negative model bias in the tropical UTLS points at a weak convective uplift in tropical Africa in April." from the discussion of ozone comparison with the

MOZAIC data. The discussion of both the $O_3$ and CO evaluation with the MOZAIC observations is now rewritten in the manuscript (see our reply in SC35).

**GC6.** All in all, the model tends to underestimate VOCs, which is mainly attributed to too low emission sources. Higher emission strengths of VOCs will lead to higher VOC concentrations in low-NOx regime, influencing the $O_3$ production. I therefore strongly suggest to perform a sensitivity simulation with up-scaled emission sources to investigate the impact on $O_3$ and $HO_x$.

- Indeed, the model tends to underestimate the $C_2H_6$ and $C_3H_8$ atmospheric mixing ratios in most of the cases. For $C_2H_4$ and $C_3H_6$, however, the model presents mixed results depending on the location of the climatological data as already mentioned by other modelling studies (e.g., Huijnen et al., 2010).
  Recently, Dalsøren et al. (2018) showed that an increase of natural (geologic) and anthropogenic fossil fuel emissions by a factor of two to three (compared to current inventories), may significantly improve the simulated $C_2H_6$ and $C_3H_8$ mixing ratios compared to observations. Additionally, applying enhanced ethane and propane emissions results in an increase of the simulated surface ozone concentrations by 5-13%, particularly in polluted regions. Since our paper is already lengthy, we prefer to refer to that study instead of performing additional sensitivity simulations.

**GC7.** Another major concern I have is the overuse of citations when referring to earlier work. A good example is page 4 line 17-20: This sentence has 12 citations but only 18 words with providing no important information about the model at all. It feels as if every paper that used the model is cited here (without evidence why this is necessary), which should not be the goal of the model description. It should be sufficient to cite e.g. Huijnen et al., 2010 since they focus on the chemical modelling in TM5. The same holds when referring to earlier studies using parts of the mechanism (e.g. page 6 line 6-7, page 6 line 32, page 7 line 3-4), especially if they are not further used in the manuscript. It would be scientifically more profound to only cite publications, in which the approach was novel or were it was used first and not every publication using this part of the mechanism or model development. I therefore strongly advise you to recheck every citation in the manuscript and limit citations to a minimum.

- We present the main (not all) publications that show how the model evolved over time, which we believe can be very useful for a reader who wants to understand each step of the model development, offering also a source for further reading. Also, this is common practice in model description papers (e.g., in GMD) that provide the reader the opportunity to search in-depth the literature for more information about the model.

**GC8.** Last but not least, when reading the manuscript, it does not feel like a coherent story and each section feels like an isolated section. Additionally, the manuscript suffers from grammatical mistakes. I therefore suggest sweeping through the document focusing on simpler sentence structures.

- Strong structural changes and grammatical corrections will be provided in the revised manuscript.

**Specific Comments:**

**SC1.**     Page 1, Line 31-33: Not much information is given about other global models in your manuscript. Therefore, you should only focus this statement on TM5.

- We agree with the reviewer. This part now reads as: "Overall, the MOGUNTIA scheme simulates a large suite of oxygenated VOCs that are observed in the atmosphere at significant levels. This significantly expands the possible applications of TM5-MP"

**SC2.**     Page 4, Line 28-29: What influence does this approach have on the stratospheric-tropospheric exchange in your budget analysis?

- TM5-MP is a chemistry-transport model that focusses on the troposphere and no explicit stratospheric chemistry is considered. The stratospheric $O_3$ concentrations are nudged to ozone datasets to ensure realistic stratospheric $O_3$ overhead concentrations and thus a realistic chemical tropopause level (i.e., 150 ppb $O_3$ mixing ratio) for the budget analysis. A free running simulation without nudging stratospheric conditions of $O_3$ (as well as for $HNO_3$, $CH_4$) would lead to great discrepancies in tropospheric mixing ratios due to the omission of explicit stratospheric chemistry that is a source of $O_3$ (and $HNO_3$ and a sink of $CH_4$). Also, the chemical tropopause level used for the budget analysis would significantly change.

**SC3.**     Page 5, Line 4-5: When using 150 ppb as definition, the tropopause altitude will differ when using different chemical mechanisms or integrators. Do you use the same tropopause altitude for each simulation? And if so, on which simulation is this definition based? Is the tropopause altitude calculated for each time step or is it based on mean data? What impact do you expect from this?

- As a reference for this study we use the monthly mean $O_3$ concentrations from the mCB05-EBI configuration of the model, since the EBI configuration of the model has been already published multiple times in the literature. As stated in the manuscript, the differences of $O_3$ mixing ratios close to the chemical tropopause considered for this study are, however, negligible, and in all model configurations the same tropopause height is calculated. This is, we believe, due to the strong influence of nudging at these altitudes.

**SC4.**     Page 5, Line 7: The only $O_3$ chemical aqueous-phase sink considered here is $SO_2$. However, the major aqueous-phase sink of $O_3$ is the reaction with $O_2^-$ (Liang and Jacob, 1997). By not taking this sink into account, what impact do you expect this has on the $O_3$ budget and the $O_3$ burden in your analysis?

- We do not expect significant differences on a global scale. Even though aqueous phase chemistry may impact the oxidative capacity of the troposphere, this is expected to be minor compared to gas-phase sinks. Liang and Jacob (1997) clearly indicated that including aqueous phase HOx, chemistry in regional and global models of tropospheric $O_3$, is less than 3%. In contrast, hydrolysis of $NO_3$ and $N_2O_5$ on aerosols and clouds that is included in our model is, indirectly, far more important for the $O_3$ budget. Note also the relatively low Henry constant of $O_3$ (e.g., $\sim 1 \times 10_{-4}$ mol/m3/Pa @ 273.15 K; see Sander, 2015) For clarity, we note that when a detailed aqueous-phase chemistry scheme (unpublished results; work in progress) is considered in our model, a global $O_3$ sink on clouds is roughly 20 Tg/yr, thus very low compared to the gas-phase sinks.

**SC5.**     Page 7, Line 13-15: Due to the citation style used, it is not at all obvious in which publication each of the advances have been published.

- The citation style we use is the recommended by the GMD journal. Moreover, the reference(s) for each reaction are also presented in detail in Tables 1 and 2 as clearly stated in the manuscript.

**SC6.** Page 7, Line 26: How are meteorological conditions simulated in TM5? This needs to be discussed in the general description of the model (Section 2.1).

- TM5-MP is an offline CTM that reads the metrological data from the ERA-Interim database. By default, offline CTMs do not simulate meteorology but are driven by meteorological fields. In Sect. 3 we clearly state that TM5-MP is driven by meteorological fields from the ECMWF ERA-Interim reanalysis (Dee et al., 2011) with an update frequency of 3 hours. For clarity we included this description in Sect. 2.1 where we now clearly state that TM5MP is an "*offline*" CTM.

**SC7.** Page 8, Line 23-26: This information is useful to understand why KPP was implemented into TM5. I would suggest you mention this first (i.e. page 8 line 8 and in the introduction).

- We agree with the referee. This information has been moved to the beginning of Sect. 2.3 and the introduction.

**SC8.** Page 10, Line 12: What complexity has the chemical mechanism used for mCB05? Provide more information about this mechanism.

- mCB05 is a chemistry scheme which is based on the structural lumping of atmospheric species. CB05 has already published in numerous papers in the literature (e.g., Flemming et al., 2015; Houweling et al., 1998; Luecken et al., 2008; Yarwood et al., 2005; Zaveri and Peters, 1999) and the specific implementation of this chemistry scheme in the TM5-MP mode has been recently published by Williams et al. (2017). We have a separate paragraph in the introduction focusing specifically on this mechanism in Sect. 1.

**SC9.** Page 11, Line 1-15: How is this model performance analysis performed (e.g. which software)? What are the expected limitations?

- The model performance calculations are based on the timings of each procedure in the model. There is no specific software for this, but the analysis is based on the on-line calculations of the time spent per procedure as the model runs (see Table S3). The limitations for the model performance may, however, depend on the hardware.

**SC10.** Page 11, Line 2-4: This information should be included in Section 2.5.

- This part is now moved to Sect. 2.5.

**SC11.** Page 11, Line 8-9: The transport of tracers seems to be important for the model performance. How is it decided which tracer is transported and which not? This should be discussed in the model/mechanism description.

- The transport of a tracer in the model domain is mainly dependent on its lifetime relative to the applied timestep of the transport. In TM5-MP, as in most offline CTMs, all species are considered as transported except for the radicals due to their extremely short lifetime. This has already discussed in previous publications of the model, such as by Huijnen et al. (2010) and references therein.

*SC12.* Page 12, Line 7-9: This is not clear. Why is the chemical destruction higher due to changes in the $O_3$ precursors?

- We thank the reviewer for attracting our attention to this. Indeed, we think that, given the differences in the chemical scheme, chemical destruction is rather similar. Moreover, switching from EBI to the KPP-based solver has a larger influence. So, we propose: *"Chemical destruction in the troposphere is similar in the MOGUNTIA and mCB05(KPP) chemistry configurations."*

**SC13.** Page 12, Line 8-9: How do the changes in the $O_3$ precursors look like? This is a nice example were a statement is given without providing any results or argument why this must be the case (see general comments).

- This remark links to the previous one (i.e., SC12). In the manuscript we present the changes due to the different model configurations for NOx, OH and CO, which play an important role in the $O_3$ budget. However, ozone formation and destruction are non-linear processes that critically depend on the NOx/VOC ratio. A complete analysis of the ozone budget is, however, beyond the scope of this manuscript. Following the reviewer recommendation, we now provide in the Supplement the changes of the organic nitrates (ORGNTR) concentrations that represent an important pool of NOx in the model (see also our reply in SC27).

**SC14.** Page 12, Line 12: Why is it necessary to used NOy mass fixing when using EBI? This needs to be discussed in the model description since this is a major difference between EBI and KPP!

- The $NO_Y$ mass fixing in case of intense $NO_X$ photochemistry, is applied due to the approach of the EBI solver. To save computational resources, EBI employs a fixed time step with a restricted number of iterations. In some grid boxes this approach leads to incomplete convergence. This is not, however, a major difference between EBI and KPP, but a way not to miscalculate the N-budget when EBI is used. For the KPP configurations this is not needed, since the KPP-based solver (Rosenbrock) uses a variable sub-time step which ensures absolute mass conservation of N. These numerical issues are, of course, a major reason to investigate the implementation of KPP-based solvers.

**SC15.** Page 12, Line 19: This is unclear. By referring to table 3 it implies that different emission datasets are used for the different simulations. If so, why is that the case? This needs to be elaborated in Section 2.4.

- We thank the reviewer for pointing this out. Indeed, this is a typo and Table 3 should be "*Table 4*".

**SC16.** Page 13, Line 4: With the 150 ppb definition your simulation are already up to 15% higher. How does your model compare to Lamarque et al. (2012) when using 100 ppb as tropopause definition? It would be best to provide both budgets (i.e. in Table 4) for the 100 and 150 ppb definition to allow a fair comparison.

- The relative difference when accounting for the 100 ppb $O_3$ tropopause definition is added in the respective Tables within parenthesis.

**Table 1. Tropospheric budgets of $O_3$ for the year 2006 in $Tg(O_3)$ $yr^{-1}$ and burden in $Tg(O_3)$, using the 150 ppb $O_3$ mixing ratio to define tropopause level. In parenthesis the relative differences using the 100 ppb $O_3$ mixing ratios are also presented, calculated by reference to the 150 ppb $O_3$ definition of tropopause level.**

| Production terms | mCB05 (EBI) | | mCB05 (KPP) | | MOGUNTIA | | Loss terms | mCB05 (EBI) | | mCB05 (KPP) | | MOGUNTIA | |
|---|---|---|---|---|---|---|---|---|---|---|---|---|---|
| Stratospheric inflow* | 632 | (10%) | 429 | (32%) | 424 | (30%) | Deposition | 955 | (0%) | 932 | (0%) | 913 | (0%) |
| Trop. chem. production | 5589 | (-3%) | 5719 | (-3%) | 5709 | (-3%) | Trop. chem. loss | 5192 | (-1%) | 5216 | (-1%) | 5219 | (-1%) |
| Trop. burden | 385 | (-8%) | 384 | (-8%) | 375 | (-8%) | Trop. lifetime (days) | 22.8 | (-8%) | 22.8 | (-8%) | 22.3 | (-6%) |

*sum of the deposition and the tropospheric chemical loss minus the production

**Table 2. Tropospheric chemical budget of OH for the year 2006 in Tg(OH) yr$^{-1}$, using the 150 ppb O$_3$ mixing ratio to define tropopause level. In parenthesis the relative differences using the 100 ppb O$_3$ mixing ratios are also presented, calculated by reference to the 150 ppb O$_3$ definition of tropopause level.**

| Production terms | mCB05 (EBI) | | mCB05 (KPP) | | MOGUNTIA | | Loss terms | mCB05 (EBI) | | mCB05 (KPP) | | MOGUNTIA | |
|---|---|---|---|---|---|---|---|---|---|---|---|---|---|
| $O(_1D) + H_2O$ | 1960 | (0%) | 1953 | (0%) | 1878 | (0%) | $OH + CO$ | 1665 | (-2%) | 1671 | (-2%) | 1775 | (-2%) |
| $NO + HO_2$ | 1268 | (-4%) | 1312 | (-4%) | 1426 | (-4%) | $OH + CH_4$ | 613 | (0%) | 626 | (0%) | 644 | (-1%) |
| $O_3 + HO_2$ | 560 | (-1%) | 566 | (-1%) | 561 | (-1%) | $OH + O_3$ | 254 | (-2%) | 260 | (-2%) | 262 | (-3%) |
| $H_2O_2 + h\nu$ | 262 | (-1%) | 265 | (-1%) | 303 | (-1%) | $OH + ISOP$ | 114 | (-1%) | 115 | (-1%) | 120 | (0%) |
| Other | 203 | (-2%) | 201 | (-2%) | 120 | (-1%) | Other | 1606 | (-1%) | 1626 | (-1%) | 1487 | (-1%) |

**Table 3. Global budgets of CO for the year 2006 in Tg(CO) yr$^{-1}$ and burden in Tg(CO), using the 150 ppb O$_3$ mixing ratio to define tropopause level. In parenthesis the relative differences using the 100 ppb O$_3$ mixing ratios are also presented, calculated by reference to the 150 ppb O$_3$ definition of tropopause level.**

| Production terms | mCB05 (EBI) | | mCB05 (KPP) | | MOGUNTIA | | Loss terms | mCB05 (EBI) | | mCB05 (KPP) | | MOGUNTIA | |
|---|---|---|---|---|---|---|---|---|---|---|---|---|---|
| Emissions | 1097 | (0%) | 1097 | (0%) | 1097 | (0%) | Deposition | 98 | (0%) | 97 | (0%) | 99 | (0%) |
| Trop. chem. production | 1809 | (-1%) | 1818 | (-1%) | 1992 | (-1%) | Trop. chem. loss | 2840 | (-6%) | 2849 | (-6%) | 2924 | (-2%) |
| Strat. chem. production | 26 | (69%) | 26 | (73%) | 26 | (65%) | Strat. chem. loss | 87 | (68%) | 89 | (69%) | 90 | (68%) |
| Atmos. burden | 370 | (0%) | 360 | (0%) | 361 | (0%) | Lifetime (days) | 47.5 | (2%) | 46.2 | (2%) | 43.6 | (3%) |

**SC17.** Page 13, Line 13: It is not at all clear in Section 2.4 that different emissions are used. What is the impact of using different emissions?

- As clarified above we use the same emission datasets for the different chemistry configurations of the model. We here refer to the different "speciation" of the emitted species due to the required lumping, i.e., how the same VOC emissions are represented in each mechanism. To avoid confusion, we changed the word "speciation" to "*representation*".

**SC18.** Page 13, Line 30-31: This is a good argument for the model description to justify why this approach is used.

- We agree with the reviewer. We moved this part to *Sect. 2.1*.

**SC19.** Page 14, Line 4: The contribution of the "other reactions" changes from about 200 to 120 Tg/yr. What causes these changes and what is included in this category?
- This category includes the rest of the reactions in the chemical scheme. However, due to the different representation of the VOC species in mCB05 and MOGUNTIA, there is not one way to exactly match the VOC oxidation reactions, and for this reason they are added in the same pool. More details are explicitly presented in Tables 1 and 2.

**SC20.** Page 14, Line 9-10: This should be mentioned in the model description.
- We moved this part to *Sect. 2.1*.

**SC21.** Page 14, Line 12: Which tropopause definition did van Noije et al. (2014) use?
- "*150 ppb $O_3$ level for the tropopause definition*" is added in the text.

**SC22.** Page 14, Line 27: The difference is about 15%, so using "somewhat shorter" is a slight underestimation.
- "somewhat shorter" is changed to "*roughly 15% shorter*"

**SC23.** Page 14, Line 34: What lifetime do you get when using 100 ppb as tropopause definition?
- The lifetime of $CH_4$ changes only marginally (i.e., from 7.18 yr to 7.22 yr). This is, however, expected due to the relative low differences (i.e., -1%) of tropospheric $CH_4$ oxidation by OH radicals (see the new Table 5).

**SC24.** Page 15, Line 9: To what else can these differences be attributed to?
- Differences can be also attributed to differences in the general model set-up, the chemistry scheme used, the meteorology, etc.

**SC25.** Page 17, Line 18-19: This is a bit confusing. The dataset used to compare 2006 is published in 2000? What are the limitations of this comparison when using different years?
- Aircraft observations are used as climatological data, as we clearly stated in the manuscript. Some small differences are of course expected due to annual variation of emission and local meteorology changes. However, since no large differences are expected, these observations can be safely used to determine the state of model simulations.

**SC26.** Page 18, Line 21-22: Due to the lack of specific details on mCB05 in the manuscript, it is impossible to identify why this must be the case. More details are necessary here.
- The mCB05 mechanism is well documented and we deem it not necessary to repeat the tables in the manuscript (see also our reply to SC8). Moreover, the two mechanisms are presented in detail online on Zenodo. In general, the more explicit a chemical scheme, the more formation pathways are considered.

*SC27.* Page 18, Line 24: Provide more details on how NOx reservoir species differ in their concentration and spatial distribution between both mechanisms.
- The simulated annual mean surface and zonal mean organic nitrates mixing ratios for the MOGUNTIA chemistry scheme for the year 2006 and the respective differences compared to mCB05(KPP) are now added in the Sup. Material:
  "*Simulated annual mean surface (left columns) and zonal mean (right columns) mixing ratios (ppb) of organic nitrates (ORGNTR) for the MOGUNTIA chemistry scheme for the year 2006 (a,b), and the respective differences compared to mCB05(KPP) (c,d).*

[Figure]

*where, for the MOGUNTIA configuration ORGNTR represents the sum of $CH_3ONO_2$, $C_2H_5ONO_2$, $OHCH_2CH_2ONO_2$, $CH_3CH_3CH_2ONO_2$, $CH_3CH(ONO_2)CH_3$, $CH_3CH_2CH(ONO_2)CH_3$, nitrates from isoprene (ISOPNO_3), nitrates from methyl-ethyl ketone (MEKNO_3,), nitrates from methyl vinyl ketone (MVKNO_3) and nitrates from methacrolein (MACRNO_3)."*

**Table S4. Tropospheric chemical budget of ORGNTR for the year 2006 in Tg(N) yr$_{-1}$, using the 150 ppb O$_3$ mixing ratio to define tropopause level. Tropospheric burdens in Gg(N) yr$_{-1}$.**

| Production terms | mCB05 (EBI) | mCB05 (KPP) | MOGUNTIA* | Loss terms | mCB05 (EBI) | mCB05 (KPP) | MOGUNTIA* |
|---|---|---|---|---|---|---|---|
| XO$_2$N/RO$_2$ + NO | 8.586 | 8.122 | 7.030 | ORGNTR + hv | 4.077 | 4.037 | 2.621 |
| RH + NO$_3$ | 4.336 | 4.190 | 6.732 | ORGNTR + OH | 1.315 | 1.377 | 5.848 |
| Tropospheric Burden | 159.579 | 159.822 | 63.054 | Deposition | 7.424 | 7.627 | 5.132 |

*For the MOGUNTIA configuration ORGNTR represents the sum of $CH_3ONO_2$, $C_2H_5ONO_2$, $OHCH_2CH_2ONO_2$, $CH_3CH_3CH_2ONO_2$, $CH_3CH(ONO_2)CH_3$, $CH_3CH_2CH(ONO_2)CH_3$, nitrates from isoprene (ISOPNO_3), nitrates from methyl-ethyl ketone (MEKNO_3,), nitrates from methyl vinyl ketone (MVKNO_3) and nitrates from methacrolein (MACRNO_3)*

This part now reads as: "*Overall, since deep convection may efficiently transport ORGNTRs to the upper troposphere, the more explicit representation of VOC chemistry in the MOGUNTIA chemistry scheme alters the distribution of ORGNTR compared to the more lumped chemistry of mCB05. Although production of ORGNTR is about 10% larger in the MOGUNTIA scheme, the ORGNTR burden is dominated by the loss term (Table S4). Due to the more detailed speciation of the ORGNTR species in the MOGUNTIA scheme, the destruction becomes significantly more efficient compared to the mCB05 configuration. As a result, the global ORGNTR burden calculated using the MOGUNTIA scheme in the model is about 60% smaller*".

**SC28.** Page 19, Line 2-3: How well does your model compare when using 7.9 Tg-N/yr?
- The dataset with the 7.9 Tg-N yr$_{-1}$ is not available to us. Increasing the soil emissions to 7.9 Tg-N yr$_{-1}$ will not match the data from field observations.

***SC29.*** Page 19, Line 14-17: Provide evidence why this is the case.

- As we stated in our reply in SC27, a more efficient removal of the organic nitrogen is simulated for the MOGUNTIA compared to the mCB05 mechanism. This is due to the more detailed representation of these NOx reservoir species in the more explicit MOGUNTIA scheme. Organic nitrogen in the MOGUNTIA mechanism includes several species (i.e., $CH_3ONO_2$, $C_2H_5ONO_2$, $HOCH_2CH_2ONO_2$, $CH_3CH_3CH_2ONO_2$, $CH_3CH(ONO_2)CH_3$, $CH_3CH_2CH(ONO_2)CH_3$, nitrates from isoprene (ISOPNO$_3$), nitrates from methyl-ethyl ketone (MEKNO$_3$,), nitrates from methyl vinyl ketone (MVKNO$_3$) and nitrates from methacrolein (MACRNO$_3$)), while in the mCB05 mechanism, all these species are represented by one lumped ORGNTR species. Budget calculations show that although the production of ORGNTR is roughly 10% higher for the MOGUNTIA configurations compared to mCB05, the destruction is significantly more efficient (~56%) in MOGUNTIA. Therefore, the reactivity of the mixture of organic nitrogen species in MOGUNTIA mechanism is higher than that of the lumped species in mCB05 as shown in Table S4, with chemical loss of organic nitrogen by reaction with OH in the MOGUNTIA mechanism which largely compensates for the faster photolysis of these compounds in mCB05. Overall, this results in a lower tropospheric burden of ORGNTR of about 60% for the MOGUNTIA compared to mCB05 configuration. Thus, we conclude that the MOGUNTIA speciation leads to increased destruction of the organic nitrates and consequently to lower mixing ratios at higher altitudes. Concerning the impact of organic NOx reservoir species on troposphere OH mixing ratios, we note that due to the NOx release upon the destruction of ORGNTR, O$_3$ will be formed in remote locations, and thus OH recycling will be stimulated. However, a more detailed analysis would be needed to examine how the ORGNTR destruction affects NOx, O$_3$, and finally OH mixing ratios. This would be out of the scope of this paper that is focused on model development. Overall, the developments presented in this work further indicates the benefits of using the MOGUNTIA configuration in the model, since we can have a more accurate representation of ORGNTRs, and can overall predict better their distribution.
  This part now reads as: "*These relatively small differences in OH mixing ratios are mainly related to the HOx regeneration, as well as to the differences of NOx and ORGNTR species that impacts on the distribution of OH in the troposphere. The more detailed representation of ORGNTR in the MOGUNTIA chemistry scheme results in more efficient NOx release upon the ORGNTR destruction (Table S4), leading overall to O$_3$ formation in remote locations, and thus to the stimulation of HOx recycling in higher altitudes.*"

SC30.    Page 20, Line 3: What about comparing your model simulations to satellite observations of O$_3$ (e.g. OMI)?

- For this work we used two extended surface ozone observation databases and one ozonesonde database to evaluate the model and discuss the differences of the different configurations. More extended model evaluation, although always interesting, is not expected to change the conclusions of this work, especially for the simulated tropospheric ozone mixing ratios. As we refer to in the summary (Sect. 6) a more dedicated comparison of the model with the MOGUNTIA configuration with *in-situ* observations and satellite retrievals is planned to be performed in the future. Indeed, we prepare a study with an extended model evaluation with satellite retrievals. As an example of our work in progress, the reviewer can find bellow an evaluation of tropospheric O$_3$ columns (for the three configurations of this study) with OMI monthly tropospheric retrievals:

**TM5-MP (mCB05ebi mechanism) vs OMI/MLS annual 2006**

[Figure]

**Model (mCB05ebi mechanism)**

**OMI/MLS**

**abs. diff. model-satellite**

**rel. diff. model-satellite**

**TM5-MP (mCB05kpp mechanism) vs OMI/MLS annual 2006**

**Model (mCB05kpp mechanism)**

**OMI/MLS**

**abs. diff. model-satellite**

**rel. diff. model-satellite**

[Figure]

**TM5-MP (moguntia mechanism) vs OMI/MLS annual 2006**

Overall, it is obvious from this evaluation that the MOGUNTIA scheme better simulates the OMI retrievals, thus changing the model in the right direction. Note, again, that we choose not to present this evaluation in this paper, since a separate paper is in progress.

**SC31.** Line 18-20: The surface ozone bias is lowest for mCB05(KPP) but at the same time the ozone burden is higher than for MOGUNTIA. What causes this difference? Are there significant differences in free tropospheric ozone?

- Indeed, the surface ozone biases are slightly lower for mCB05(KPP). However, this conclusion cannot be straightforwardly applied to the burden differences presented in Table 4, since burdens refer to the whole troposphere, and not only to the surface level. We note also that the ~ 1ppb difference is relatively small compared to the range of $O_3$ observations.

**SC32.** Page 21, Line 10-11: This conclusion is not obvious based on the results you provided. Further analysis is needed here. How well are transport processes modelled in TM5?
- We consider such analysis outside the scope of the current paper. We indicate in the paper that model resolution "*could*" be a reason. Note that the current version of the TM5 model was included in a model intercomparison (Krol et al., 2018), in which vertical resolution were specifically addressed.

*SC33.* Page 21, Line 15-18: Are these speculations or do you have evidence that this must be the case? If so provide further details.
- As in our answer to SC32, we have no solid proof from the present study, but refer to a previous study that addressed these issues in more detail (Williams et al., 2012*)*. We think this is good practice. However, we agree that the word "can" suggests some form of evidence. Therefore, we changed this to "*could*" in the revised manuscript.

*SC34.* Page 21, Line 32: This statement is unclear. The current sentence structure implies that the emissions in the SH are lower when using KPP.
- Thanks for pointing this out. This part now reads as: "*Notably, the mCB05(EBI) model configuration tends to produce lower biases in the SH, where the emission strengths*

*are in general low, compared to the other two configurations (i.e., approximately -3 vs. -4 and -5 ppb for mCB05(KPP) and MOGUNTIA, respectively). In contrast, the MOGUNTIA chemistry configuration results in lower biases in the NH where the majority of anthropogenic emissions occur (i.e., approximately -30 vs. -31 and -33 ppb for mCB05(EBI) and mCB05(KPP), respectively).*"

**SC35.** Page 22, Line 21-23: Earlier (i.e. page 21, line 10-11) you state that the convective uplift is too low but now you state that it is too strong. Which is correct? The presented data do not support either. More evidence is needed. I strongly suggest you to perform an elaborated analysis of the performance of TM5 with respect to transport processes, to justify these claims.

- We improved changed this section and added further analyses. Overall, these parts are now read as:
  - $O_3$: *The model evaluation at pressure levels < 300 hPa indicates there is good agreement of both configuration with the observed mixing ratios. A positive bias in April in the order of ~20 ppb is calculated for the model, but smaller biases are found around the tropics and in the latitudes north of 40oN (Fig. S4a). In October (Fig. 4Sb), a constant positive bias of roughly 20 ppb is calculated for both configurations. This could be caused by the limited vertical resolution of this model version in the UTLS region. Note here that 34 vertical levels were employed for this study with a higher resolution in the upper troposphere–lower stratosphere compared to the low and mid-troposphere region. Part of the model overestimation could also be attributed to systematic errors, as also presented in previous studies (e.g., Huijnen et al., 2010), caused possibly by cumulative effects such as a lack of a diurnal or weekly variation in the NOx emissions from the road transport sector, an underestimation of surface deposition during summer or even errors in the representation of nocturnal boundary layer dynamics (e.g., Williams et al., 2012), which are common issue in global chemistry transport model.*

  - CO: *Model evaluation at pressure levels < 300 hPa shows a good correlation for both configurations in the SH, with a small positive bias (up to ~20 ppb) for the mCB05(KPP) configuration in April around the equator and a small negative bias (~10 ppb) for the MOGUNTIA configuration for latitudes below 10oN. Both configurations present a strong negative bias (~30 ppb) for latitudes above 20oN (Fig. S4c). In October (Fig. S4d), both the mCB05(KPP) and MOGUNTIA configurations tend to underestimate the observations with a negative bias of ~20 ppb, except for a small positive bias between 0-20oN. This positive model bias in the UTLS could point to a stronger convective uplift in tropical Africa in April or to a possible misrepresentation of biomass burning emissions that are generally uncertain (Nechita-Banda et al., 2018). Indeed, MOZAIC data presents an increase in CO mixing ratios from the NH (April) to the SH (October), owing mainly to the impact of biomass burning processes. Overall, the model configurations of this work present both positive and negative biases compared to the MOZAIC observations, with the observations to exhibit in general larger latitudinal CO variability.*

**SC36.** Page 22, Line 24-25: Have you analysed biomass burning hotspots to support this claim?

- Analyzing biomass burning hotspots separately would be out of the scope of this work. However, previous studies with the TM5 model show large uncertainties in bottom-up estimates of biomass burning emissions, likely caused by uncertainties in emissions factors (Nechita-Banda et al., 2018). In addition to the biomass burning emission strength and geographic distribution, Daskalakis et al. (2015) have shown the sensitivity

of the model results to the biomass burning emissions injection heights. We added these references to better highlight this uncertainty.

**SC37.** Page 23, Line 2: What causes the opposite annual cycle? i.e. indicate that $C_2H_6$ surface mixing ratios are strongly underestimated by all configurations at Mace Head (Fig. 9a) by ~80%, mainly during the winter, indicating also an opposite annual cycle.

- $C_2H_6$ surface mixing ratios and their seasonal cycle in the model depend on the emission strength and the oxidation by OH radicals. Underestimation of emissions or a faster oxidation could explain the differences between model and observations. We propose to add the following sentence "*This can be attributed to the misinterpretation of seasonal variation of anthropogenic $C_2H_6$ emission and/or to a winter overestimate of $C_2H_6$ oxidation by OH radicals in the model.*"

*SC38.* Page 23, Line 9: Your model underestimates propane but you use a lower emission than other studies. How does your model compare when you use higher emissions?

- The emissions used for this study come from the CMIP6 databases. Indeed, an increase (or decrease) of emissions may help to investigate the response of the model to identify possible biases in the emission databases. To show here how the model responds to an increase of emissions for both ethane and propane, we present bellow a model comparison with flask measurements using 1) the base case emission scenario, 2) doubling (2x) of the anthropogenic fossil fuel emissions of $C_2H_6$ and $C_3H_8$, resulting in ~17.1 Tg yr-1 and ~14.9 Tg yr-1, respectively, and 3) quadrupling (4x) of the anthropogenic fossil fuel emissions of $C_2H_6$ and $C_3H_8$, resulting in ~29.5 Tg yr-1 and ~27.9 Tg yr-1 respectively. For this sensitivity study, we run the model in 3o x 2o horizontal resolution in longitude by latitude, and 34 hybrid levels in the vertical, which is much cheaper compared to 1x1 horizontal resolution used in the paper. Note that our approach is based on the recent study by Dalsøren et al. (2018) (see also our reply in GC6), showing that an increase of natural (geologic) and anthropogenic fossil fuel emissions by two to three times may improve the simulated $C_2H_6$ and $C_3H_8$ mixing ratios compared to observations.

-

[Figure]

[Figure]

[Figure]

**Figure: comparison between TM5 (MOGUNTIA scheme) simulations and observations of ethane (left) and propane (right) for 4 stations.**

From the figures above, it is obvious that the increase of $C_2H_6$ anthropogenic emissions by two or four times does not significantly increase the simulated mixing ratios (please mind here the log scale in the y-axis). This means that (1) even more aggressive increase of emissions (at least over specific regions) is required, (2) other missing sources are needed, or (3) that the oxidation of $C_2H_6$ is too fast in the model. In contrast, the increase of $C_3H_8$ emissions by two times tends to improve the model simulations in most of the cases, where an increase by a factor 4 tends to overestimate the observed mixing ratios. Overall, our results suggest that changes in emissions should not be based on fixing the model to a specific value. Instead, scientifically accepted methods, such as data assimilation, should be used to minimize the difference between observations and the model by emissions optimization. Nevertheless, these sensitivity studies give interesting insights!

We suggest adding the following parts in
i)        Sect. 5.5.1:
*"Dalsøren et al. (2018) showed recently that an increase of natural and anthropogenic fossil fuel emissions by a factor of two to three may significantly improves the simulated $C_2H_6$ and $C_3H_8$ mixing ratios compared to observations. This would result in source estimates close to the 16 Tg yr-1 and 23 Tg yr-1 for $C_2H_6$ and $C_3H_8$ respectively, as have been calculated by the first global 2-d modeling study of these two hydrocarbons by Kanakidou et al. (1991). To investigate here how the model responses to an increase of ethane emissions, sensitivity simulations with the MOGUNTIA configuration are here performed by i) doubling and ii) quadrupling the anthropogenic $C_2H_6$ fossil fuel emissions, resulting overall in total $C_2H_6$ emissions of ~17.1 Tg yr-1 and ~29.5 Tg yr-, respectively. The comparison with the with flask data (Fig. S7) indicates that the increase of $C_2H_6$ anthropogenic emissions does not significantly affect the simulated mixing ratios in the model. Overall, this means that i) even a more aggressive increase of emissions (at least over specific regions) or a different geographic distribution of emission is required, ii) other missing sources are needed to be considered in the model, or iii) the oxidation of $C_2H_6$ is too fast in the model."*
ii) Sect. 5.5.1:
*" Additional simulations for C3H8 are performed by i) doubling and ii) quadrupling the anthropogenic fossil fuel emissions, resulting overall in total $C_3H_8$ emissions of ~14.9 Tg yr-1 and ~27.9 Tg yr-1 respectively. Figure S7 indicates that an increase of $C_3H_8$ emissions by two times tends to improve the model simulations in most of the cases, whereas an increase by a factor 4 tends to overestimate the observed mixing ratios."*
iii) Sect. 6:
*"Sensitivity simulations of this work indicate that increases in emissions may have a significant impact on some light VOC atmospheric concentrations, such as the $C_3H_8$. However, our results suggest that changes in emissions should not be based on fixing the model to a specific (constant) value. Instead, scientifically accepted methods should be used."*

***SC39.*** Page 24, Line 1-2: Could this underestimation be related to underestimated transport processes (see Page 21 & 22)?

- Some discrepancies in transport could explain part of the model underestimation. However, propane emissions strength or misrepresentation of their horizontal or/and vertical distribution along with a fast propane oxidation by OH radicals seem to be the main reasons for the differences between model and observations.

**SC40.** Page 24, Line 20: What needs to be done to account for the "secondary production from VOC oxidations"?

- We should investigate possible unknown chemical pathways via heavier VOCs oxidation (e.g. in smog chamber studies).

***SC41.*** Page 25, Line 30-33: Can you provide some suggestions on how to improve these uncertainties?

- We suggest to add: "*Future studies should aim at improving source estimates and a better understanding of the processes that govern the budgets of the light VOCs. From a chemistry point of view, it would be interesting to study the chemical formation pathways from higher VOCs. Inverse modelling or data-assimilation studies might be used to "optimize" the emissions in order to minimize the differences between observations and model simulations.*"

**SC42.** Page 53, Table 4: What about $O_3$ scavenging?

- TM5-MP, following a common practice in global chemistry transport models, does not include wet scavenging processes for $O_3$. Since the washout effects depend on the species' solubility and considering the low solubility of $O_3$ (see Sander, 2015), scavenging is not a significant removal process from the atmosphere. This is also supported by observations based on analysis from long-term hourly data (Yoo et al., 2014), where the impact of washout on $O_3$ was negligible.

**Technical corrections:**

**TC1.** Page 2, lines 4-20: A graphical illustration of the NOx-VOC-O₃ relation would be helpful here.

- A graphical illustration of the NOx-VOC-O₃ relation is well documented, e.g., see Rethinking the Ozone Problem in Urban and Regional Air Pollution (National Research Council, 1991):

[Figure]

Although such a graphical illustration could be helpful for the reader, this is would be out of the scope of the current work which is focused on model development.

**TC2.** Page 4, line 22: Check gramma and wording.

- This part now reads as: 'In this new MP version, the two-way zoom capability of TM5 is no longer available.'

**TC3.** Page 14, line 1-2: Check gramma and wording.

- This part now reads as: 'The MOGUNTIA model configuration yields direct gas-phase OH formation (via $O_3$ photolysis in the presence of water molecules, Reactions 3 and 4) of 1878 Tg yr$^{-1}$. Radical recycling terms (Reactions 1 and 5) contribute 1987 Tg yr$^{-1}$. Finally, the $H_2O_2$ photodissociation, i.e., $H_2O_2 + h\nu \rightarrow 2$ OH (7) produces 303 Tg yr$^{-1}$, and all other reactions add another 120 Tg yr$^{-1}$ to the global tropospheric OH production in the model.'

**TC4.** Page 18, line 11-13: Check gramma and wording.

- This part now reads as: "Some discrepancies are nevertheless expected in such a comparison since no seasonal cycle in anthropogenic emissions is considered. Anthropogenic emissions are the major source of NOx in the Northern Hemisphere (NH)."

**TC5.** Page 5, line 3: The statement that this study focuses on the troposphere is stated multiple times. Do not use double statements, to improve the reading flow.

- Statement removed.

**TC6.** Page 6, line 1: This should be Section 2.2.

- Done

**TC7.** Page 9, line 10-14: Listing all species greatly disturbs the reading flow. I would remove this listing and just refer to Table 3 instead.

- These species refer to the database and not to the model as clearly stated in the beginning of Sect. 2.4. Thus, we cannot just refer to Table 3 since the provided emissions are not the same with the model's species because the required

lumpings/sums have to be performed. This is also stated in the 3rd paragraph of Sect. 2.4, i.e., Overall, the MOGUNTIA chemical scheme considers direct emissions…

**TC8.** Page 10, line 12-28: A table summarizing all simulations performed could be useful.
- We think that such a table is of little added value, as we present only the results of three simulations. that are already presented multiple times in each budget table and each plot. Moreover, a complete description of each simulation (although simple) is provided in each caption.

**TC9.** Page 12, line 22-26: This is a rather complicated sentence. Consider using simpler language (i.e. multiple short sentences).
- This part now reads as:" The calculated net influx from the stratosphere for the MOGUNTIA configuration (~424 Tg yr-1) remains within one standard deviation of a multi-model mean estimate (552 ± 168 Tg yr-1), as reported by Stevenson et al. (2006) and Young et al. (2013). MOGUNTIA calculations are also in line with estimates based on observations (Hsu, 2005; Olsen, 2004) (~400 Tg yr$_{-1}$). Our estimates are higher compared to the 306 Tg yr$_{-1}$ calculated in an earlier version of the TM5 model driven by the same meteorological fields (van Noije et al., 2014)."

**TC10.** Page 14, line 26: The word "arrive" should not be used here.
- This part now reads as: "*an atmospheric lifetime of about 7.18 yr is derived*"

**TC11.** Page 17, line 2-19: Presenting the different observations and possible comparisons in a table would be more efficient.
- We prefer to keep the text as is.

**TC12.** Page 17, line 25: In order to improve the reading flow, it would be best to first compare each tracer discussed in Section 4 (in the same order).
- We thank the reviewer for this comment. This is what we intended to do in the presentation of the results. In more detail, in Sect. 4 the budget follows the way the reactions are described (i.e., $O_3$, OH and CO). However, for a useful model evaluation of $O_3$, CO, and VOCs, we need first a discussion of the modelled NOx and OH atmospheric mixing ratios.

**TC13.** Page 20, line 25: Is the reference to the introduction correct?
- We thank the referee for pointing out this typo. Sect. 1. changed to *Sect. 3.*

**TC14.** Page 20, line 27: Please be more specific and refer to Section 2.1.
- Done

**TC15.** Page 41-51: Most of the information presented in Tables 1, 2 and even 3 are well documented elsewhere. Thus, I strongly recommend you to move these tables to the supplemental material.
- The information in Tables 1-3 is of course documented elsewhere in the literature since all reactions are based on state-of-the-art databases such as the IUPAC, MCM, but their combination and the assumptions applied for this work are not. Moreover, compared to the previous version of the MOGUNTIA chemistry scheme, numerous updates have been performed. Overall, since the aim of this paper is to present the new coupling of the MOGUNTIA chemistry scheme to the TM5MP CTM, these tables should remain the main text. All other model development papers which are focused to chemistry follow the same procedure.

**TC16.** Supplement, page 3: Table S3 (including caption) cannot be read completely.
- Corrected.

  Following, however, the reviewer's comment, brief description and references of the transport processes parameterizations in TM5 are added in Model Description (Sect. 2.1): "*The advection scheme used in TM5 is based on the slopes scheme (Russell and Lerner, 1981) and the deep and shallow cumulus convection scheme is parameterized according to Tiedtke (1989). The performance of the transport in the model has been evaluated by (Peters et al., 2004) using sulphur hexafluoride simulations and by analyzing the vertical and horizontal distribution of radon ($222Rn$) to simulate the boundary layer dynamics (Koffi et al., 2016; Williams et al., 2017). More recently, global transport features, such as the transport times associated with inter-hemispheric transport, vertical mixing in the troposphere, transport to and in the stratosphere, and transport of air masses between land and ocean, were evaluated via an inter-comparison of six global transport models (Krol et al., 2018).*"

**Specific Comments:**

**SC1.** Page 4, lines 4-5: Use of 150 ppb, or any concentration level has caveats, e.g. model bias. Why not use the meteorological tropopause instead? The implications should be addressed. this?

- For this work, as we stated in the manuscript, we use the chemical tropopause level defined by a 150 ppb $O_3$ mixing ratio following the well-documented model intercomparison study by Stevenson et al. (2006). The use of the 150 ppb $O_3$ level has been used so far in numerous studies, as also with previous versions of the TM5 model, providing thus an opportunity of a direct comparison of model results with other estimates. On the other hand, the tropopause levels in a model may have various definitions, such as the temperature and the potential vorticity gradients, the altitude or the standard World Meteorological Organization definition that the lowest level above 500 hPa where the vertical temperature gradient decreases to less than or equal 2 $_o$C km$_{-1}$.

  We agree with the reviewer that the definition of the tropopause may lead to great differences, and for this, we stated in the manuscript that the tropopause definition should always be reported when comparing modelling estimates.

  For this work, however, we prefer to keep the tropopause based on the 150 ppb $O_3$ mixing ratio since we here mostly focused on the differences between the different configurations of the model. However, to show the impact of the use of different tropopause levels on the calculated tropospheric budgets, we now provide the relative differences of using the 100 ppb $O_3$ level, i.e.:

**Table 1. Tropospheric budgets of $O_3$ for the year 2006 in Tg($O_3$) yr$_{-1}$ and burden in Tg($O_3$), using the 150 ppb $O_3$ mixing ratio to define tropopause level. In parenthesis the relative differences using the 100 ppb $O_3$ mixing ratios are also presented, calculated by reference to the 150 ppb $O_3$ definition of tropopause level.**

| Production terms | mCB05 (EBI) | | mCB05 (KPP) | | MOGUNTIA | | Loss terms | mCB05 (EBI) | | mCB05 (KPP) | | MOGUNTIA | |
|---|---|---|---|---|---|---|---|---|---|---|---|---|---|
| Stratospheric inflow* | 632 | (10%) | 429 | (32%) | 424 | (30%) | Deposition | 955 | (0%) | 932 | (0%) | 913 | (0%) |
| Trop. chem. production | 5589 | (-3%) | 5719 | (-3%) | 5709 | (-3%) | Trop. chem. loss | 5192 | (-1%) | 5216 | (-1%) | 5219 | (-1%) |
| Trop. burden | 385 | (-8%) | 384 | (-8%) | 375 | (-8%) | Trop. lifetime (days) | 22.8 | (-8%) | 22.8 | (-8%) | 22.3 | (-6%) |

*sum of the deposition and the tropospheric chemical loss minus the production*

**Table 2. Tropospheric chemical budget of OH for the year 2006 in Tg(OH) yr−1, using the 150 ppb O3 mixing ratio to define tropopause level. In parenthesis the relative differences using the 100 ppb O3 mixing ratios are also presented, calculated by reference to the 150 ppb O3 definition of tropopause level.**

| Production terms | mCB05 (EBI) | | mCB05 (KPP) | | MOGUNTIA | | Loss terms | mCB05 (EBI) | | mCB05 (KPP) | | MOGUNTIA | |
|---|---|---|---|---|---|---|---|---|---|---|---|---|---|
| $O(^1D) + H_2O$ | 1960 | (0%) | 1953 | (0%) | 1878 | (0%) | OH + CO | 1665 | (-2%) | 1671 | (-2%) | 1775 | (-2%) |
| $NO + HO_2$ | 1268 | (-4%) | 1312 | (-4%) | 1426 | (-4%) | $OH + CH_4$ | 613 | (0%) | 626 | (0%) | 644 | (-1%) |
| $O_3 + HO_2$ | 560 | (-1%) | 566 | (-1%) | 561 | (-1%) | $OH + O_3$ | 254 | (-2%) | 260 | (-2%) | 262 | (-3%) |
| $H_2O_2 + h\nu$ | 262 | (-1%) | 265 | (-1%) | 303 | (-1%) | OH + ISOP | 114 | (-1%) | 115 | (-1%) | 120 | (0%) |
| Other | 203 | (-2%) | 201 | (-2%) | 120 | (-1%) | Other | 1606 | (-1%) | 1626 | (-1%) | 1487 | (-1%) |

**Table 3. Global budgets of CO for the year 2006 in Tg(CO) yr−1 and burden in Tg(CO), using the 150 ppb O3 mixing ratio to define tropopause level. In parenthesis the relative differences using the 100 ppb O3 mixing ratios are also presented, calculated by reference to the 150 ppb O3 definition of tropopause level.**

| Production terms | mCB05 (EBI) | | mCB05 (KPP) | | MOGUNTIA | | Loss terms | mCB05 (EBI) | | mCB05 (KPP) | | MOGUNTIA | |
|---|---|---|---|---|---|---|---|---|---|---|---|---|---|
| Emissions | 1097 | (0%) | 1097 | (0%) | 1097 | (0%) | Deposition | 98 | (0%) | 97 | (0%) | 99 | (0%) |
| Trop. chem. production | 1809 | (-1%) | 1818 | (-1%) | 1992 | (-1%) | Trop. chem. loss | 2840 | (-6%) | 2849 | (-6%) | 2924 | (-2%) |
| Strat. chem. production | 26 | (69%) | 26 | (73%) | 26 | (65%) | Strat. chem. loss | 87 | (68%) | 89 | (69%) | 90 | (68%) |
| Atmos. burden | 370 | (0%) | 360 | (0%) | 361 | (0%) | Lifetime (days) | 47.5 | (2%) | 46.2 | (2%) | 43.6 | (3%) |

**SC2.** Page 13, line 13. Use of different emissions are not clearly mentioned in section 2.4. Authors should justify the use of different emissions and how this impacts the changes they see in the different scenarios.

- As explained in our replies to the other reviewer (RC1), we use the same emissions (and boundary conditions) for the different chemistry configurations of the model. This choice is made in order to specifically focus only on their differences between the two mechanisms in the model as explicitly presented in Sect. 3. In the manuscript we refer to the different "speciation" of the emitted volatile organic compounds (VOC) i.e. how the VOC emissions are distributed among the VOC species considered in the different chemical mechanisms: the more lumped mCB05 does not resolve all of the NMVOCs provided by the emission datasets, whereas MOGUNTIA explicitly simulates the NMVOCs (C1-4) and isoprene. To make this point clearer, however, we changed the word "speciation" with "*representation*" when we refer here to the differences between the two chemical schemes (see also our reply to SC17) and we clearly state in the manuscript that both mechanisms use the same emission datasets.
-

**SC3.** Page 20, Line 3: It would be great if the results are compared with satellites

- For this work we used two extended surface ozone observation databases and one ozonesonde database to evaluate the model and discuss the differences of the different configurations. More extended model evaluation, although always interesting, is not however expected to change the conclusions of this work, especially for the simulated tropospheric ozone mixing ratios. On the other hand, as also we refer in the summary (Sect. 6) a more dedicated comparison of the model with the MOGUNTIA configuration with *in-situ* observations and satellite retrievals is planned to be performed in the future. As an example of our work in progress, the reviewer can find bellow an evaluation of tropospheric $O_3$ columns (for the three configurations of this study) with the respective OMI monthly tropospheric retrievals:

[Figure]

**TM5-MP (mCB05kpp mechanism) vs OMI/MLS annual 2006**

[Figure]

**TM5-MP (moguntia mechanism) vs OMI/MLS annual 2006**

[Figure]

Overall, it is obvious from this evaluation, that the MOGUNTIA scheme simulates better the OMI retrievals, thus leading the model in the right direction. Note, again, that we choose not to present this evaluation in this paper, since a separate paper is in progress.

**References:**

[revised manuscript text omitted]

| | | | |
|---|---|---|---|
| J40 | HOC$_3$H$_6$OOH + $h\nu$ | $\rightarrow$ CH$_3$CHO + HCHO + HO$_2$ | J13 |
| J41 | CH$_3$COCHO + $h\nu$ | $\rightarrow$ CH$_3$C(O)OO + CO + HO$_2$ | 1 |
| J42 | C$_4$H$_9$OOH + $h\nu$ | $\rightarrow$ 0.67(CH$_3$CH$_2$COCH$_3$+ HO$_2$) + 0.33(C$_2$H$_5$OO +CH$_3$CHO) + OH | J13 |
| J43 | C$_4$H$_9$ONO$_2$ + $h\nu$ | $\rightarrow$ 0.67(CH$_3$CH$_2$COCH$_3$ + HO$_2$) + 0.33(C$_2$H$_5$OO +CH$_3$CHO) + NO$_2$ | J$_{ORGN}$ |
| J44 | CH$_3$CH$_2$C(O)CH$_3$ + $h\nu$ | $\rightarrow$ CH$_3$C(O)OO+ C$_2$H$_5$OO | 1 |
| J45 | CH$_3$CH(OOH)COCH$_3$ + $h\nu$ | $\rightarrow$ CH$_3$CHO + CH$_3$C(O)OO+ OH | J13 |
| J46 | CH$_3$CH(ONO$_2$)COCH$_3$ + $h\nu$ | $\rightarrow$ CH$_3$CHO + CH$_3$C(O)OO+ NO$_2$ | J$_{ORGN}$ |
| J47 | ISOPOOH + $h\nu$ | $\rightarrow$ HCHO + 0.64 MVK + 0.36 MACR + HO$_2$ + OH | 13 |
| J48 | ISOPONO$_2$ + $h\nu$ | $\rightarrow$ HCHO + 0.64 MVK + 0.36 MACR + HO$_2$ + NO$_2$ | J$_{ORGN}$ |
| J49 | MACR + $h\nu$ | $\rightarrow$ 0.5 MACROO + 0.5 HCHO + 0.175 CH$_3$C(O)OO+ 0.325 CH$_3$OO + 0.825 CO + HO$_2$ | 1 |
| J50 | MACROOH + $h\nu$ | $\rightarrow$ CH$_3$COCH$_2$OH + CO + HO$_2$ + OH | J13 |
| J51 | MACRONO$_2$ + $h\nu$ | $\rightarrow$ CH$_3$COCH$_2$OH + CO + HO$_2$ + NO2 | J$_{ORGN}$ |
| J52 | MVK + $h\nu$ | $\rightarrow$ 0.6 (C$_3$H$_6$ + CO) + 0.4 (CH$_3$C(O)OO + CH$_3$OO + HCHO) | 1 |
| J53 | MVKOOH + $h\nu$ | $\rightarrow$ 0.7(CH$_3$C(O)OO+ HOCH$_2$CHO) + 0.3(CH$_3$COCHO + 
[revised manuscript text omitted]

|---|---|---|---|---|---|
| K83 | HOCH$_2$CHO + OH | $\rightarrow$ | CHOCHO + HO$_2$ | 1.6 x 10$^{-12}$ | 1 |
| K84 | CHOCHO + OH | $\rightarrow$ | 2CO + HO$_2$ | 3.1 x 10$^{-12}$exp(340/T) | 1 |
| K85 | CHOCHO + NO$_3$ | $\rightarrow$ | 2CO + HO$_2$ + HNO$_3$ | 4.0 x 10$^{-16}$ | 1 |
| K86 | CH$_3$COOH + OH | $\rightarrow$ | CH$_3$OO + CO$_2$ | 4.0 x 10$^{-14}$exp(850/T) | 1 |
| K87 | CH$_3$CH$_2$OH + OH | $\rightarrow$ | 0.95 (CH$_3$CHO + HO$_2$) + 0.05 HOCH$_2$CH$_2$OO | 3.0 x 10$^{-12}$exp(20/T) | 1 |
| K88 | C$_3$H$_8$ + OH | $\rightarrow$ | 0.264 $n$-C$_3$H$_7$O$_2$ + 0.736 $i$-C$_3$H$_7$O$_2$ | 7.6 x 10$^{-12}$exp(-585/T) | 1, 3 |
| K89 | $n$-C$_3$H$_7$O$_2$+ HO$_2$ | $\rightarrow$ | $n$-C$_3$H$_7$OOH | 0.52 x 2.91 x 10$^{-13}$exp(1300/T) | 3 |
| K90 | $n$-C$_3$H$_7$O$_2$ + NO | $\rightarrow$ | C$_2$H$_5$CHO + HO$_2$ + NO$_2$ | (1 - RTC3P) x 2.9 x 10$^{-12}$exp(350/T) | 1, 4 |
| K91 | $n$-C$_3$H$_7$O$_2$ + NO | $\rightarrow$ | $n$-C$_3$H$_7$ONO$_2$ | RTC3P x 2.9 x 10$^{-12}$exp(350/T) | 1, 4 |
| K92 | $n$-C$_3$H$_7$O$_2$ + CH$_3$OO | $\rightarrow$ | C$_2$H$_5$CHO + CH$_3$OH | 0.8 x (3.5 x 10$^{-13}$ x 3.0 x 10$^{13}$)$^{0.5}$ | 3 |
| K93 | $n$-C$_3$H$_7$O$_2$ + CH$_3$OO | $\rightarrow$ | C$_2$H$_5$CHO + HCHO + 2HO$_2$ | 0.2 x (3.5 x 10$^{-13}$ x 3.0 x 10$^{13}$)$^{0.5}$ | 3 |
| K94 | $n$-C$_3$H$_7$OOH + OH | $\rightarrow$ | $n$-C$_3$H$_7$O$_2$ | K76 | |
| K95 | $n$-C$_3$H$_7$OOH + OH | $\rightarrow$ | C$_2$H$_5$CHO + OH | 1.66 x 10$^{-11}$ | 3 |
| K96 | $n$-C$_3$H$_7$ONO$_2$ + OH | $\rightarrow$ | C$_2$H$_5$CHO + NO$_2$ | 5.8 x 10$^{-13}$ | 1 |
| K97 | $i$-C$_3$H$_7$O$_2$ + HO$_2$ | $\rightarrow$ | $i$-C$_3$H$_7$OOH | K89 | |
| K98 | $i$-C$_3$H$_7$O$_2$ + NO | $\rightarrow$ | CH$_3$COCH$_3$ + HO$_2$ + NO$_2$ | (1 - RTC3S) * 2.7 x 10$^{-12}$exp(360/T) | 1, 4 |
| K99 | $i$-C$_3$H$_7$O$_2$ + NO | $\rightarrow$ | $i$-C$_3$H$_7$ONO$_2$ | RTC3S * 2.7 x 10$^{-12}$exp(360/T) | 1, 4 |
| K100 | $i$-C$_3$H$_7$O$_2$ + CH$_3$OO | $\rightarrow$ | CH$_3$COCH$_3$ + HCHO +2HO$_2$ | 0.8 * (1.03 x 10$^{-13}$exp(365/T) * 1.6 x 10$^{-12}$exp(-2200/T))$^{0.5}$ | 3 |
| K101 | $i$-C$_3$H$_7$O$_2$ + CH$_3$OO | $\rightarrow$ | CH$_3$COCH$_3$ + CH$_3$OH | 0.2 * (1.03 x 10$^{-13}$exp(365/T) x 1.6 x 10$^{-12}$exp(-2200/T))$^{0.5}$ | 3 |
| K102 | $i$-C$_3$H$_7$OOH + OH | $\rightarrow$ | $i$-C$_3$H$_7$O$_2$ | 1.9 x 10$^{-12}$exp(190/T) | 3 |
| K103 | $i$-C$_3$H$_7$OOH + OH | $\rightarrow$ | CH$_3$COCH$_3$ + OH | 1.66 x 10$^{-11}$ | 3 |
| K104 | $i$-C$_3$H$_7$ONO$_2$ + OH | $\rightarrow$ | CH$_3$COCH$_3$ + NO$_2$ | 6.2 x 10$^{-13}$exp(-230/T) | 1 |
| K105 | C$_2$H$_5$CHO + OH | $\rightarrow$ | CH$_3$C(O)OO + CO | 4.9 x 10$^{-12}$exp(405/T) | 1 |
| K106 | C$_2$H$_5$CHO + NO$_3$ | $\rightarrow$ | CH$_3$C(O)OO + CO + HNO$_3$ | 6.3 x 10$^{-15}$ | 1 |
| K107 | CH$_3$COCH$_3$ + OH | $\rightarrow$ | CH$_3$COCH$_2$OO | 8.8 x 10$^{-12}$exp(-1320/T) + 1.7 x 10$^{-14}$exp(423/T) | 1 |
| K108 | CH$_3$COCH$_2$OO+ NO | $\rightarrow$ | CH$_3$COCHO + NO$_2$ + HO$_2$ | 2.7 x 10$^{-13}$exp(360/T) | 3 |
| K109 | CH$_3$COCH$_2$OO+ HO$_2$ | $\rightarrow$ | CH$_3$COCH$_2$OOH | 1.36 x 10$^{-13}$exp(1250/T) | 3 |
| K110 | CH$_3$COCH$_2$OOH + OH | $\rightarrow$ | 0.7 CH$_3$COCHO + 0.3 CH$_3$COCH$_2$OO + OH | 1.90 x 10$^{-12}$exp(190/T) | 3 |
| K111 | C$_3$H$_6$ + OH | $\rightarrow$ | HOC$_3$H$_6$OO | 8 x 10$^{-27}$(T/300)$^{-3.5}$[N$_2$] 3.0 x 10$^{-11}$($T$/300)$^{-1.0}$ $F$c = 0.5 | 1 |
| K112 | C$_3$H$_6$ + NO$_3$ | $\rightarrow$ | 0.35 $n$-C$_3$H$_7$ONO$_2$ + 0.65 $i$-C$_3$H$_7$ONO$_2$ | 4.6 x 10$^{-13}$exp(-1155/T) | 1, 3 |
| K113 | C$_3$H$_6$ + O$_3$ | $\rightarrow$ | 0.62 HCHO + 0.62 CH$_3$CHO + 0.38 CH$_3$OO + 0.56 CO + 0.36 HO$_2$ + 0.36 OH + 0.2 CO$_2$ | 5.77 x 10$^{-15}$exp(-1880/T) | 1, 3 |
| K114 | HOC$_3$H$_6$OOH + OH | $\rightarrow$ | 0.928 CH$_3$COCH$_2$OH + 0.072 HOC$_3$H$_6$OO + 0.928 OH | 2.44 x 10$^{-11}$ + 1.9 x 10$^{-12}$exp(190/T) | 3 |
| K115 | HOC$_3$H$_6$OO + HO$_2$ | $\rightarrow$ | HOC$_3$H$_6$OOH | K89 | 3 |
| K116 | HOC$_3$H$_6$OO + NO | $\rightarrow$ | CH$_3$CHO + HCHO + HO$_2$ + NO$_2$ | (1 – 0.35$RTC3P$ – 0.65$RTC3S$) * 2.55 x 10$^{-12}$exp(380/T) | 1, 3 |
| K117 | HOC$_3$H$_6$OO + NO | $\rightarrow$ | 0.35 $n$-C$_3$H$_7$ONO$_2$ + 0.65 $i$-C$_3$H$_7$ONO$_2$ | (0.35$RTC3P$ + 0.65$RTC3S$) * | 1, 3 |

| | | | 2.55 x 10$^{-12}$exp(380/T) | |
|---|---|---|---|---|
| K118 | HOC$_3$H$_6$OO + CH$_3$OO | → CH$_3$CHO + 2HCHO +2HO$_2$ | 0.8 * 6.0 x 10$^{-13}$ | 3 |
| K119 | HOC$_3$H$_6$OO + CH$_3$OO | → CH$_3$COCH$_2$OH + CH$_3$OH | 0.2 * 6.0 x 10$^{-13}$ | 3 |
| K120 | CH$_3$COCH$_2$OH + OH | → CH$_3$COCHO + HO$_2$ | 1.6 x 10$^{-12}$exp(305/T) | 1 |
| K121 | CH$_3$COCHO + OH | → CH$_3$C(O)OO + CO | 1.9 x 10$^{-12}$exp(575/T) | 1 |
| K122 | CH$_3$COCHO + NO$_3$ | → CH$_3$C(O)OO + CO + HNO$_3$ | 5.0 x 10$^{-16}$ | 1 |
| K123 | CH$_3$C(O)COOH + OH | → CH$_3$C(O)OO + CO$_2$ | 8.0 x 10$^{-13}$ | 3 |
| K124 | C$_4$H$_{10}$ + OH | → C$_4$H$_9$OO | 9.8 x 10$^{-12}$exp(-425/T) | 3 |
| K125 | C$_4$H$_{10}$ + NO$_3$ | → C$_4$H$_9$OO + HNO$_3$ | 2.8 x 10$^{-12}$ exp(-3280/T) | 1 |
| K126 | C$_4$H$_9$OO + HO$_2$ | → C$_4$H$_9$OOH | 0.625 * 2.91 x 10$^{-13}$exp(1300/T) | 3 |
| K127 | C$_4$H$_9$OO + NO | → NO$_2$ + 0.67(CH$_3$CH$_2$COCH$_3$ + HO$_2$) + 0.33(C$_2$H$_5$OO + CH$_3$CHO) | (1 -RTC4P) x 8.3 x 10$^{-12}$ | 1, 4 |
| K128 | C$_4$H$_9$OO + NO | → C$_4$H$_9$ONO$_2$ | RTC4P x 8.3 x 10$^{-12}$ | 1, 4 |
| K129 | C$_4$H$_9$OO + CH$_3$OO | → HCHO + HO$_2$+ 0.67(CH$_3$CH$_2$C(O)CH$_3$ + HO$_2$) + 0.33(CH$_3$CHO + CH$_3$CH$_2$OO) | 0.8 * 1.3 x 10$^{-12}$ | 3 |
| K130 | C$_4$H$_9$OO + CH$_3$OO | → CH$_3$CH$_2$COCH$_3$ + CH$_3$OH | 0.2 * 1.3 x 10$^{-12}$ | 3 |
| K131 | C$_4$H$_9$OOH + OH | → C$_4$H$_9$OO | 1.90 x 10$^{-12}$exp(190/T) | 3 |
| K132 | C$_4$H$_9$OOH + OH | → CH$_3$CH$_2$COCH$_3$ + OH | 2.15 x 10$^{-11}$ | 3 |
| K133 | C$_4$H$_9$ONO$_2$ + OH | → CH$_3$CH$_2$COCH$_3$ + NO$_2$ | 8.6 × 10$^{-13}$ | 1 |
| K134 | CH$_3$CH$_2$COCH$_3$ + OH | → CH$_3$CH(OO)COCH$_3$ | 1.5 x 10$^{-12}$exp(-90/T) | 1 |
| K135 | CH$_3$CH(OO)COCH$_3$ + HO$_2$ | → CH$_3$CH(OOH)COCH$_3$ | K126 | |
| K136 | CH$_3$CH(OO)COCH$_3$ + NO | → CH$_3$CHO + CH$_3$C(O)OO + NO$_2$ | (1 -RTC4S) x 2.55 x 10$^{-12}$ exp(380/T) | 1, 4 |
| K137 | CH$_3$CH(OO)COCH$_3$ + NO | → CH$_3$CH(ONO$_2$)COCH$_3$ | RTC4S x 2.55 x 10$^{-12}$ exp(380/T) | 1, 4 |
| K138 | CH$_3$CH(OOH)COCH$_3$ + OH | → CH$_3$CH(OO)COCH$_3$ | K131 | |
| K139 | CH$_3$CH(OOH)COCH$_3$ + OH | → CH$_3$C(O)C(O)CH$_3$ + OH | 1.88 x 10$^{-11}$ | 3 |
| K140 | CH$_3$CH(ONO$_2$)COCH$_3$ + OH | → CH$_3$C(O)C(O)CH$_3$ + NO$_2$ | 1.2 x 10$^{-12}$ | 1 |
| K141 | ISOP + OH | → 0.98 ISOPOO + 0.0003 ELVOC + 0.007 SVOC | 2.7 x 10$^{-11}$exp(390/T) | 1, 3 |
| K142 | ISOP + NO$_3$ | → ISOPONO$_2$ | 2.95 x 10$^{-12}$ exp(-450/T) | 1, 3 |
| K143 | ISOP + O$_3$ | → 0.98 * (0.3 MACR + 0.3 MACROO + 0.2 MVK + 0.2 MVKOO + 0.78 HCHO + 0.22CO + 0.125 HO$_2$ + 0.125OH) + 0.0001 ELVOC + 0.009 SVOC | 1.05 x 10$^{-14}$exp(-2000/T) | 1, 3 |
| K144 | ISOPOO + HO$_2$ | → ISOPOOH | 2.06 x 10$^{-13}$exp(1300/T) | 3, 7 |
| K145 | ISOPOO + NO | → HCHO + 0.64 MVK + 0.36 MACR + HO$_2$ + NO$_2$ | (1-RTC5S) * 2.7 x 10$^{-12}$exp(360/T) | 3 |
| K146 | ISOPOO + NO | → ISOPONO$_2$ | RTC5S * 2.7 x 10$^{-12}$exp(360/T) | 3 |
| K147 | ISOPOO + NO$_3$ | → HCHO + 0.64 MVK + 0.36 MACR + HO$_2$ + NO$_2$ | 2.3 x 10$^{-12}$ | 3 |
| K148 | ISOPOO + CH$_3$OO | → 0.64 MVK + 0.36 MACR + 2HCHO + 2HO$_2$ | 0.8 * 2.65 x 10$^{-12}$ | 3 |

| | | | | | |
|---|---|---|---|---|---|
| K149 | ISOPOO + CH$_3$OO | → | 0.64 MVK + 0.36 MACR + HCHO + CH$_3$OH | 0.2 * 2.65 x 10$^{-12}$ | 3 |
| K150 | ISOPOO | → | HPALD + HO$_2$ | 4.12×10$^8$exp(-7700/T) | 6, 7 |
| K151 | ISOPOOH + OH | → | IEPOX + OH | 1.9×10$^{-11}$exp(-390/T) | 8 |
| K152 | ISOPOOH + OH | → | ISOPOO | 0.7 * 3.8×10$^{-12}$exp(-200/T) | 8 |
| K153 | ISOPOOH + OH | → | 0.64 CH$_3$COCHO + 0.64 HOCH$_2$CHO + 0.36 HOCH$_2$C(O)CH$_3$ + 0.36 CHOCHO + OH | 0.3 * 3.8×10$^{-12}$exp(-200/T) | 8, 9 |
| K154 | ISOPONO$_2$ + OH | → | 0.64 CH$_3$COCHO + 0.64 HOCH$_2$CHO + 0.36 HOCH$_2$C(O)CH$_3$ + 0.36 CHOCHO + NO$_2$ | 1.77×10$^{-11}$exp(-500/T) | 8 |
| K155 | HPALD + OH | → | 0.5 HOCH$_2$C(O)CH$_3$ + 0.5 CH$_3$C(O)CHO + 0.25 HOCH$_2$CHO + 0.25 CHOCHO + HCHO + HO$_2$ + OH | 4.6×10$^{-11}$ | 6 |
| K156 | IEPOX + OH | → | IEPOXOO | 5.78×10$^{-11}$exp(-400/T) | 8 |
| K157 | IEPOXOO + HO$_2$ | → | 0.725 HOCH$_2$C(O)CH$_3$+ 0.275 HOCH$_2$CHO + 0.275 HOCH$_2$CHO + 0.275 CH$_3$C(O)CHO + 1.125 OH + 0.825 HO$_2$ + 0.2 CO$_2$ + 0.375 HCHO + 0.074 HCOOH + 0.251 CO | 7.4×10$^{-13}$exp(700/T) | 8 |
| K158 | IEPOXOO + NO | → | 0.725 HOCH$_2$C(O)CH$_3$+ 0.275 HOCH$_2$CHO + 0.275 HOCH$_2$CHO + 0.275 CH$_3$C(O)CHO + 1.125 OH + 0.825 HO$_2$ + 0.2 CO$_2$ + 0.375 HCHO + 0.074 HCOOH + 0.251 CO + NO$_2$ | 2.7×10$^{-12}$exp(360/T) | 3 |
| K159 | IEPOXOO + NO$_3$ | → | 0.725 HOCH$_2$C(O)CH$_3$+ 0.275 HOCH$_2$CHO + 0.275 HOCH$_2$CHO + 0.275 CH$_3$C(O)CHO + 1.125 OH + 0.825 HO$_2$ + 0.2 CO$_2$ + 0.375 HCHO + 0.074 HCOOH + 0.251 CO + NO$_2$ | 1.74 * 2.3×10$^{-12}$ | 3 |
| K160 | MVK + OH | → | MVKOO | 2.6 x 10$^{-12}$exp(610/T) | 1 |
| K161 | MVK + NO$_3$ | → | 0.65 HCOOH + 0.65 CH$_3$COCHO + 0.35 HCHO + 0.35 CH$_3$C(O)OOH + HNO3 | 6.0 x 10$^{-16}$ | 1 |
| K162 | MVK + O$_3$ | → | 0.38 CH$_3$COCHO + 0.2088 CH$_3$C(O)OO + 0.26 CH$_3$COCOOH + 0.26 CO + 0.0432 CH$_3$COOH + 0.108 CH$_3$CHO + 0.62 HCHO + 048 CO$_2$ + 0.54 HO$_2$ + 0.1008 OH | 8.5 x 10$^{-16}$exp(-1520/T) | 1, 3 |
| K163 | MVKOO + HO$_2$ | → | MVKOOH | K144 | |
| K164 | MVKOO + NO | → | 0.295 CH$_3$C(O)CHO + 0.295 HCHO + 0.670 CH$_3$CHO + 0.670 HOCHCHO + 0.295 HO$_2$ + 0.965 NO$_2$ + 0.0352 MVKONO$_2$ | 2.7 x 10$^{-12}$exp(360/T) | 3 |
| K165 | MVKOOH + OH | → | CH$_3$C(O)CHO + CO + 2HO$_2$ + OH | 2.55 x 10$^{-11}$ | 3 |
| K166 | MVKOOH + OH | → | MVKOO | 1.9 x 10$^{-12}$exp(190/T) | 3 |
| K167 | MVKONO$_2$ + OH | → | CH$_3$C(O)CHO + CO + HO$_2$ + NO$_2$ | 1.33 x 10$^{-12}$ | 3 |
| K168 | MACR + OH | → | MACROO | 8.0 x 10$^{-12}$exp(380/T) | 1 |
| K169 | MACR + NO$_3$ | → | MACROO + HNO$_3$ | 3.4 x 10$^{-15}$ | 1 |
| K170 | MACR + O$_3$ | → | 0.90 CH$_3$COCHO + 0.5 HCHO + 0.5 CO + 0.14 HO$_2$ + 0.24 OH | 1.4 x 10$^{-15}$exp(-2100/T) | 1, 3 |
| K171 | MACROO + HO$_2$ | → | MACROOH | 0.625 * 2.91 x 10$^{-13}$exp(1300/T) | 3 |
| K172 | MACROO + NO | → | 0.987 (CH$_3$COCH$_2$OH + CO + NO$_2$ + HO$_2$) + 0.013 MACRONO$_2$ | K164 | 1, 3 |
| K173 | MACROOH + OH | → | CH$_3$COCH$_2$OH + CO + OH | 3.77 x 10$^{-11}$ | |

[revised manuscript text omitted]

**Supplementary Tables**

**Table S1: Selection of effective Henry law coefficients (H\*) used in TM5-MP for the MOGUNTIA chemical scheme.**

| Trace gas | H* (M atm⁻¹) | ΔH R⁻¹ (K) | Reference |
|---|---|---|---|
| $CH_3OOH$, $n$-$C_3H_7OOH$, $i$-$C_3H_7OOH$, $CH_3COCH_2OH$, $C_4H_9OOH$, MEKOOH, ISOPOOH, MVKOOH, MACROOH | $2.9 \times 10^2$ | 5200 | 1 |
| $CH_3ONO_2$ | 2.0 | 4700 | 1 |
| $CH_3OONO_2$ | 2.0 | 4700 | 1 |
| HCHO | $3.2 \times 10^3$ | 6800 | 1 |
| $CH_3OH$ | $2.0 \times 10^2$ | 5600 | 1 |
| HCOOH | $8.8 \times 10^3$ | 6100 | 1 |
| $CH_3CH_2OOH$ | 3.3 | 6000 | 1 |
| $CH_3CH_2ONO_2$ | 1.6 | 5400 | 1 |
| $HOCH_2CH_2OOH$ | $1.7 \times 10^6$ | 9700 | 1 |
| $HOCH_2CH_2ONO_2$ | $3.9 \times 10^4$ | | 1 |
| $CH_3CHO$ | 13 | 5900 | 1 |
| $CH_3COOH$ | $8.3 \times 10^2$ | 5300 | 1 |
| $HOCH_2CHO$ | $4.1 \times 10^4$ | 4600 | 1 |
| CHOCHO | $4.19 \times 10^5$ | 7500 | 1 |
| $CH_3CH_2OH$ | 190 | 6400 | 1 |
| $CH_3COOH$ | $4.0 \times 10^3$ | 6200 | 1 |
| $n$-$C_3H_7ONO_2$ | 1.1 | 5500 | 1 |
| $i$-$C_3H_7ONO_2$ | 0.78 | 5400 | 1 |
| $HOC_3H_6OOH$ | $1.7 \times 10^6$ | 9700 | 1 |
| $CH_3COCH_3$ | 27 | 5500 | 1 |
| $CH_3CH_2CHO$ | 9.9 | 4300 | 1 |
| $CH_3COCHO$ | $3.2 \times 10^3$ | 7500 | 1 |
| $CH_3C(O)COOH$ | $3.1 \times 10^5$ | 5100 | 1 |
| $C_4H_9ONO_2$ | 1 | 5800 | 1 |
| MEK | 18 | 5700 | 1 |
| $MEKONO_2$ | 0.7 | 5200 | 1 |
| $CH_3COCOCH_3$ | 73 | 5700 | 1 |
| $ISOPONO_2$, $MACRONO_2$, $MVKONO_2$ | $1.7 \times 10^4$ | 9200 | 2 |
| IEPOX | $9.1 \times 10^4$ | 6600 | 3 |
| HPALD | 2.3 | | 1 |
| MVK | 26 | 4800 | 1 |
| MACR | 4.8 | 4300 | 1 |

[1] Sander (2015) and references therein
[2] Ito et al. (2007) for all biogenic hydroxy nitrates
[3] Browne et al. (2014), as for $H_2O_2$

**Table S2: Soil, water, snow/ice and mesophyl resistances (s m$^{-1}$) used in TM5-MP for the CB05 and MOGUNTIA chemical schemes.**

| Trace gas | $r_{soil}$ | $r_{wat}$ | $r_{snow/ice}$ | $r_{mes}$ | $r_{cut}$ |
|---|---|---|---|---|---|
| $O_3$ | 400 | 2000 | 2000 | 1 | $10^5$ |
| CO | 5000 | $10^5$ | $10^5$ | 5000 | $10^5$ |
| NO | $10^5$ | $10^5$ | $10^5$ | 500 | $10^5$ |
| $NO_2/NO_3$ | 600 | 3000 | 3000 | 1 | $10^5$ |
| $HNO_3/N_2O_5$ | 1 | 1 | 1 | 1 | 1 |
| $H_2O_2$, IEPOX | 80 | 72 | 80 | 1 | $10^5$ |
| $SO_2$ | 100 | 1 | 1 | 1 | $10^5$ |
| $CH_3ONO_2$, $CH_3OONO_2$, $CH_3C(O)OONO_2$, $n$-$C_3H_7ONO_2$, $i$-$C_3H_7ONO_2$, $C_4H_9ONO_2$, MEKONO$_2$, ISOPONO$_2$ | 3994 | 295 | 3394 | 1 | $10^5$ |
| $CH_3CHO$, $C_2H_5CHO$, $CH_3C(O)CH_3$, $CH_3C(O)C(O)CH_3$, $HOCH_2C(O)CH_3$, MEK, MVK, MACR, HPALD | $10^5$ | 300 | $10^5$ | 200 | $10^5$ |
| HCHO, $CH_3COCHO$, CHOCHO, $HOCH_2CHO$, | 1666 | 254 | 1666 | 1 | $10^5$ |
| $CH_3OOH$, $CH_3OH$, HCOOH, $CH_3CH_2OOH$, $CH_3CH_2OH$, $CH_3COOH$, n-$C_3H_7OOH$, i-$C_3H_7OOH$, $CH_3C(O)CH_2OOH$, $n$-$C_3H_7OOH$, $i$-$C_3H_7OOH$, HOC$_3H_6OOH$, $CH_3C(O)COOH$, $C_4H_9OOH$, MEKOOH, MVKOOH, MACROOH, $CH_3C(O)OOH$, ISOPOOH | 3650 | 293 | 3650 | 1 | $10^5$ |
| $NH_3$ | 100 | 1 | $10^5$ | 1 | $10^5$ |

**Table S3: TM5-MP performance calculations of the mCB05(EBI), mCB05(KPP) and MOGUNTIA configurations for the different components, i.e., the transport (advection in the x-, y- and z-directions along with the vertical transport), the chemistry as well as all other procedures contribution, the simulated years per day (SYPD), and the core-hours per simulated years (CHPSY) using a) 360 cores, and b) 450 cores. Timings are in seconds and changes are in %. In parentheses, the runtime and the SYPD without the meteorology reading are also presented. All simulations have been performed in the ECMWF CRAY XC40 high-performance computer facility.**

a)

| 360 cores / Configuration | Transport Adv$_x$ | Adv$_y$ | Adv$_z$ | Vertical | Total | Chemistry | Other | Runtime | SYPD | CHPSY |
|---|---|---|---|---|---|---|---|---|---|---|
| CB05(EBI) | 1322 | 948 | 165 | 364 | 2799 | 3338 | 3925 | 10062 (6723) | 0.73 (1.10) | 12000 |
| CB05(KPP) | 1312 | 934 | 165 | 362 | 2773 | 5301 | 4222 | 12296 (9105) | 0.60 (0.81) | 14000 |
| MOGUNTIA | 1892 | 1303 | 233 | 527 | 3955 | 8230 | 4680 | 16865 (13556) | 0.44 (0.54) | 20000 |
| % solver changes | -1% | -1% | 0% | -1% | -1% | -1% | 59% | 8% (35%) | -18% (-26%) | 17% |
| % chemistry scheme changes | 44% | 40% | 41% | 46% | 43% | 43% | 55% | 11% (49%) | -27% (-33%) | 43% |

b)

| 450 cores / Configuration | Transport Adv$_x$ | Adv$_y$ | Adv$_z$ | Vertical | Total | Chemistry | Other | Runtime | SYPD | CHPSY |
|---|---|---|---|---|---|---|---|---|---|---|
| CB05(EBI) | 1268 | 860 | 138 | 292 | 2558 | 2639 | 3687 | 8884 (5696) | 0.83 (1.30) | 13000 |
| CB05(KPP) | 1292 | 853 | 133 | 300 | 2578 | 4320 | 4079 | 10977 (7733) | 0.67 (0.95) | 16000 |
| MOGUNTIA | 1806 | 1126 | 193 | 423 | 3548 | 6526 | 4376 | 14450 (11211) | 0.51 (0.65) | 21000 |
| % solver changes | 2% | -1% | -4% | 3% | 1% | 64% | 11% | 24% (36%) | -19% (-27%) | 23% |
| % chemistry scheme changes | 40% | 32% | 45% | 41% | 38% | 51% | 7% | 32% (45%) | -24% (-32%) | 31% |

**Table S4: Tropospheric chemical budget of ORGNTR[a] for the year 2006 in Tg(N) yr$^{-1}$, using the 150 ppb O$_3$ mixing ratio to define tropopause level. Tropospheric burdens in Gg(N) yr$^{-1}$.**

| Production terms | mCB05 (EBI) | mCB05 (KPP) | MOGUNTIA | Loss terms | mCB05 (EBI) | mCB05 (KPP) | MOGUNTIA |
|---|---|---|---|---|---|---|---|
| XO$_2$N/RO$_2$ + NO | 8.6 | 8.1 | 7.0 | ORGNTR + hv | 4.1 | 4.0 | 2.6 |
| RH + NO$_3$ | 4.3 | 4.2 | 6.7 | ORGNTR + OH | 1.3 | 1.4 | 5.8 |
| Tropospheric Burden | 159.6 | 159.8 | 63.0 | Deposition | 7.4 | 7.6 | 5.1 |

[a]*For the MOGUNTIA configuration ORGNTR represents the sum of CH$_3$ONO$_2$, C$_2$H$_5$ONO$_2$, OHCH$_2$CH$_2$ONO$_2$, CH$_3$CH$_2$CH$_2$ONO$_2$, CH$_3$CH(ONO$_2$)CH$_3$, CH$_3$CH$_2$CH(ONO$_2$)CH$_3$, nitrates from isoprene (ISOPNO$_3$), nitrates from methyl-ethyl ketone (MEKNO$_3$,), nitrates from methyl vinyl ketone (MVKNO$_3$) and nitrates from methacrolein (MACRNO$_3$)*

**Supplementary Equations**

Statistics Formulas: Correlation coefficient (R; Eq. S1), mean normalized bias (MNB; Eq. S2), root mean square error (RMSE; Eq. S3), mean normalized error (MNE; Eq. S4) and standard error (STD; Eq. S5) values have been calculated to compare the model calculations, where $O_i$ and $P_i$ stand for observations and predictions respectively and N is the number of pairs (observations, predictions) that are compared.

$$R = \left[ \frac{\frac{1}{N}\sum_{i=1}^{N}(O_i - \overline{O})(P_i - \overline{P})}{\sigma_O \sigma_P} \right] \qquad \text{(Eq. S1)}$$

$$NMB = \frac{\sum_{i=1}^{N}(M_i - O_i)}{\sum_{i=1}^{N} O_i} \times 100 \qquad \text{(Eq. S2)}$$

$$RMSE = \sqrt{\frac{1}{N}\sum_{i=1}^{N}(P_i - O_i)^2} \qquad \text{(Eq. S3)}$$

$$NME = \frac{\sum_{i=1}^{N}|M_i - O_i|}{\sum_{i=1}^{N} O_i} \times 100 \qquad \text{(Eq. S4)}$$

$$STD = \frac{\sqrt{\frac{1}{N}\sum_{i=1}^{N}(O_i - \overline{O})^2}}{\sqrt{N}} \qquad \text{(Eq. S5)}$$

Field Code Changed

Field Code Changed

Field Code Changed

Field Code Changed

**Supplementary Figures**

a)

b)

[Figure]

**Figure S_1: Comparison of simulated a) tropospheric NO₂ columns with OMI retrievals from the QA4ECV dataset and b) simulated total CO columns with MOPITT retrievals (vers. MOP02J_V008) for the year 2006. Green, orange, and blue bars show the comparison of OMI with the MOGUNTIA, mCB05(KPP), and mCB05(EBI) chemistry mechanisms, respectively: Pearson correlation coefficient (top left), root mean square error (top right), mean bias (measurement minus model, bottom left), and normalized mean bias (measurement minus model, bottom right) are given for both daily (D) and yearly (Y) averages per model grid cell.**

[Figure]

**Figure S2: Simulated annual mean surface (left columns) and zonal mean (right columns) mixing ratios (ppb) of organic nitrates (ORGNTR) for the MOGUNTIA chemistry scheme for the year 2006 (a,b), and the respective differences compared to mCB05(KPP) (c,d). For the MOGUNTIA configuration, ORGNTR represents the sum of $CH_3ONO_2$, $C_2H_5ONO_2$, $OHCH_2CH_2ONO_2$, $CH_3CH_3CH_2ONO_2$, $CH_3CH(ONO_2)CH_3$, $CH_3CH_2CH(ONO_2)CH_3$, nitrates from isoprene ($ISOPNO_3$), nitrates from methyl-ethyl ketone ($MEKNO_3$,), nitrates from methyl vinyl ketone ($MVKNO_3$) and nitrates from methacrolein ($MACRNO_3$)."**

[Figure]

[Figure]

[Figure]

[Figure]

[Figure]

[Figure]

[Figure]

[Figure]

**db)** Rao, Sweden (58° N, 12° E)

**dc)** Norrakvill, Sweden (57° N, 15° E)

**dd)** Vindeln, Sweden (64° N, 19° E)

**de)** Grimso, Sweden (60° N, 15° E)

**df)** Iskrba, Slovenia (45° N, 14° E)

**dg)** Zarodnje, Slovenia (47° N, 15° E)

**dh)** Krvavec, Slovenia (47° N, 14° E)

**di)** Kovk, Slovenia (46° N, 15° E)

**dj)** Stara Lesna, Slovakia (49° N, 20° E)

**dk)** Liesek, Slovakia (49° N, 19° E)

**dl)** Topolniky, Slovakia (48° N, 18° E)

**dm)** Ahtari, Finland (62° N, 24° E)

**dn)** Arrival Heights, New Zealand (77° S, 166° E)

**do)** Assekrem, Algeria (23° N, 5° E)

**dp)** Baring Head, New Zealand (41° S, 174° E)

OBS    mCB05(EBI)    mCB05(KPP)    MOGUNTIA

[Figure]

**dq)** Bukit Koto Tabang, Indonesia (0° S, 100° E)

mCB05(EBI): R=-0.39, BIAS=29.9 ppb, NME=235.7 %
mCB05(KPP): R=-0.39, BIAS=29.2 ppb, NME=229.5 %
MOGUNTIA: R=-0.39, BIAS=26.3 ppb, NME=207.0 %

**dr)** Tudor Hill, UK (32° N, 64° W)

mCB05(EBI): R=-0.48, BIAS=18.3 ppb, NME=47.5 %
mCB05(KPP): R=-0.51, BIAS=16.1 ppb, NME=42.1 %
MOGUNTIA: R=-0.44, BIAS=17.6 ppb, NME=45.6 %

**ds)** Barrow, USA (71° N, 156° W)

mCB05(EBI): R=0.43, BIAS=7.5 ppb, NME=40.1 %
mCB05(KPP): R=-0.45, BIAS=7.2 ppb, NME=39.5 %
MOGUNTIA: R=-0.41, BIAS=8.2 ppb, NME=40.2 %

**dt)** Cape Point, South Africa (34° S, 18° E)

mCB05(EBI): R=0.95, BIAS=8.0 ppb, NME=34.5 %
mCB05(KPP): R=0.96, BIAS=8.0 ppb, NME=34.6 %
MOGUNTIA: R=0.96, BIAS=6.3 ppb, NME=27.0 %

**du)** Cape Point, South Africa (34° S, 18° E)

mCB05(EBI): R=0.95, BIAS=8.0 ppb, NME=34.5 %
mCB05(KPP): R=0.96, BIAS=8.0 ppb, NME=34.6 %
MOGUNTIA: R=0.96, BIAS=6.3 ppb, NME=27.0 %

**dv)** Giordan Lighthouse, Malta (36° N, 14° E)

mCB05(EBI): R=0.65, BIAS=6.9 ppb, NME=24.5 %
mCB05(KPP): R=0.65, BIAS=4.2 ppb, NME=24.6 %
MOGUNTIA: R=0.69, BIAS=5.1 ppb, NME=22.4 %

**dw)** Hohenpeissenberg, Germany (47° N, 11° E)

mCB05(EBI): R=0.59, BIAS=-0.8 ppb, NME=14.7 %
mCB05(KPP): R=0.59, BIAS=-1.7 ppb, NME=13.5 %
MOGUNTIA: R=0.62, BIAS=-0.1 ppb, NME=13.5 %

**dx)** Heimaey, Iceland (63° N, 20° W)

mCB05(EBI): R=0.30, BIAS=-4.6 ppb, NME=17.2 %
mCB05(KPP): R=0.31, BIAS=-5.4 ppb, NME=17.5 %
MOGUNTIA: R=0.48, BIAS=-4.1 ppb, NME=14.1 %

**dy)** Iskrba, Slovenia (45° N, 14° E)

mCB05(EBI): R=0.45, BIAS=13.2 ppb, NME=42.6 %
mCB05(KPP): R=0.44, BIAS=12.5 ppb, NME=39.9 %
MOGUNTIA: R=0.49, BIAS=13.5 ppb, NME=43.5 %

**dz)** Issykkul, Kyrgyzstan (42° N, 76° E)

mCB05(EBI): R=0.78, BIAS=6.0 ppb, NME=16.0 %
mCB05(KPP): R=0.77, BIAS=4.4 ppb, NME=15.1 %
MOGUNTIA: R=0.74, BIAS=5.1 ppb, NME=13.5 %

**ea)** Tenerife, Spain (28° N, 16° W)

mCB05(EBI): R=0.77, BIAS=-0.5 ppb, NME=5.4 %
mCB05(KPP): R=0.73, BIAS=-3.5 ppb, NME=11.4 %
MOGUNTIA: R=0.77, BIAS=-1.0 ppb, NME=9.0 %

**eb)** Jungfraujoch, Switzerland (46° N, 7° E)

mCB05(EBI): R=0.72, BIAS=-9.1 ppb, NME=20.1 %
mCB05(KPP): R=0.72, BIAS=-10.0 ppb, NME=21.6 %
MOGUNTIA: R=0.72, BIAS=-8.6 ppb, NME=18.2 %

**ec)** Kollumerwaard, The Netherlands (53° N, 6° E)

mCB05(EBI): R=0.81, BIAS=13.6 ppb, NME=52.6 %
mCB05(KPP): R=0.80, BIAS=12.6 ppb, NME=49.4 %
MOGUNTIA: R=0.81, BIAS=14.7 ppb, NME=56.8 %

**ed)** Kosetice, Czech Republic (49° N, 15° E)

mCB05(EBI): R=0.56, BIAS=7.2 ppb, NME=24.5 %
mCB05(KPP): R=0.56, BIAS=6.5 ppb, NME=23.0 %
MOGUNTIA: R=0.61, BIAS=8.3 ppb, NME=25.3 %

**ee)** Kovk, Slovenia (46° N, 15° E)

mCB05(EBI): R=0.68, BIAS=8.0 ppb, NME=22.7 %
mCB05(KPP): R=0.68, BIAS=7.1 ppb, NME=26.8 %
MOGUNTIA: R=0.67, BIAS=8.5 ppb, NME=23.9 %

• • OBS          mCB05(EBI)          mCB05(KPP)          MOGUNTIA

[Figure]

[Figure]

[Figure]

**Figure S3: Comparison of monthly mean surface O₃ observations (black dots) in ppb with model results (red-line for mCB05(EBI), green-line for mCB05(KPP) and blue-line for MOGUNTIA) at various stations around the globe, as obtained from the European Monitoring and Evaluation Programme (EMEP; http://www.emep.int) and the World Data Centre for Greenhouse Gases (WDCGG; http://ds.data.jma.go.jp/gmd/wdcgg/introduction.html), for the year 2006.**

[Figure]

[Figure]

[Figure]

[Figure]

**Figure S4: Comparison of monthly mean ozone sonde observations (black line) in ppb with model results (red-line for mCB05 configuration using the EBI solver, green-line for mCB05 configuration using the solver as generated by the KPP software and blue-line for MOGUNTIA configuration) at various stations around the globe, as obtained from the World Data Centre for Greenhouse Gases (WDCGG; https://gaw.kishou.go.jp), for the year 2006.**

a)

[Figure]

b)

c)

d)

5    **Figure S5: Monthly mean comparisons of TM5-MP UTLS O$_3$ (top) and CO (bottom) mixing ratios (ppb) for the two chemistry schemes; mCB05(KPP) (blue line) and MOGUNTIA (red line), sampled at the measurement place and time against MOZAIC flight data (black line) between Frankfurt (50.0º N, 8.6º E) and Windhoek (22.5º S, 17.7º E) for April (left column) and October 2006 (right column). Data at pressures (P) lower than 300 hPa has been filtered out.**

[Figure]

[Figure]

[Figure]

[Figure]

[Figure]

[Figure]

[Figure]

[Figure]

**Figure S6: Comparison of monthly mean surface CO flask measurements (black dots) in ppb with model results (red-line for mCB05(EBI), green-line for mCB05(KPP) and blue-line for MOGUNTIA) at various stations around the globe, as obtained from NOAA database, for the year 2006.**

[Figure]

**Figure S7: Monthly mean comparison of TM5-MP surface $C_2H_6$ (left column) and $C_3H_8$ (right column) using the base case emission scenario, doubling (2x) of the anthropogenic fossil fuel emissions, and quadrupling (4x) of the anthropogenic fossil fuel emissions of $C_2H_6$ and $C_3H_8$, against flask measurements (black dots) in ppt for the MOGUNTIA chemistry scheme (green line), using co-located model output for 2006 sampled at the measurement times. Shaded areas indicate the range of model results due to the different emission strengths. For this sensitivity analysis, the model runs in 3º x 2º horizontal resolution in longitude by latitude, and 34 hybrid levels in the vertical.**

[Figure]

[Figure]

[Figure]

[Figure]

[Figure]

**Figure S8: Comparison of monthly mean surface C$_2$H$_6$ flask measurements (black dots) in ppb with model results (red-line for mCB05(EBI), green-line for mCB05(KPP) and blue-line for MOGUNTIA) at various stations around the globe, as obtained from NOAA database, for the year 2006.**

[Figure]

[Figure]

OBS     mCB05(EBI)     mCB05(KPP)     MOGUNTIA

[Figure]

OBS    mCB05(EBI)    mCB05(KPP)    MOGUNTIA

[Figure]

**at)** Cold Bay Alaska
United States (55.21° N, 162.72° W)

**au)** Mahe Island
Seychelles (4.682° S, 55.532° E)

**av)** Trinidad Head California
United States (41.054° N, 124.151° W)

**aw)** Ochsenkopf
Germany (50.03° N, 11.808° E)

•  •  OBS    •——•  mCB05(EBI)    +——+  mCB05(KPP)    – –  MOGUNTIA

**Figure S9: Comparison of monthly mean surface propane flask measurements (black dots) in ppb with model results (red-line for mCB05(EBI), green-line for mCB05(KPP) and blue-line for MOGUNTIA) at various stations around the globe, as obtained from NOAA database, for the year 2006.**

[Figure]

[Figure]

[Figure]

[Figure]

[Figure]

**bi)** Ethane (20N - 40N, 50W - 30W)
Japan, TRACEPP3

[Figure]

[Figure]

OBS   •—• mCB05(EBI)   ╉ mCB05(KPP)   – – MOGUNTIA

**Figure S10: Comparison of TM5-MP vertical profiles (in km) of $C_2H_6$ against aircraft observations (black line) in ppt with model results (red-line for mCB05(EBI), green-line for mCB05(KPP) and blue-line for MOGUNTIA), using co-located model output for 2006 sampled at the measurement times; error bars indicate the standard deviation. The numbers on the right vertical axis indicate the number of available measurements.**

[Figure]

[Figure]

[Figure]

[Figure]

[Figure]

[Figure]

**Figure S11: Comparison of TM5-MP vertical profiles (in km) of $C_3H_8$ against aircraft observations (black line) in ppt, with model results (red-line for mCB05(EBI), green-line for mCB05(KPP) and blue-line for MOGUNTIA), using co-located model output for 2006 sampled at the measurement times; error bars indicate the standard deviation. The numbers on the right vertical axis indicate the number of available measurements.**

[Figure]

[Figure]

[Figure]

[Figure]

[Figure]

**Figure S12: Comparison of TM5-MP vertical profiles (in km) of C$_2$H$_4$ against aircraft observations (black line) in ppt, with model results (red-line for mCB05(EBI), green-line for mCB05(KPP) and blue-line for MOGUNTIA), using co-located model output for 2006 sampled at the measurement times; error bars indicate the standard deviation. The numbers on the right vertical axis indicate the number of available measurements.**

[Figure]

[Figure]

[Figure]

[Figure]

[Figure]

[Figure]

Figure S13: Comparison of TM5-MP vertical profiles (in km) of C$_3$H$_6$ against aircraft observations (black line) in ppt, with model results (red-line for mCB05(EBI), green-line for mCB05(KPP) and blue-line for MOGUNTIA), using co-located model output for 2006 sampled at the measurement times; error bars indicate the standard deviation. The numbers on the right vertical axis indicate
5  the number of available measurements.

**Supplementary References**

Browne, E. C., Wooldridge, P. J., Min, K.-E. and Cohen, R. C.: On the role of monoterpene chemistry in the remote continental boundary layer, Atmos. Chem. Phys., 14(3), 1225–1238, doi:10.5194/acp-14-1225-2014, 2014.

Ito, A., Sillman, S. and Penner, J. E.: Effects of additional nonmethane volatile organic compounds, organic nitrates, and direct emissions of oxygenated organic species on global tropospheric chemistry, J. Geophys. Res., 112(D6), D06309, doi:10.1029/2005JD006556, 2007.

Sander, R.: Compilation of Henry's law constants (version 4.0) for water as solvent, Atmos. Chem. Phys., 15(8), 4399–4981, doi:10.5194/acp-15-4399-2015, 2015.

---

## Author Response (AR3)

We thank the reviewer for appreciating our efforts to improve the manuscript. Please find bellow our point-by-point replies to the new comments.

**General Comments**

**GC1.** **I agree with the authors, when justifying the use of many sources on page 4 line 19-22, that the important sources that report the model development should be cited here. However, some of the listed sources do not fall into this category and, by the authors own definition, should not be cited when presenting the historical development of the model. In particular, Dentener et al., 2003, and van Noije, 2004 do not present any obvious model development and are not cited anywhere else in the manuscript. At a minimum these should be removed from this part of the manuscript and all other citations should be evaluated regarding their contribution to the model development.**

- We agree with the reviewer. We removed the Dentener et al., 2003 reference since it mainly used the Houweling et al. (1998)' version of the model. On the other hand, the van Noije et al. (2004) is the first publication that describes the new developments adopted in TM4 (which is different from the TM3 model). However, based on the reviewer's comments, we also removed the Daskalakis et al. (2003) reference, since it mainly uses the Myriokefalitakis et al. (2008) version of the model which presented the implementation of a new chemistry version in the TM4 model.

**GC2.** **I agree with the authors that using the same emission dataset for all simulations (as you clearly state on page 10 line 28) is the right approach. However, there are still some formulations in the manuscript that are misleading and could be misunderstood. For example, page 10, line 20-21: "A list of the global annual emission strengths considered for the MOGUNTIA chemical configuration is presented in Table 3." could imply that other emissions are used for the non-MOGUNTIA simulations. The caption of Table 3 gives the same impression. Please revise these statements and check if there are other formulations in the manuscript that could be misleading.**

- We want to clarify again that although the same emission databases are used in the model, the resulting emission strengths cannot be exactly the same, since different chemical mechanisms consider underline{different chemical species}. For example, Table 3 presents the emission strengths as calculated in the model for the MOGUNTIA chemistry configuration. When the model uses the mCB05 configuration instead, the same emissions would be represented by different lumped species (using, thus, different molecular weights), giving overall not directly comparable emission strengths. For instance:

**Table: Global annual emissions of trace gases used for the mCB05 chemistry scheme in TM5-MP for the year 2006, in Tg yr$^{-1}$ unless specified otherwise.**

| Species | Long name | Emissions |
|---|---|---|
| CO | carbon monoxide | 1097 |
| HCHO | formaldehyde | 12.3 |
| HCOOH | formic acid | 9.8 |
| CH$_3$OH | methanol | 146.4 |
| PAR [*] | paraffinic carbon atoms | 67.2 |
| C$_2$H$_6$ | ethane | 10.9 |
| C$_2$H$_4$ | ethene | 29.8 |
| ALD2 [$] | acetaldehyde and higher aldehydes | 13.2 |
| CH$_3$COOH | acetic acid | 26.1 |
| CH$_3$CH$_2$OH | ethanol | 19.3 |
| C$_3$H$_8$ | propane | 8.5 |
| C$_3$H$_6$ | propene | 24.3 |
| CH$_3$COCH$_3$ | acetone | 42.1 |

| | | | |
|---|---|---|---|
| $CH_3COCHO$ | methylglyoxal | | 5.0 |
| OLE [$] | olefinic carbon bonds | | 4.3 |
| $C_5H_8$ | isoprene | | 579.4 |
| $C_{10}H_{16}$ | monoterpenes | | 97.9 |
| $NO_x$ [#] | nitrogen oxides | | 59.9 |
| $NH_3$ | ammonia | | 70.9 |
| $SO_2$ | sulfur dioxide | | 132.1 |
| $CH_3SCH_3$ | dimethylsulphide | | 97.5 |

[*] in Tg-C yr $^{-1}$
[$] in Tg-$C_2$ yr $^{-1}$
[#] in Tg-N yr $^{-1}$

For this, we clearly state in the manuscript that "*A list of the global annual emission strengths considered for the **MOGUNTIA** chemical configuration is presented in Table 3*". A generic statement such as "*A list of the global annual emission strengths considered in the model*" would be indeed misleading. Hence, we keep this as it is in the current version of the manuscript.

**GC3.  You changed/updated some rate constants in Table 2 (Page 44 – 51). In your simulations, did you use the values presented in the first or in the new version of the manuscript? If you didn't re-simulate all simulations using the new values, what impact do you expect these changes have on your results?**

- Indeed, we have corrected some typos in Table 2 that did not agree either with the respective references or with the rate constants applied in the model. We have also (re-)checked all values and reactions provided in all Tables, and now we hope that no more differences/typos exist between the Tables, the documented references, and the model. It goes without saying that we carefully checked that the results presented in the paper comply with the rate values presented in the tables.

**Specific Comments**

**SC1) Page 4, line 30: Correct notation of radon-222. Here, 222 should be in superscript-format.**

- We thank the reviewer for attracting our attention to this issue. For an unexplained reason, the super- and sub-script formats are not properly shown in the pdf file, although they are correctly displayed in the docx file, upon conversion. We recreated the pdf files and now they seem correct. Nevertheless, we could also provide the docx files to avoid similar problems.

**SC2) Page 5, line 13: Please mention that you also present values for the 100 ppb tropopause definition, in order to allow comparability with other studies.**

- We propose to add the following sentence: "*Moreover, budget results using the 100 ppb $O_3$ mixing ratios (e.g., Lamarque et al., 2012) as a tropopause level in the model are also provided.*"

**SC3) Page 23, line 14: The "with the" should be deleted.**

- Deleted

**SC4) In many instances the degree sign is in subscript-format where it should be clearly superscript-format. Please adapt the manuscript accordingly.**

- Please see our reply to SC1.

[revised manuscript text omitted]